# DeformTime: capturing variable dependencies with deformable attention for time series forecasting

**Yuxuan Shu**  *yuxuan.shu.22@ucl.ac.uk*
*Centre for Artificial Intelligence*
*Department of Computer Science*
*University College London*

**Vasileios Lampos**  *v.lampos@ucl.ac.uk*
*Centre for Artificial Intelligence*
*Department of Computer Science*
*University College London*

**Reviewed on OpenReview:** *https://openreview.net/forum?id=M62P7iOT7d*

## Abstract

In multivariable time series (MTS) forecasting, existing state-of-the-art deep learning approaches tend to focus on autoregressive formulations and often overlook the potential of using exogenous variables in enhancing the prediction of the target endogenous variable. To address this limitation, we present DEFORMTIME, a neural network architecture that attempts to capture correlated temporal patterns from the input space, and hence, improve forecasting accuracy. It deploys two core operations performed by deformable attention blocks (DABs): learning dependencies across variables from different time steps (variable DAB), and preserving temporal dependencies in data from previous time steps (temporal DAB). Input data transformation is explicitly designed to enhance learning from the deformed series of information while passing through a DAB. We conduct extensive experiments on 6 MTS data sets, using previously established benchmarks as well as challenging infectious disease modelling tasks with more exogenous variables. The results demonstrate that DEFORMTIME improves accuracy against previous competitive methods across the vast majority of MTS forecasting tasks, reducing the mean absolute error by 7.2% on average. Notably, performance gains remain consistent across longer forecasting horizons.

## 1 Introduction

Time series forecasting models provide multifaceted solutions for many application domains, including health (Shaman & Karspeck, 2012; Ioannidis et al., 2022), climate (Reichstein et al., 2019), energy (Deb et al., 2017), transport (Li et al., 2018), and finance (Sezer et al., 2020). In their pursuit for greater accuracy, forecasting methods have always been trying to benefit from multimodal exogenous predictors (De Gooijer & Hyndman, 2006; Athanasopoulos et al., 2011). Over recent years, a plethora of new digitised information sources has become available, accompanied by a rapid development of capable and efficient deep learning architectures (Rangapuram et al., 2018; Lai et al., 2018). Current state-of-the-art (SOTA) time series forecasting models are drawing particular focus on the deployment of many-to-many problem formulations (Nie et al., 2023; Wu et al., 2023; Luo & Wang, 2024b; Wang et al., 2024a). However, this approach does not necessarily provide the right learning mechanism for multivariable time series (MTS) forecasting (Hidalgo & Goodman, 2013), where we have a definitive target (endogenous) variable and a considerable amount of external (exogenous) input variables. Furthermore, although various neural network (NN) architectures were proposed to address forecasting challenges with multiple inputs (Zhang & Yan, 2023; Yi et al., 2023a; Huang et al., 2023; Luo & Wang, 2024b), many (Zeng et al., 2023; Nie et al., 2023; Jia et al., 2023; Lee et al., 2024;

Jin et al., 2024; Lin et al., 2024) were not designed to incorporate inter-variable dependencies, a key property of conventional forecasting models (Pi & Peterson, 1994; Hyndman & Ullah, 2007; Bontemps et al., 2008).

There certainly exist neural forecasting models that attempt to leverage dependencies across variables and time. Solutions based on recurrent neural network (RNN) architectures (Lai et al., 2018; Lin et al., 2023) can capture information from previous time steps (up to an extent), but have a limited capacity in establishing inter-variable dependencies. LightTS (Zhang et al., 2022) uses multi-layer perceptrons to model dependencies within the input data, first along the temporal dimension and then across variables. For transformer-based models, Crossformer (Zhang & Yan, 2023) proposes to first partition the input into temporal patches, and then use a router module to capture information across variables from different time steps, improving performance compared to pre-established RNN and Graph NN architectures (Lai et al., 2018; Wu et al., 2020). However, both LightTS and Crossformer have been outperformed by later proposed models that do not establish dependencies between variables (Zeng et al., 2023; Nie et al., 2023; Jia et al., 2023; Lee et al., 2024; Jin et al., 2024; Lin et al., 2024). The counterargument from these approaches is that models without inter-variable dependency modules benefit from being able to use longer look-back windows without significantly increasing model complexity (Han et al., 2023). Nevertheless, a common issue throughout the deep learning forecasting literature that casts doubt on some of these conclusions is the inappropriateness of various benchmark tasks (see Appendix A.4). In addition, recent work has argued that to improve performance while exploiting inter-variable dependencies, more effort is required in temporally aligning the input time series (Zhao & Shen, 2024). Attempts have also been made to capture dependencies with large receptive fields within segmented time series patches (Luo & Wang, 2024b). Motivated by this, we have introduced guided re-arrangements of the input to better capture inter- and intra-variable dynamics.

Deformable neural networks were initially proposed in computer vision (Dai et al., 2017) to accommodate geometric transformations with Convolutional Neural Networks (CNNs). Combined with transformer-based NN architectures, deformation has achieved SOTA performance in various tasks (Zhu et al., 2021; Chen et al., 2021; Xia et al., 2022). Prior work (Wang et al., 2024c) has also deployed deformable mechanisms within time series forecasting. However, their application was limited to establishing intra-variable dependencies, a structural design also shared in (Luo & Wang, 2024a), making them less effective in capturing correlations between multiple variables over time. Importantly, this model was outperformed by later ones (Nie et al., 2023; Liu et al., 2024). Based on the aforementioned insights, this paper introduces DEFORMTIME, a novel MTS forecasting model that deploys a deformable module on top of transformer encoders to introduce some flexibility in the determination of receptive fields across different variables and time steps. The premise of DEFORMTIME is the inclusion of learnable mechanisms that we refer to as deformable attention blocks (DABs). These enable transformations of the input information stream that enhance learning from key patterns within endo- but most importantly exogenous variables, an operation that ultimately improves forecasting accuracy (for the target endogenous variable). Additionally, input is transformed in alignment with the operational goal of the DAB modules, guiding the modelling of dependencies within time intervals and at various temporal resolutions. Our model also deploys both fixed and learnable positional embeddings to preserve the sequential order of the input time series and capture relative positional information after deformation, respectively. These embeddings work in tandem within the proposed architecture.

We summarise the main contributions of this paper as follows:

(a) We propose DEFORMTIME, a novel MTS forecasting model, that better captures inter- and intra-variable dependencies at different temporal granularities. It comprises two DABs which facilitate learning from adaptively transformed input across variables (V-DAB) and time (T-DAB).

(b) We assess the predictive accuracy of MTS forecasting models on 3 established benchmarks as well as 3 new infectious disease prevalence tasks, each across 4 increasing horizons. We argue that the disease prevalence tasks support a more thorough evaluation because data sets include a substantial amount of exogenous variables, and encompass multiple years, 4 of which are used as distinct annual test periods.

(c) In our experiments, DEFORMTIME reduces the mean absolute error (MAE) by 7.2% on average compared to the most competitive forecasting method (which may be a different one for each evaluation). There is a 5% MAE reduction based on established benchmarks and 9.3% for the disease forecasting tasks. Overall, performance gains remain stable as the forecasting horizon increases.

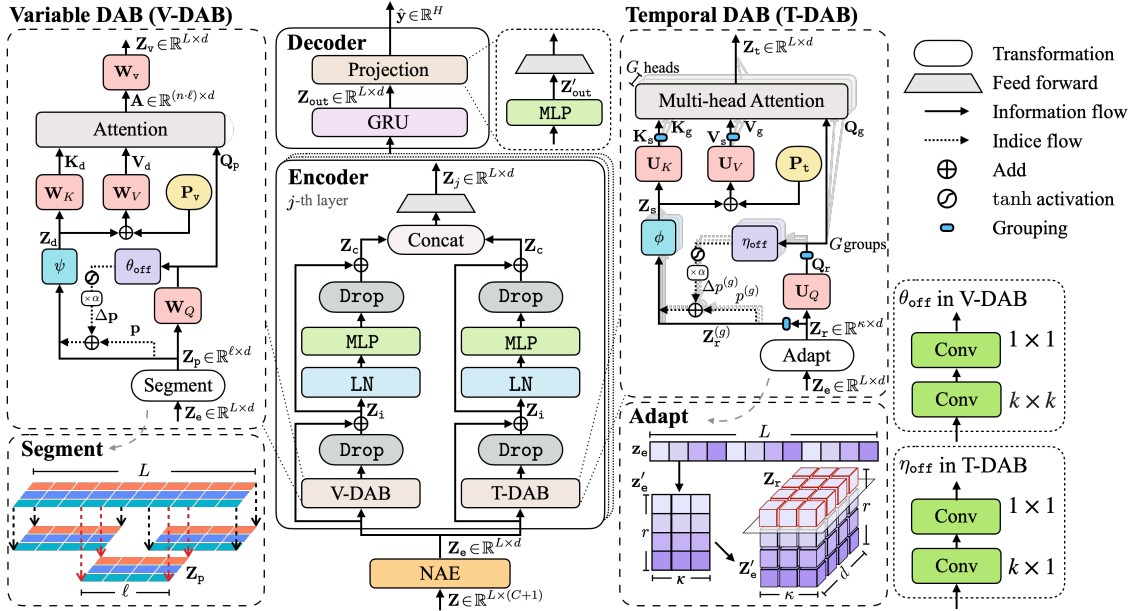

Figure 1: The architecture of DEFORMTIME. We use the notation introduced in sections 2 and 3. DEFORMTIME's core modules comprise two deformable attention blocks (DABs), a variable DAB (**V-DAB**) and a temporal DAB (**T-DAB**) that respectively capture inter- and intra-variable dependencies. Both DABs reside in the **Encoder** module. We deploy a 2-layer GRU as the **Decoder**. Finally, we have visualised key data operations (**Segment** and **Adapt** blocks) that take place in the DABs.

(d) As an aside, we note that models that attempt to establish some form of inter-variable dependency performed better in evidently (based on the results) the more challenging task of disease forecasting. This also highlights the need for more appropriate methods of assessment.

## 2 MTS forecasting task definition

Across our experiments, we consider an MTS forecasting task where the focus is on a single target variable, i.e. there might exist multiple inputs, and optionally multiple outputs, but we are only concerned with the predictions of one output variable. All models follow a rolling window forecasting setting with a look-back window of $L$ time steps. To be more precise, at time step $t$, $\mathbf{Q}_t \in \mathbb{R}^{L \times C}$ holds the time series of $C$ exogenous variables over the $L$ time steps $\{t-L+1, \ldots, t-1, t\}$. In addition, $\mathbf{y}_{t-\delta} \in \mathbb{R}^L$ holds the corresponding $L$ autoregressive historical values for the target variable for time steps $\{t-\delta-L+1, \ldots, t-\delta-1, t-\delta\}$. Note that $\delta \in \mathbb{N}_0$ introduces an optional delay of $\delta$ time steps between the observed exogenous variables and the endogenous (target) variable. This becomes relevant in a real-time infectious disease forecasting task, where estimates for the rate of an infectious disease are becoming available with a delay of $\delta = 7$ or $14$ days, when other indicators are not affected by this (see Appendix A.2). Joining both input signals, i.e. the exogenous and endogenous (autoregressive) variables, we obtain the input matrix $\mathbf{Z}_t = [\mathbf{Q}_t, \mathbf{y}_{t-\delta}] \in \mathbb{R}^{L \times (C+1)}$. Commonly, the prediction target is denoted by $\mathbf{y}_{t+H} \in \mathbb{R}^H$, where $H$ indicates the number of time steps we are forecasting ahead. However, some (baseline) models follow a multi-task learning formulation whereby every input variable becomes a prediction target. On this occasion, the prediction targets are denoted by $\mathbf{Y}_{t+H} = [\mathbf{Q}_{t+H}, \mathbf{y}_{t+H}] \in \mathbb{R}^{H \times (C+1)}$. Hence, the aim of the forecasting task is to learn a function $f(\cdot)$ such that $f : \mathbf{Z}_t \rightarrow \mathbf{y}_{t+H}$ or $\mathbf{Y}_{t+H}$. Irrespective of the number of outputs, forecasting accuracy is reported on the final estimate of $\mathbf{y}_{t+H}$, $y_{t+H} \in \mathbb{R}$, which denotes the value of the target variable at time step $t+H$. For notational simplicity, we have chosen to only imply temporal subscripts for the remainder of the manuscript (i.e. use $\mathbf{Z}$ over $\mathbf{Z}_t$).

## 3 Time series forecasting with DeformTime

An overview of DEFORMTIME's structure is presented in Figure 1. Motivated by our assumption that variables (endo- and exogenous) in an MTS task may not only be correlated with the historical values of the target variable, but also with other predictors within adjacent time steps, DEFORMTIME deploys a modified transformer encoder layer with two core modules that attempt to account for both dependencies: a Variable Deformable Attention Block (V-DAB) to capture inter-variable dependencies across time, and a Temporal Deformable Attention Block (T-DAB) to capture intra-variable dependencies across different time periods. In the following sections, we provide a detailed description of all modules and operations of DEFORMTIME.[1]

### 3.1 Multi-head attention

We first revisit the multi-head attention mechanism originally proposed by Vaswani et al. (2017). Given an input $\mathbf{Z} \in \mathbb{R}^{v \times u}$, $\mathbf{Q}, \mathbf{K}$, and $\mathbf{V} \in \mathbb{R}^{v \times u}$ respectively denote the query, key, and value embeddings projected from $\mathbf{Z}$ with learnable linear projection matrices $\mathbf{W}_Q, \mathbf{W}_K$, and $\mathbf{W}_V \in \mathbb{R}^{u \times u}$ (as in $\mathbf{Q} = \mathbf{Z}\mathbf{W}_Q$). Assuming $M$ attention heads, these embeddings are each partitioned into $M$ non-overlapping submatrices column-wise. For example, the $m$-th submatrix of $\mathbf{Q}$, denoted by $\mathbf{Q}_m$, has a dimensionality of $v \times (u/M)$. Following up on this notational convention, the output of the $m$-th attention head, $\mathbf{A}_m \in \mathbb{R}^{v \times (u/M)}$, is given by

$$\mathbf{A}_m = \texttt{softmax}\Big(\mathbf{Q}_m \mathbf{K}_m^\top / \sqrt{u/M}\Big) \mathbf{V}_m . \tag{1}$$

The outputs from each head are then concatenated along the second dimension and linearly projected into the hidden dimension $u$ with a weight matrix $\mathbf{W} \in \mathbb{R}^{u \times u}$ to form the output $\mathbf{A} \in \mathbb{R}^{v \times u}$. When we set $M = 1$, we refer to this method as single-head attention.

### 3.2 Neighbourhood-aware input embedding (NAE)

The receptive field of DEFORMTIME is determined by a CNN with sampling offset (see section 3.3). This introduces a limitation in capturing relationships between neighbouring variables. We seek to minimise the impact of this by learning a neighbourhood-aware input embedding (NAE) as follows. We first re-arrange the order of variables in the input $\mathbf{Z}$, ranking them based on their linear correlation with the target variable in a temporally aligned fashion (see also Appendix D.1). We then embed the correlation-driven re-arranged version of the input that captures $C + 1$ variables to a hidden dimension $d$. However, we do not perform this embedding as a single holistic step, but embed $G$ neighbouring groups of variables, each one to a shortened embedding of size $d/G$, using a fully connected layer.[2] Our expectation is that the presence of stronger intra-group correlations will be subsequently leveraged by our model. Both $d$ and $G$ are hyperparameters that we learn during training. The embeddings of each input variable grouping are concatenated to form $\mathbf{E} \in \mathbb{R}^{L \times d}$. Finally, we also add a sinusoidal position embedding (Vaswani et al., 2017; Zhou et al., 2021), $\mathbf{P}_\mathrm{n} \in \mathbb{R}^{L \times d}$, to $\mathbf{E}$, and normalise within layer as widely adopted in prior work (Dosovitskiy et al., 2021; Zhang & Yan, 2023). Hence, the embedding $\mathbf{Z}_\mathrm{e} \in \mathbb{R}^{L \times d}$ of the input $\mathbf{Z}$ is given by

$$\mathbf{Z}_\mathrm{e} = \texttt{LN}(\mathbf{E} + \mathbf{P}_\mathrm{n}) , \tag{2}$$

where $\texttt{LN}(\cdot)$ denotes layer normalisation (Ba et al., 2016).

### 3.3 Variable deformable attention block (V-DAB)

V-DAB aims to capture inter-variable dependencies across time. This is achieved by performing cross-attention over embeddings of both endo- and exogenous variables within proximal time steps. The embedded input sequence, $\mathbf{Z}_\mathrm{e}$, is first segmented into patches along the temporal dimension that encompasses a total of $L$ time steps. This reduces model complexity as we only consider dependencies within adjacent time steps. By deploying a segmentation length $\ell$ and a stride $s \leq \ell$ (to ensure all data points are included), we obtain

---

[1]The source code of DEFORMTIME is available at `github.com/ClaudiaShu/DeformTime`.
[2]The input is zero-padded to a proper length, if the number of variables is not divisible by $G$.

$n = \lfloor (L - \ell)/s \rfloor + 1$ patches, denoted by $\mathbf{Z}_{\mathsf{p}} \in \mathbb{R}^{\ell \times d}$.[3] We augment the vanilla Transformer attention (Vaswani et al., 2017) with a deformable mechanism which allows the network to adaptively attend across time and variables. Each data point in a patch $\mathbf{Z}_{\mathsf{p}}$ can be represented by a pair of (row, column) indices, $\mathbf{p} = (i, j)$. We sample from a deformed position $\mathbf{p} + \Delta\mathbf{p}$ over $\mathbf{Z}_{\mathsf{p}}$ to obtain the key and value embeddings, $\mathbf{K}_{\mathsf{d}}, \mathbf{V}_{\mathsf{d}} \in \mathbb{R}^{\ell \times d}$, respectively.[4]

To determine the position offset $\Delta\mathbf{p} \in \mathbb{R}^2$ we conduct the following series of operations. We first obtain a query embedding, $\mathbf{Q}_{\mathsf{p}} \in \mathbb{R}^{\ell \times d}$, over a patch of the input as described in section 3.1, i.e. $\mathbf{Q}_{\mathsf{p}} = \mathbf{Z}_{\mathsf{p}} \mathbf{W}_Q$. We then pass $\mathbf{Q}_{\mathsf{p}}$ from a 2D CNN, denoted by $\theta_{\mathsf{off}}$, comprising a convolutional layer with a $k \times k$ kernel that captures neighbouring information, and a $1 \times 1$ convolutional layer that projects the embedded feature. We use the tanh activation function and a learnable amplitude $\alpha \in \mathbb{R}_{>0}$ (that controls the upper bound of the sampling range) to obtain $\Delta\mathbf{p}$. Hence, $\Delta\mathbf{p}$ is given by

$$\Delta\mathbf{p} = \alpha \cdot \tanh\big(\theta_{\mathsf{off}}\big(\mathbf{Q}_{\mathsf{p}}\big)\big) \,. \tag{3}$$

We use $\Delta\mathbf{p}$ and bilinear interpolation ($\psi$) over a $2 \times 2$ grid determined by $\mathbf{p} + \Delta\mathbf{p}$ (Dai et al., 2017), to obtain the deformed patch $\mathbf{Z}_{\mathsf{d}}$:

$$\mathbf{Z}_{\mathsf{d}} = \psi\big(\mathbf{Z}_{\mathsf{p}}; \mathbf{p} + \Delta\mathbf{p}\big) \,.[5] \tag{4}$$

We then obtain $\mathbf{K}_{\mathsf{d}}, \mathbf{V}_{\mathsf{d}}$ from $\mathbf{Z}_{\mathsf{d}}$ with

$$\mathbf{K}_{\mathsf{d}} = \mathbf{Z}_{\mathsf{d}} \mathbf{W}_K \quad \text{and} \quad \mathbf{V}_{\mathsf{d}} = \mathbf{Z}_{\mathsf{d}} \mathbf{W}_V + \mathbf{P}_{\mathsf{v}} \,, \tag{5}$$

where $\mathbf{P}_{\mathsf{v}} \in \mathbb{R}^{\ell \times d}$ is a relative positional bias (Shaw et al., 2018) that is added to assist in maintaining some pairwise positional information after the deformation.

A single head self-attention layer is applied to attend $\mathbf{Q}_{\mathsf{p}}$ to $\mathbf{K}_{\mathsf{d}}, \mathbf{V}_{\mathsf{d}}$ given by

$$\mathbf{A}_i = \Big(\texttt{softmax}\big(\mathbf{Q}_{\mathsf{p}} \mathbf{K}_{\mathsf{d}}^\top / \sqrt{d}\big) \mathbf{V}_{\mathsf{d}}\Big) \mathbf{W}_i \,, \tag{6}$$

where $\mathbf{W}_i \in \mathbb{R}^{d \times d}$ is the weight matrix, to form the attention embedding $\mathbf{A}_i \in \mathbb{R}^{\ell \times d}$ for the $i$-th segmented patch. The output from all patches is concatenated across the temporal dimension, forming $\mathbf{A} \in \mathbb{R}^{(n \cdot \ell) \times d}$. We conduct cross-attention over the original latent variable captured in $\mathbf{Q}_{\mathsf{p}}$ and the deformed latent variables $\mathbf{K}_{\mathsf{d}}, \mathbf{V}_{\mathsf{d}}$. Specifically, elements at position $\mathbf{p}$ can attend to elements at position $\mathbf{p} + \Delta\mathbf{p}$. This enables the model to learn from interactions between neighbouring variables across time.

The final output of the V-DAB module, $\mathbf{Z}_{\mathsf{v}} \in \mathbb{R}^{L \times d}$, is given by

$$\mathbf{Z}_{\mathsf{v}} = \mathbf{W}_{\mathsf{v}}^\top \mathbf{A} \,, \tag{7}$$

where $\mathbf{W}_{\mathsf{v}} \in \mathbb{R}^{(n \cdot \ell) \times L}$ denotes the weight matrix of a fully connected layer.

### 3.4 Temporal deformable attention block (T-DAB)

We use the T-DAB module in parallel with V-DAB to capture intra-variable dependencies across different time periods. This is achieved by applying cross-attention to embeddings that hold information from the same input embedding over neighbouring time steps. The T-DAB module receives the same input as the V-DAB module ($\mathbf{Z}_{\mathsf{e}}$). Based on insights from prior work (Wu et al., 2023; Jia et al., 2023), we first adapt $\mathbf{Z}_{\mathsf{e}}$ to support learning from different temporal resolutions. Specifically, every column $\mathbf{z}_{\mathsf{e}} \in \mathbb{R}^L$ of $\mathbf{Z}_{\mathsf{e}}$ is transformed to $\mathbf{z}_{\mathsf{e}}' \in \mathbb{R}^{r \times \kappa}$, where $r$ denotes the time window (amount of time steps) that T-DAB is using for capturing temporal relationships, and $\kappa = L/r$.[6] This converts a vector representing $L$ time steps, $\{t_1, t_2, \ldots, t_L\}$, to an $r \times \kappa$ matrix where the starting time for each row is $\{t_1, t_2, \ldots, t_r\}$, and the time stamps for the elements of the $j$-th row are $\{t_j, t_{j+r}, t_{j+2r} \ldots, t_{j+(\kappa-1)r}\}$. Therefore, $\mathbf{Z}_{\mathsf{e}}$ becomes $\mathbf{Z}_{\mathsf{e}}' \in \mathbb{R}^{r \times \kappa \times d}$, which comprises $r$ patches; a patch for T-DAB is denoted by $\mathbf{Z}_{\mathsf{r}} \in \mathbb{R}^{\kappa \times d}$.

---

[3]$\mathbf{Z}_{\mathsf{e}}$ is zero-padded into proper length, if $L - \ell$ not divisible by $s$.

[4]See Appendix B for a detailed example of how deformation works in V-DAB.

[5]If a sampling position is out-of-bounds, we interpolate using zero values.

[6]$\mathbf{z}_{\mathsf{e}}'$ is padded to a proper length with the last available value, if $L$ is not divisible by $r$.

The factor $r$ (time steps) can be set to different values across encoder layers to capture dependencies at various temporal granularities (see Appendix D). If we set $r = 1$, $\mathbf{Z_e}$ remains unchanged. Similarly to V-DAB, we obtain the query embedding $\mathbf{Q_r} \in \mathbb{R}^{\kappa \times d}$ over the transformed input using $\mathbf{Q_r} = \mathbf{Z_r U}_Q$, where $\mathbf{U}_Q \in \mathbb{R}^{d \times d}$ is a weight matrix. We then implement an augmented Transformer attention that contains deformed information across the temporal dimension. We work our way column-wise, and for every column $\mathbf{z_r} \in \mathbf{Z_r}$, we sample (linearly as opposed to the bilinear approach in V-DAB) from an index position $p + \Delta p \in \mathbb{R}$.

The position offset ($\Delta p$) is shared among groups of correlated embedded sequences as they may have similar temporal dependencies. Given that the original input variables were already grouped based on their correlation with the target signal (section 3.2), we expect that correlations are already captured in feature neighbourhoods of $\mathbf{Z_r}$. Therefore, to obtain $\Delta p$, we first divide $\mathbf{Q_r}$ into the same $G$ subgroups (column-wise) as in the NAE operation. Each subgroup, $\mathbf{Q}_g \in \mathbb{R}^{\kappa \times (d/G)}$ with $g = \{1, \ldots, G\}$, shares the same temporal deformation controlled by $\Delta p^{(g)}$. Similarly to Eq. 3, $\Delta p^{(g)}$ is defined as

$$\Delta p^{(g)} = \alpha \cdot \tanh(\eta_{\mathtt{off}}(\mathbf{Q}_g)) \, , \tag{8}$$

with $\eta_{\mathtt{off}}$ denoting a 1D CNN (with a $k \times 1$ and $1 \times 1$ convolution layers), and $\alpha$ being the same offset amplitude as in V-DAB. We then obtain the matrix of temporally deformed sequences $\mathbf{Z_s} = \left[ \mathbf{Z_s}^{(1)}, \ldots, \mathbf{Z_s}^{(G)} \right] \in \mathbb{R}^{\kappa \times d}$; each submatrix, $\mathbf{Z_s}^{(g)} \in \mathbb{R}^{\kappa \times (d/G)}$, is given by

$$\mathbf{Z_s}^{(g)} = \phi\left( \mathbf{z_r} \in \mathbf{Z_r}^{(g)}; p + \Delta p^{(g)} \right) \, , \tag{9}$$

where $\phi(\cdot)$ denotes a linear interpolation function over 2 adjacent points of $\mathbf{z_r} \in \mathbf{Z_r}^{(g)}$ determined by the deformed index $p + \Delta p^{(g)}$. If a sampling position is outside of the matrix boundaries, we interpolate with 0.

Similarly to V-DAB, $\mathbf{K_s}, \mathbf{V_s} \in \mathbb{R}^{\kappa \times d}$ are obtained from $\mathbf{Z_s}$, using $\mathbf{K_s} = \mathbf{Z_s U}_K$, and $\mathbf{V_s} = \mathbf{Z_s U}_V + \mathbf{P_t}$, where $\mathbf{P_t} \in \mathbb{R}^{\kappa \times d}$ is a relative positional bias. $\mathbf{K_s}$ and $\mathbf{V_s}$ thus contain elements sampled from the neighbouring time steps ($z_{i,j-\alpha}, z_{i,j+\alpha}$). We then apply multi-head attention to have elements at the position $p$ attend to elements at the deformed position $p + \Delta p^{(g)}$. This is achieved by first splitting $\mathbf{K_s}$ and $\mathbf{V_s}$ into $G$ groups (submatrices) denoted by $\mathbf{K}_g$ and $\mathbf{V}_g \in \mathbb{R}^{\kappa \times (d/G)}$, and then by using $\mathbf{A}_g = \mathtt{softmax}\left( \mathbf{Q}_g \mathbf{K}_g^\top / \sqrt{d/G} \right) \mathbf{V}_g$. The outputs from different heads are concatenated column-wise and linearly projected with a weight matrix $\mathbf{W}_i \in \mathbb{R}^{d \times d}$ to obtain the attention embedding of the $i$-th patch $\mathbf{A}_i \in \mathbb{R}^{\kappa \times d}$. To form the output of T-DAB, denoted by $\mathbf{Z_t} \in \mathbb{R}^{L \times d}$, the outputs of all patches are concatenated along the first dimension and re-arranged back to the original temporal structure (matching the input $\mathbf{Z_e}$ as well as V-DAB's output $\mathbf{Z_v}$).

## 3.5 Encoder

The encoder of DEFORMTIME may consist of more than one layer. In our experiments, we deploy 2 encoder layers, similarly to related work (Liu et al., 2024; Wu et al., 2023). Each encoder layer contains two DAB branches (on the left and right side, respectively – see also Figure 1), comprising stacked transformer blocks (Dosovitskiy et al., 2021) with layer normalisation placed within the residual connection (Xiong et al., 2020; Xia et al., 2022). Instead of using vanilla attention, we are introducing a V-DAB (left side) and a T-DAB (right side) module to capture inter- and intra-variable dependencies, respectively. The operations of a branch can be summarised by

$$\mathbf{Z_i} = \mathtt{Drop}(\mathtt{DAB}\,(\mathbf{Z_e})) + \mathbf{Z_e} \quad \text{and} \tag{10}$$

$$\mathbf{Z_c} = \mathtt{Drop}(\mathtt{MLP}(\mathtt{LN}(\mathbf{Z_i}))) + \mathbf{Z_e} \, , \tag{11}$$

where $\mathtt{DAB}(\cdot)$ denotes the V-DAB or T-DAB operation, $\mathbf{Z_i}, \mathbf{Z_c} \in \mathbb{R}^{L \times d}$ are an intermediate output and the output of the DAB encoder layer, respectively, $\mathtt{Drop}(\cdot)$ denotes stochastic depth (Huang et al., 2016) where layers within the network are randomly dropped during training with a learnable probability, and $\mathtt{MLP}(\cdot)$ is a 2-layer perceptron with $\mathtt{ReLU}$ activation and a hidden size of $d$. The outputs from the two encoder branches are concatenated over the second dimension. We feed the concatenated output into a fully connected layer to form the output $\mathbf{Z}_j \in \mathbb{R}^{L \times d}$ of the $j$-th encoder layer.

### 3.6 Encoder-decoder structure

In T-DAB (section 3.4), we set the time window $r$ to different values across encoder layers, enabling the model to attend to information at multiple temporal granularities. To effectively learn from this encoding, we construct a hierarchical encoder structure motivated by prior related work (Zhou et al., 2021; Liu et al., 2022a; Zhang & Yan, 2023). Specifically, the output from encoder layer $j-1$, $\mathbf{Z}_{j-1} \in \mathbb{R}^{L \times d}$, becomes the input to encoder layer $j$. This is formulated by

$$\mathbf{Z}_j = \begin{cases} \texttt{Enc}(\mathbf{Z}_{\mathsf{c}}) & \text{if } j=1 \\ \texttt{Enc}(\mathbf{Z}_{j-1}) & \text{otherwise} \end{cases}, \tag{12}$$

where $\texttt{Enc}(\cdot)$ denotes the operations of an encoder layer with deformable attention (section 3.5). The output of the final encoder layer is fed to a 2-layer Gated Recurrent Unit (GRU) NN with a hidden dimension of $d$ that acts as the decoder. The output of the GRU decoder, $\mathbf{Z}_{\texttt{out}}$, maintains the same dimensionality as the input, i.e. $L \times d$. Finally, we use a 2-layer perceptron with a hidden dimension of $d$ and $\texttt{LeakyReLU}$ activation to project $\mathbf{Z}_{\texttt{out}}$ along the temporal dimension, forming an intermediate output $\mathbf{Z}'_{\texttt{out}} \in \mathbb{R}^{H \times d}$, where $H$ denotes the number of steps we are forecasting ahead as defined in section 2. We then feed $\mathbf{Z}'_{\texttt{out}}$ to a fully connected layer to linearly project it into the target forecasting output $\hat{\mathbf{y}} \in \mathbb{R}^H$.

## 4 Results

We present forecasting accuracy results across 3 benchmark and 3 disease rate prediction tasks, comparing DEFORMTIME to 9 competitive baseline models. In addition, we provide an ablation study and a computational complexity / efficiency analysis.

### 4.1 Experimental settings

**Forecasting tasks and baseline methods.** Experiments are conducted on 6 real-world data sets. These include 3 established benchmarks from previously published papers (Wu et al., 2021; Zeng et al., 2023; Yi et al., 2023b; Wu et al., 2023; Nie et al., 2023; Luo & Wang, 2024b), and specifically a weather-related and 2 electricity transformer temperature (ETTh1 and ETTh2) forecasting tasks with respective temporal resolutions of 10 minutes and 1 hour. In addition, we have formed 3 disease prevalence modelling tasks, focusing on the prediction of influenza-like illness (ILI) rates in England (ILI-ENG) and US Health & Human Services (HHS) Regions 2 and 9 (ILI-US2 and ILI-US9). For the ILI rate forecasting tasks, we also introduce the frequency time series of web searches as exogenous predictors. In the ETT and weather tasks, we consider oil temperature and carbon dioxide level, respectively, as the target variables; we refer to the rest indicators as exogenous variables. We note that the ILI forecasting tasks have considerably more exogenous variables compared to the ETTh1, ETTh2, and weather tasks. Based on the evaluation settings in prior work (Nie et al., 2023; Liu et al., 2024), for the ETT and weather tasks, we set the forecasting horizon $H$ to 96, 192, 336, and 720 time steps ahead, and use a single test fold of consecutive unseen instances. For the ILI forecasting task, we conduct experiments on 4 consecutive influenza seasons (2015/16 to 2018/19) as separate test sets (4 test folds); the forecasting horizons we consider are $H = \{7, 14, 21, 28\}$ days ahead. Further details, including the evaluation setup are available in Appendix A. While we are aware of other datasets used in MTS forecasting, not all of them are suitable because they may not have an explicitly defined target variable or pursue an ill-defined prediction task (see Appendix A.4 for further justification).

We compare DEFORMTIME to 9 competitive time series forecasting models that to the best of our knowledge, form the current SOTA methods. These are LightTS (Zhang et al., 2022), DLinear (Zeng et al., 2023), Crossformer (Zhang & Yan, 2023), PatchTST (Nie et al., 2023), iTransformer (Liu et al., 2024), TimeMixer (Wang et al., 2024a), ModernTCN (Luo & Wang, 2024b), CycleNet (Lin et al., 2024), and TimeXer (Wang et al., 2024b). We also include a naïve persistence model baseline. Further details and justification for the selection of baselines can be found in Appendix C. For the ILI forecasting tasks, we conduct hyperparameter tuning for all models. In the other benchmark tasks, with the exception of Crossformer and LightTS, we adopt the settings from their official repositories to reproduce results.

Table 1: Forecasting accuracy results across all tasks and methods. $H$ denotes the forecasting horizon time steps. For the ILI forecasting tasks, the table enumerates the average error across the 4 test seasons. Complete results per season are shown in Appendix E. $\epsilon$ % denotes sMAPE. The best results are in **bold** font and the second best are underlined.

| Models | | DEFORMTIME | | ModernTCN | | CycleNet | | TimeXer | | PatchTST | | iTransformer | | TimeMixer | | Crossformer | | LightTS | | DLinear | | Persistence | |
|---|---|---|---|---|---|---|---|---|---|---|---|---|---|---|---|---|---|---|---|---|---|---|---|
| | $H$ | MAE | $\epsilon$ % | MAE | $\epsilon$ % | MAE | $\epsilon$ % | MAE | $\epsilon$ % | MAE | $\epsilon$ % | MAE | $\epsilon$ % | MAE | $\epsilon$ % | MAE | $\epsilon$ % | MAE | $\epsilon$ % | MAE | $\epsilon$ % | MAE | $\epsilon$ % |
| ETTh1 | 96 | **0.1941** | **14.96** | 0.2047 | 15.66 | 0.1976 | 15.05 | 0.2135 | 16.03 | 0.2017 | 15.41 | 0.2052 | 15.46 | 0.2112 | 16.32 | 0.2126 | 16.52 | 0.2215 | 17.24 | 0.2599 | 20.82 | 0.2371 | 18.47 |
| | 192 | **0.2116** | **16.08** | 0.2417 | 18.32 | 0.2411 | 18.25 | 0.2322 | 16.96 | 0.2409 | 18.29 | 0.2429 | 18.13 | 0.2382 | 17.91 | 0.2820 | 21.63 | 0.2636 | 20.55 | 0.3798 | 31.78 | 0.2803 | 21.46 |
| | 336 | **0.2158** | **16.27** | 0.2415 | 18.52 | 0.2425 | 18.35 | 0.2414 | 17.34 | 0.2559 | 19.29 | 0.2593 | 19.11 | 0.2625 | 19.72 | 0.2947 | 22.65 | 0.2807 | 22.15 | 0.6328 | 58.34 | 0.3028 | 22.90 |
| | 720 | 0.2862 | 21.81 | 0.2785 | 20.44 | 0.3018 | 22.98 | **0.2617** | **18.58** | 0.3087 | 23.89 | 0.2886 | 22.05 | 0.3055 | 23.25 | 0.3350 | 24.84 | 0.5334 | 44.57 | 0.7563 | 69.52 | 0.3222 | 25.29 |
| ETTh2 | 96 | **0.3121** | 40.07 | 0.3199 | 40.68 | 0.3282 | 40.74 | 0.3346 | 41.04 | 0.3145 | **39.25** | 0.3420 | 42.41 | 0.3454 | 41.27 | 0.3486 | 40.71 | 0.3507 | 41.80 | 0.3349 | 41.68 | 0.3522 | 43.85 |
| | 192 | **0.3281** | **37.90** | 0.3887 | 47.08 | 0.3687 | 41.78 | 0.4154 | 47.07 | 0.3839 | 45.45 | 0.4233 | 47.44 | 0.4183 | 47.49 | 0.4035 | 43.16 | 0.4022 | 48.01 | 0.4084 | 50.67 | 0.4416 | 50.24 |
| | 336 | **0.3450** | **37.00** | 0.3904 | 50.54 | 0.3815 | 42.57 | 0.4041 | 42.26 | 0.4018 | 46.77 | 0.4332 | 45.95 | 0.4380 | 46.79 | 0.4487 | 49.44 | 0.4425 | 51.35 | 0.4710 | 55.53 | 0.4836 | 53.70 |
| | 720 | **0.3640** | **34.99** | 0.5728 | 63.04 | 0.4827 | 49.93 | 0.5135 | 56.17 | 0.4960 | 55.27 | 0.4565 | 45.40 | 0.4729 | 46.37 | 0.5832 | 61.45 | 0.6252 | 70.50 | 0.7981 | 94.67 | 0.5199 | 58.75 |
| Weather | 96 | **0.0244** | **37.89** | 0.0279 | 42.49 | 0.0253 | 38.87 | 0.0301 | 44.04 | 0.0258 | 39.37 | 0.0277 | 42.39 | 0.0322 | 45.90 | 0.0271 | 44.92 | 0.0293 | 48.48 | 0.0251 | 39.03 | 0.0329 | 51.83 |
| | 192 | **0.0260** | **39.33** | 0.0314 | 46.05 | 0.0270 | 41.72 | 0.0326 | 46.85 | 0.0279 | 42.02 | 0.0277 | 42.77 | 0.0347 | 48.62 | 0.0308 | 54.14 | 0.0319 | 51.45 | 0.0270 | 42.68 | 0.0361 | 54.92 |
| | 336 | **0.0291** | **44.26** | 0.0351 | 50.12 | 0.0301 | 45.31 | 0.0348 | 49.12 | 0.0303 | 45.31 | 0.0308 | 46.01 | 0.0359 | 49.75 | 0.0345 | 62.53 | 0.0317 | 50.83 | 0.0305 | 47.68 | 0.0361 | 55.14 |
| | 720 | 0.0363 | 53.72 | 0.0414 | 57.66 | 0.0383 | 56.47 | 0.0437 | 58.70 | 0.0389 | 56.04 | 0.0395 | 57.01 | 0.0457 | 59.82 | 0.0395 | 65.47 | 0.0386 | 62.96 | **0.0352** | 54.54 | 0.0394 | 56.04 |
| ILI-ENG | 7 | **1.6417** | 28.61 | 1.9489 | 28.28 | 2.5554 | 29.59 | 2.8084 | 33.66 | 2.3115 | 27.61 | 2.3084 | 26.38 | 2.1748 | 25.68 | 1.8698 | 25.71 | 2.2397 | 52.25 | 2.8214 | 43.02 | 2.1710 | **24.96** |
| | 14 | **2.2308** | 33.98 | 2.7050 | 36.01 | 3.3911 | 39.42 | 3.4937 | 41.88 | 3.2547 | 37.76 | 3.2301 | 36.67 | 3.0209 | 35.39 | 2.6543 | **30.97** | 2.6879 | 38.29 | 3.7922 | 55.28 | 3.0625 | 33.77 |
| | 21 | **2.6500** | **32.70** | 3.0400 | 40.02 | 4.4519 | 52.83 | 4.3337 | 51.56 | 4.3192 | 51.11 | 4.2347 | 48.93 | 3.5501 | 49.36 | 3.0014 | 50.74 | 3.3616 | 51.78 | 4.4739 | 61.25 | 3.8617 | 42.03 |
| | 28 | **2.7228** | 40.44 | 3.3611 | 47.87 | 5.0259 | 59.93 | 4.9013 | 61.60 | 4.9964 | 59.60 | 4.8125 | 55.35 | 4.0000 | 54.27 | 3.1983 | 46.14 | 3.4132 | 55.59 | 5.0347 | 67.75 | 4.5857 | 49.49 |
| ILI-US2 | 7 | **0.4122** | **16.01** | 0.4398 | 16.55 | 0.6951 | 24.67 | 0.6083 | 23.38 | 0.7097 | 24.52 | 0.6507 | 23.24 | 0.5284 | 20.07 | 0.4400 | 16.46 | 0.4632 | 16.74 | 0.7355 | 27.94 | 0.6474 | 22.48 |
| | 14 | **0.4752** | **17.73** | 0.5279 | 20.22 | 0.8219 | 29.15 | 0.7725 | 29.07 | 0.8635 | 30.11 | 0.7896 | 28.17 | 0.6556 | 24.61 | 0.5852 | 20.98 | 0.5827 | 23.11 | 0.8435 | 32.22 | 0.8135 | 28.24 |
| | 21 | **0.5425** | 22.13 | 0.5781 | 23.85 | 1.0469 | 37.98 | 0.8243 | 31.46 | 1.0286 | 36.70 | 0.8042 | 30.03 | 0.6794 | 27.68 | 0.6245 | 22.29 | 0.6683 | 29.27 | 0.9124 | 34.93 | 0.9635 | 33.51 |
| | 28 | **0.5538** | 22.25 | 0.5710 | 23.66 | 1.1388 | 42.31 | 0.9074 | 34.72 | 1.1525 | 42.61 | 0.9619 | 36.75 | 0.8853 | 36.53 | 0.6512 | **23.91** | 0.7175 | 27.73 | 0.9805 | 37.62 | 1.1007 | 38.54 |
| ILI-US9 | 7 | **0.2622** | 12.26 | 0.2899 | 14.17 | 0.4480 | 21.05 | 0.3813 | 18.20 | 0.4116 | 19.34 | 0.4057 | 18.57 | 0.3239 | 15.21 | 0.3149 | 14.44 | 0.3185 | 15.65 | 0.4675 | 23.47 | 0.4057 | 18.49 |
| | 14 | **0.3084** | 13.80 | 0.3417 | 15.29 | 0.5072 | 24.02 | 0.4665 | 22.14 | 0.5020 | 24.09 | 0.4702 | 22.44 | 0.4060 | 19.08 | 0.3571 | 17.23 | 0.3791 | 19.04 | 0.5467 | 27.35 | 0.5008 | 23.07 |
| | 21 | **0.3179** | 14.24 | 0.3710 | 15.43 | 0.5926 | 28.47 | 0.5715 | 27.42 | 0.5935 | 29.40 | 0.5106 | 24.11 | 0.4576 | 21.40 | 0.3418 | 15.90 | 0.4754 | 23.74 | 0.6001 | 29.66 | 0.5906 | 27.41 |
| | 28 | **0.3532** | **15.74** | 0.3940 | 17.19 | 0.7031 | 34.50 | 0.6555 | 31.32 | 0.6665 | 33.35 | 0.6498 | 31.04 | 0.5124 | 24.11 | 0.3747 | 16.44 | 0.4769 | 23.22 | 0.6564 | 32.16 | 0.6799 | 31.67 |

Note that due to different task formulations, DEFORMTIME and TimeXer focus on forecasting the endogenous variables only, whereas the other baseline models also predict future values of the variables that we consider as exogenous. Notably, we only use the last value of the predicted target variable $y_{t+H}$ (see section 2) for model evaluation, regardless of the number of outputs each model makes (see also Appendix D).

**DeformTime's setup.** We use $G=4$ variable neighbourhoods in all our experiments (see section 3.2). This setting ($G=4$) also determines the number of group partitions for offset generation and multi-head attention in T-DAB (section 3.4). We provide a more detailed analysis of the effect of this specific parameter in Appendix E.4. The learnable amplitude $\alpha \in \{3, 5, 7, 9\}$ (see sections 3.3 and 3.4) is tuned collectively for both V-DAB and T-DAB. We keep the kernel size $k$ of the interpolation functions $\phi(\cdot)$ and $\psi(\cdot)$ identical to $\alpha$ throughout the experiments. In all experiments, DEFORMTIME has 2 encoder layers (see section 3.5). For the ETT and weather tasks, we set $r=1$ in the first encoder layer and select from $r \in \{6, 12, 24\}$ time steps in the second encoder layer (see section 3.4). In addition, we select the segmentation length from $\ell \in \{6, 12, 24\}$ time steps (see section 3.3) collectively for both encoder layers. Given the more direct weekly temporal resolution in the reported ILI rates and the fact that web search activity trends do have weekly patterns, in the ILI forecasting task, we set $\ell=7$ time steps (days), and also set $r=1$ and $r=7$ time steps in the first and second encoder layer, respectively. The segmentation stride $s$ is kept the same as $\ell$ throughout the experiments.

**Hyperparameters specific to forecasting tasks and optimisation settings.** The look-back window $L$ is set to 336 time steps for the ETTh1, ETTh2, and weather forecasting tasks, based on Nie et al. (2023). For influenza forecasting, we set $L$ to $\{28, 28, 56, 56\}$ days for forecasting horizons $H = \{7, 14, 21, 28\}$ days ahead, respectively. Only for influenza forecasting, we also set the batch size and $d$ to 64. For the other tasks, these become learnable parameters. The number of training epochs is set to a maximum of 100 for ETTh1, ETTh2, and weather tasks and 50 for the influenza forecasting task. Neural networks are optimised with Adam using mean squared error (MSE) loss on all outputs. As an exception, we use MAE loss in all models for the weather forecasting task as this gives better performance on the tuned baselines. Note that the output can be a time series (sequence) of one variable (the target) or more variables (target and exogenous) depending on the forecasting method.

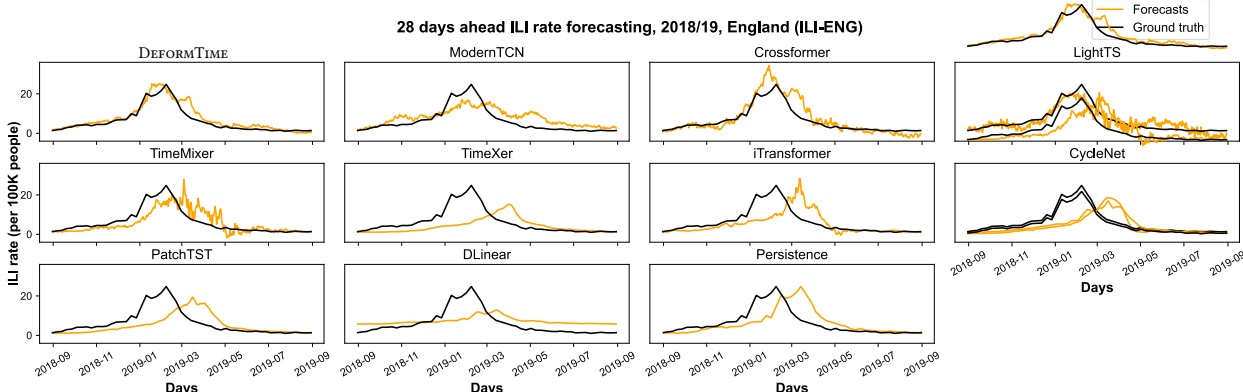

Figure 2: 28 days ahead forecasting results for influenza season 2018/19 in England (ILI-ENG) for all models. The black line denotes the ground truth, i.e. the reported (actual) ILI rates.

For the ETT and weather forecasting tasks, the batch size and the hidden dimension $d$ are both selected from $\{16, 32, 64\}$. For all tasks, Adam's initial learning rate is selected from $\{2, 1, .5, .2, .1, .05\} \times 10^{-3}$. For the ILI forecasting task, the learning rate is decayed to 0 with a linear decay scheduler applied after every epoch, whereas for the ETT and weather forecasting tasks, it is decayed by half every epoch following prior work (Zhou et al., 2021; Nie et al., 2023; Luo & Wang, 2024b). Depending on the model, the dropout (most baselines) or the layer drop rate (DEFORMTIME) is selected from $\{0, .1, .2\}$. Deciding the number of exogenous predictors in the ILI forecasting task (Lampos et al., 2017; Morris et al., 2023) introduces another group of hyperparameters that require validation (see section D.2). Hyperparameter values are determined using grid search. Training stops early when the validation error of the target variable over time $\{t+1, \cdots, t+H\}$ does not decrease further (compared to its lowest value) for 5 consecutive epochs. We then use the model with the lowest validation loss. Please refer to section D.3 for the random seed initialisation settings.

## 4.2 Forecasting accuracy

Forecasting accuracy for all tasks, models, and horizons is enumerated in Table 1. Mean Absolute Error (MAE) and symmetric Mean Absolute Percentage of Error (sMAPE[7] or $\epsilon$ %) are used as our evaluation metrics. Note that for the ILI rate prediction tasks, the error metrics represent the average MAE and sMAPE across 4 consecutive test seasons (per season results are available in Tables S1, S2, and S3, where we also include the linear correlation with the target variable as an additional performance metric), whereas for the remaining benchmark tasks performance metrics are obtained from one train / test split for each forecasting horizon, following previously reported evaluations.

DEFORMTIME displays an overall superior accuracy. It outperforms the most competitive method (that might be a different one per task and horizon), reducing MAE by 7.2% and sMAPE by 4.5% on average across all tasks and forecasting horizons. In more absolute terms, DEFORMTIME is the best performing model in all but two occasions based on MAE, and 20 out of 24 times based on sMAPE. If we exclude the weather task that uses a 1-year data set from a single location and hence is only expected to offer limited insights while attempting to predict 720 time steps (7.5 days) ahead (see also Appendix A.4), our method offers on average 9.3% of MAE reduction in the most challenging forecasting tasks (maximum forecasting horizon $H$). Specifically, MAE is reduced by 5.5% in the ETT tasks (for $H = 720$ hours), and by 14.9% (England) or 4.4% (US Regions) in the ILI tasks (for $H = 28$ days). Therefore, performance gains compared to other forecasting methods do not decrease as the forecasting horizon increases.

When comparing DEFORMTIME to a specific forecasting method (as opposed to the best performing one per task and horizon), we notice that MAE reduction is >11% on average across all tasks and forecasting horizons, ranging from a 11.9% reduction vs. ModernTCN to 34.8% vs. DLinear.[8] These significant performance gains

---

[7]sMAPE (Appendix E.1) can provide more balanced insights when metrics are averaged across different tasks.

[8]A brief note to further explain DLinear's performance is provided in Appendix E.7.

highlight our model's capacity to consistently perform well under different tasks and horizons compared to other SOTA models. Interestingly, DLinear fails to surpass the average accuracy of a naïve persistence model (DEFORMTIME reduces persistence's MAE by 30.1%).

Turning our focus to the more interpretable task of predicting ILI rates, we first notice that forecasting models that do not model inter-variable dependencies (DLinear, PatchTST, and CycleNet) or do so after embedding the input along the sequence dimension (iTransformer, TimeXer) showcase an overall inferior performance. This highlights the potential benefit of incorporating variable dependencies over time steps or slices that preserve the underlying temporal patterns. Such structure is a common design choice amongst the better-performing models (DEFORMTIME, ModernTCN, Crossformer, LightTS, and TimeMixer). Compared to DEFORMTIME's average sMAPE (across all locations) ranging from 18.96% to 26.14% for $H = 7$ to 28 days ahead forecasting respectively, the two consistent competitors are Crossformer (18.87% to 28.83%) and ModernTCN (19.67% to 29.57%). The rest of the methods do not perform well as the task becomes more challenging, reaching average sMAPEs ranging from 35.51% (LightTS) to 50.39% (TimeXer) for $H = 28$ days. Pending a more detailed evaluation (out-of-scope for this work), there is at least partial evidence to support that DEFORMTIME is a SOTA forecaster for ILI (Reich et al., 2019; Morris et al., 2021; Osthus & Moran, 2021; Morris et al., 2023). It not only demonstrates a great regression fit, but also captures the overall trend while forecasting 28 days ahead across 3 different geographical regions in 2 countries (average correlation is $> .90$, see Tables S1, S2, and S3).

We depict the ILI rate forecasts of all models in Appendix G. A snapshot is presented in Figure 2, showcasing 28 days ahead predictions for influenza season 2018/19 in England. We observe that many models resemble the naïve persistence model that predicts the last seen value of the target variable in the autoregressive input, providing smooth but uninformative and increasingly inaccurate (as $H$ increases) forecasts. DLinear provides the overall worst fit. With the exception of TimeXer and iTransformer (see also Appendix C), competitive baseline models that capture inter-variable dependencies provide more informative predictions. However, their estimates —at least in this example— appear to be more noisy, and are either not capable of capturing the onset of the influenza season (TimeMixer) or its intensity (ModernTCN, Crossformer and less so LightTS). Contrary to that, DEFORMTIME provides smoother and more accurate forecasts that corroborate our choice to account for both inter- and intra-variable dependencies.

Further experiments, presented in Appendix E.6, provide evidence that DEFORMTIME is also robust to random seed initialisation (Table S5). In Appendix E.8 and Table S6, we also provide results where the forecast error is averaged over the entire output sequence as opposed to only considering predictions at the target forecasting horizon time step, with additional discussion offering some insights as to why this may not be the most appropriate evaluation approach. Nevertheless, DEFORMTIME remains the best-performing forecasting model.

### 4.3 Ablation analysis

We perform an ablation study to understand the contribution of various operations in DEFORMTIME, namely the V-DAB and T-DAB modules, the position embeddings used in DABs (we conventionally denote both by $\mathbf{P_{v,t}}$), and the NAE module with and without position embedding $\mathbf{P_n}$. Experiments are conducted on the ILI rate forecasting task for 2 locations, England (ILI-ENG) and US Region 9 (ILI-US9), across all forecasting horizons ($H = \{7, 14, 21, 28\}$ days ahead) the 4 test seasons. Hyperparameters are re-tuned using the same validation approach separately for each ablation variant. Further details are provided in Appendix E.2.

Table 2 enumerates the ablation outcomes, showing the average MAE across all test periods. Evidently, each component or operation contributes to the reduction of MAE. On average, the V-DAB module provides stronger performance improvements (7.7%) compared to the T-DAB module (3.9%). However, for the longest forecasting horizon ($H = 28$), the difference between the level of contribution from these modules is lower (respectively, 9.8% and 8.0%). Hence, establishing inter-variable dependencies is always useful, but intra-variable dependencies appear to become more important in longer forecasting horizons.

The inclusion of the NAE component is equally important as it improves MAE by 6.4% on average. Interestingly, the level of relative improvement increases as the forecasting horizon extends (from 6.7% to 7.1%). This potentially highlights that longer forecasting horizons require more abstraction over the

Table 2: Ablation study for DEFORMTIME. We report the average MAE over the 4 test periods for ILI tasks across all forecasting horizons, $H = \{7, 14, 21, 28\}$ days ahead. '¬' denotes the absence of a module or an operation. $\mathbf{P_{v,t}}$ denotes both position embeddings used in DABs (V-DAB, T-DAB). $\mathbf{P_n}$ denotes the position embedding used in NAE.

| | $H$ | DEFORMTIME | ¬ V-DAB | ¬ T-DAB | ¬ $\mathbf{P_{v,t}}$ | ¬ NAE | ¬ $\mathbf{P_n}$ |
|---|---|---|---|---|---|---|---|
| ILI-ENG | 7 | **1.6417** | 1.8536 | 1.8043 | 1.9420 | 1.8238 | 1.8849 |
| | 14 | **2.2308** | 2.4978 | 2.3570 | 2.2646 | 2.4280 | 2.8982 |
| | 21 | **2.6500** | 2.9173 | 2.6840 | 2.8838 | 2.9523 | 2.9971 |
| | 28 | **2.7228** | 3.3685 | 3.2848 | 3.1082 | 3.1016 | 2.8446 |
| | Avg. | **2.3113** | 2.6593 | 2.5325 | 2.5497 | 2.5764 | 2.6562 |
| ILI-US2 | 7 | **0.4122** | 0.4600 | 0.4165 | 0.4507 | 0.4414 | 0.4612 |
| | 14 | **0.4752** | 0.4796 | 0.4758 | 0.5099 | 0.5033 | 0.4973 |
| | 21 | **0.5425** | 0.5790 | 0.5563 | 0.5696 | 0.5474 | 0.5545 |
| | 28 | **0.5538** | 0.5924 | 0.5794 | 0.5902 | 0.5960 | 0.5881 |
| | Avg. | **0.4959** | 0.5278 | 0.5070 | 0.5301 | 0.5220 | 0.5253 |
| ILI-US9 | 7 | **0.2622** | 0.2709 | 0.2649 | 0.2753 | 0.2665 | 0.2737 |
| | 14 | **0.3084** | 0.3138 | 0.3103 | 0.3174 | 0.3161 | 0.3270 |
| | 21 | **0.3179** | 0.3493 | 0.3236 | 0.3452 | 0.3273 | 0.3355 |
| | 28 | **0.3532** | 0.3664 | 0.3623 | 0.3689 | 0.3638 | 0.3693 |
| | Avg. | **0.3104** | 0.3251 | 0.3153 | 0.3267 | 0.3184 | 0.3264 |

feature space, favouring learning from correlated groups of variables. Finally, the use of position embeddings, albeit not entailing a very sophisticated operation, significantly enhances the impact of the V-DAB/T-DAB and NAE modules by 6.4% and 7.9%, respectively. We argue that position embeddings, obtained before ($\mathbf{P_n}$) and after ($\mathbf{P_{v,t}}$) deformation, work in tandem to maintain key information that evidently further improves the overall forecasting accuracy.

Interestingly, comparing the results in Tables 2 and 1, DEFORMTIME without the V-DAB module, which results in a simpler model that captures inter-variable dependencies via the NAE module and a GRU decoder, yields similar accuracy as ModernTCN. We consider that the added value of V-DAB stems from its ability to flexibly capture variable dependencies that may not be temporally aligned. These dependencies, if they exist, have been shown to potentially contribute to better forecasting performance when aligned (Zhao & Shen, 2024). ModernTCN, however, does not specifically model such unaligned temporal dependencies. Our analysis suggests that this omission is an important factor for the observed performance degradation.

## 4.4 Assessing the impact of an increasing number of exogenous predictors

We also assess the accuracy of models as the number of exogenous predictors increases. For this we use the ILI rate forecasting task given its relatively large amount of exogenous predictors. We set the number of exogenous input variables, i.e. the frequency time series for different search queries, to $C$ to $\{32, 64, 128, 256, 512\}$.[9] Queries are selected by obtaining the top-$C$ most correlated ones with the target variable (ILI rate) in the 5 last flu seasons in the training data (a detailed data set description is provided in Appendix A.2). We hypothesise that models that consistently give better performance as $C$ increases are more robust to handling multi-variable input and capture inter-variable dependencies better. To this end, we focus our analysis on models that have components that aim to capture information across variables, namely iTransformer, TimeXer, TimeMixer, LightTS, Crossformer, ModernTCN, and DEFORMTIME. Experiments are conducted on the ILI forecasting task for England across all 4 test seasons and we report the average MAE and sMAPE scores. All hyperparameters, except the number of input variables (see Appendix D.2), are re-tuned for each model (for all train/test splits).

---

[9]The total number of input variables is $C+1$ if we also factor in the autoregressive part of the input.

Table 3: Forecasting accuracy for models that capture inter-variable dependencies on the ILI-ENG task under increasing amount of input (exogenous) variables. The best results are in **bold** font and the second best are underlined.

| Models | iTransformer | | TimeXer | | TimeMixer | | LightTS | | Crossformer | | ModernTCN | | DEFORMTIME | |
|---|---|---|---|---|---|---|---|---|---|---|---|---|---|---|
| $H$ $C$ | MAE | $\epsilon$ % | MAE | $\epsilon$ % | MAE | $\epsilon$ % | MAE | $\epsilon$ % | MAE | $\epsilon$ % | MAE | $\epsilon$ % | MAE | $\epsilon$ % |
| 7   32 | 2.5141 | 28.67 | 2.7315 | 32.62 | 2.1333 | 24.68 | 1.9263 | 33.12 | **1.6659** | 27.51 | 1.9468 | 37.37 | 1.6790 | **23.56** |
|   64 | 2.3658 | 27.79 | 2.9042 | 33.52 | 2.1768 | 25.03 | 2.2359 | 41.94 | 1.9090 | 30.72 | 1.7984 | 41.54 | **1.4904** | **23.23** |
|   128 | 2.3346 | 27.15 | 2.8816 | 33.29 | 2.4315 | 27.96 | 2.1155 | 46.60 | 1.6160 | 24.80 | 1.8501 | 31.70 | **1.4468** | **22.68** |
|   256 | 2.3325 | 27.06 | 2.7711 | 33.60 | 2.4842 | 28.34 | 2.5989 | 35.87 | 1.8018 | **22.17** | 1.9449 | 33.33 | **1.5797** | 25.03 |
|   512 | 2.4466 | 27.98 | 2.9864 | 35.19 | 2.8672 | 32.94 | 2.7775 | 51.29 | 1.7930 | **22.53** | 2.1583 | 46.45 | **1.6314** | 23.65 |
| 14   32 | 3.4327 | 39.69 | 3.4781 | 40.86 | 3.1351 | 35.30 | 2.8046 | 38.21 | 2.5710 | 34.28 | 2.6098 | 41.93 | **2.1651** | **30.25** |
|   64 | 3.2482 | 37.76 | 3.5437 | 41.27 | 3.2472 | 37.42 | 2.9960 | 48.12 | 2.5851 | 35.37 | 2.4487 | 34.03 | **2.0055** | **29.36** |
|   128 | 3.4923 | 39.94 | 3.5598 | 41.56 | 3.3225 | 37.71 | 2.9178 | 40.72 | 2.5510 | 32.87 | 2.4755 | 37.70 | **2.0362** | **29.55** |
|   256 | 3.3671 | 38.98 | 3.4928 | 41.95 | 3.3600 | 39.06 | 2.6980 | 47.30 | 3.1166 | 38.46 | 2.7400 | 38.99 | **2.1626** | **31.91** |
|   512 | 3.3427 | 38.53 | 3.6936 | 44.14 | 3.7326 | 41.78 | 2.7095 | 39.44 | 3.1111 | 39.70 | 2.9224 | 43.48 | **2.2848** | **33.15** |
| 21   32 | 4.6290 | 55.13 | 4.4526 | 54.44 | 4.0293 | 56.15 | 3.2743 | 45.51 | 2.8971 | **33.86** | 2.7548 | 41.25 | **2.7363** | 37.05 |
|   64 | 4.1034 | 48.80 | 4.3821 | 51.99 | 3.9984 | 54.02 | 3.4519 | 52.31 | 3.0058 | **36.95** | 2.7769 | 51.61 | **2.5821** | 37.15 |
|   128 | 4.2907 | 50.79 | 4.3504 | 51.15 | 3.7035 | 53.27 | 4.5949 | 60.69 | 2.8090 | **36.49** | 2.9763 | 40.86 | **2.6646** | 37.65 |
|   256 | 4.4976 | 53.14 | 4.5287 | 56.65 | 3.7519 | 52.49 | 3.5740 | 52.22 | 3.0846 | 38.88 | 2.8275 | 41.56 | **2.5746** | **34.64** |
|   512 | 4.2563 | 49.04 | 4.6257 | 56.27 | 3.5404 | 45.61 | 3.3258 | 54.37 | 3.6267 | 44.76 | 2.9788 | 49.13 | **2.6973** | **35.55** |
| 28   32 | 4.8528 | 58.29 | 4.9145 | 61.07 | 4.8393 | 60.72 | 3.7597 | 46.94 | 3.5441 | 44.01 | 3.1705 | 41.17 | **2.8000** | **34.75** |
|   64 | 4.6811 | 56.08 | 4.8851 | 61.35 | 4.5416 | 58.54 | 3.6672 | 57.08 | 3.6433 | 46.64 | 3.3598 | 49.85 | **2.7637** | **34.17** |
|   128 | 4.8265 | 56.90 | 4.8329 | 60.65 | 4.7736 | 57.94 | 3.7655 | 56.16 | 3.4786 | 44.29 | 3.0837 | 47.69 | **2.7945** | **36.90** |
|   256 | 5.0621 | 60.88 | 4.9689 | 62.48 | 4.4152 | 57.56 | 3.6055 | 53.02 | 3.6987 | 45.14 | 3.1944 | 46.01 | **2.9963** | **41.02** |
|   512 | 5.0417 | 59.91 | 5.3472 | 68.10 | 4.0968 | 54.52 | 3.5609 | 46.86 | 4.0750 | 48.02 | 3.2858 | 47.23 | **3.1113** | **38.94** |

Forecasting accuracy results are enumerated in Table 3. First, considering the entirety of reported outcomes, we note that DEFORMTIME provides superior performance, reducing the average MAE and sMAPE by 11.8% and 10.6% respectively compared to the best-performing baseline(s). When we consider outcomes for a consistent setting of the number of input variables, DEFORMTIME yields the lowest MAE in all but one forecasting task (vs. Crossformer for $C = 32$ and $H = 7$ days ahead). This re-affirms our previously reported results and also highlights that DEFORMTIME tends to outperform competitive baselines irrespectively of the number of exogenous predictors, maintaining its superiority as the forecasting horizon increases.

Consistent with the results presented in section 4.2, the most competitive baselines are ModernTCN and Crossformer, where DEFORMTIME reduces their MAE by 13.9% and 17.1% on average, respectively. iTransformer and TimeXer are the two least competitive models, with DEFORMTIME reducing the average MAE by 37.9% and 41.9% respectively. This provides some evidence for the importance of structural design when it comes to time series modelling as highlighted in section 4. Specifically, disrupting the input's temporal patterns prior to modelling is likely to cause model collapse, resulting in learning a simplistic forecasting choice resembling the persistence model, as also reflected in Figure 2. It also underscores the need for conducting evaluation on diverse datasets and metrics to comprehensively assess model performance (TimeXer is the best-performing baseline for ETTh1 but less competitive for other tasks).

## 4.5   Computational complexity and efficiency of DeformTime

The computational complexity of DEFORMTIME is $\mathcal{O}(L^2 d + L d^2)$, where $L$ and $d$ respectively denote the size of the look-back window and the number of hidden layers throughout the method. A detailed derivation of this is provided in Appendix E.5. We note that the number of operations for an encoder layer reduces quadratically as we increase $r$ (the size of T-DAB's time window) to values greater than 1. Likewise, segmenting the $L$ input time steps to patches of a smaller length (in V-DAB) results in quadratic computational benefits.

Figure 3 depicts the GPU VRAM memory consumption of DEFORMTIME during training compared to other transformer-based models w.r.t. the length of the look-back window ($L$; part A) and the number of input variables ($C + 1$; part B). In these experiments, we set both the batch size and the hidden dimension $d$ to 64.

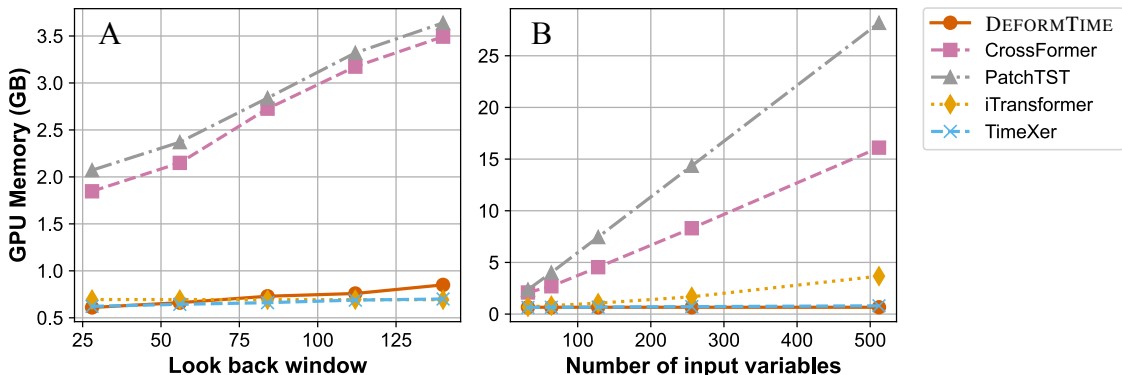

Figure 3: GPU memory (VRAM) consumption based on (**A**) the length (time steps) of the look-back window ($L$), and (**B**) the number of input variables ($C+1$).

While assessing the impact of $L$, we set $C+1$ to 32, and while assessing the impact of the amount of input variables, we set $L$ to 56. We then present the average memory consumption across 50 training epochs.

Overall, DEFORMTIME has a relatively small (GPU VRAM) memory footprint. As the length of the look-back window increases (from 28 to 140 time steps), iTransformer, TimeXer, and DEFORMTIME are showing a stable memory consumption that does not exceed 1 GB. We note that the low memory consumption of iTransformer / TimeXer is expected as they encode all input / exogenous input over the sequence dimension, which guarantees consistent GPU memory usage regardless of the value of $L$. In DEFORMTIME, the input partitioning and adaptive transformations within the V-DAB and T-DAB modules reduce computational complexity, leading to reduced memory consumption, especially when $L$ is large. At the same time, the memory footprints of PatchTST and Crossformer linearly increase with $L$.

As we now increase the number of input variables from (32 to 512), other methods display a linear increase in memory consumption (delayed and considerably lower for iTransformer and TimeXer compared to PatchTST and Crossformer), whereas for DEFORMTIME memory consumption is almost unaffected and stable (e.g. $< 1$ GB for DEFORMTIME, but $> 24$ GB for PatchTST). Hence, we argue that our method can handle a greater amount of exogenous predictors more efficiently.

Compared to other transformer-based MTS forecasting methods (Crossformer, PatchTST, and iTransformer), our method exhibits consistently low memory consumption as the number of input variables increases (vs. a linear increase for the other methods). Furthermore, increasing the look-back window does not overly affect memory consumption either (vs. a more accelerated linear increase for Crossformer and PatchTST). Hence, we deduce that DEFORMTIME is adept to MTS forecasting tasks with a considerable amount of exogenous predictors and longer look-back windows.

## 5 Conclusions

We propose DEFORMTIME, a novel deep learning architecture for multi-variable time series forecasting that uses deformable attention blocks to effectively learn from exogenous predictors, deploying specific operations that aim to capture inter- and intra-variable dependencies. We assess forecasting accuracy using 3 established benchmark tasks as well as 3 ILI rate prediction tasks with web search frequency time series as exogenous variables, covering 3 locations (in 2 countries) and spanning a time period longer than 12 years. DEFORMTIME yields strong forecasting accuracy across the board, reducing mean absolute error on average by 7.2% compared to the best baseline model (different for each task and horizon) and by at least 11.9% (for ModernTCN (Luo & Wang, 2024b)) when compared to a specific model. Importantly, performance gains remain stable for longer forecasting horizons. Our experiments, including the ablation study, highlight that modelling variable dependencies is an important attribute. Specifically in the disease forecasting task, where more exogenous predictors are present, the most competitive baselines capture variable dependencies to some extent, whereas models that do not on many occasions cannot surpass the performance of a naïve persistence

model (see also Appendix E.9). In contrast to other methods, DeformTime's GPU memory footprint is not significantly affected by the amount of exogenous variables.

## Acknowledgements

The authors would like to thank the RCGP for providing ILI rates for England. V. Lampos would like to acknowledge all levels of support from the EPSRC grant EP/X031276/1.

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

# Supplementary Material / Appendix

## A  Data sets for time series forecasting

This section offers a detailed description of the data sets used in our experiments to assess the performance of a series of methods for MTS forecasting. We provide links to data sets that are publicly available. For the ILI data sets formed by us, while we hope that we will be able to share the exact data set with the broader community in the future, we cannot do so due to copyright restrictions at the time of publication. As an alternative, we provide instructions as to how research teams can obtain access directly from the data provider.

### A.1  Established time series forecasting benchmarks

**ETTh1 and ETTh2.**  ETTh1 and ETTh2 are electricity transformer temperature data sets that were obtained from 2 different counties in China (Zhou et al., 2021).[10] Each data set contains 6 exogenous variables capturing power load attributes, and the target variable which is the temperature of oil. The data set covers a period from July 1, 2016 to June 26, 2018 (although not all data points are used in our experiments to match the setup in related methods that we compare to). The temporal resolution of these data sets is hourly. Adopting the setup in prior work (Wu et al., 2023; Nie et al., 2023; Liu et al., 2024), we use a total amount of 14,400 time steps (starting from July 1, 2016) where the first 8,640 are used for training, the next 2,880 for validation, and the last 2,880 for testing.

**Weather.**  The weather data set (Wu et al., 2021) contains meteorological measurements collected from a weather station at the Max Planck Biogeochemistry Institute. It covers a year from January 1, 2020 to January 1, 2021.[11] The temporal resolution is 10 minutes and 20 meteorological indicators (exogenous variables) are being reported; the target variable is carbon dioxide concentration level. This results in a total of 52,696 time steps (samples). Based on prior work (Zhang & Yan, 2023), we use 70%, 10%, and 20% of the time steps for training, validation, and testing, respectively.

### A.2  Forecasting influenza-like illness rates using web search activity

Important components of our data set are shared in the paper's online repository. At the time of publication, these allow replication of our work for teams who have obtained the same data-sharing agreements (mainly access to the Google Health Trends API).

**Influenza-like illness (ILI) rates.**  ILI is defined as the presence of common influenza symptoms (e.g. sore throat, cough, headache, runny nose) in conjunction with high fever. We obtain ILI rates for England (in the United Kingdom) and two US HHS Regions, specifically Region 2 (states of New Jersey and New York) and Region 9 (states of Arizona and California). For England, we obtain ILI rates from the Royal College of General Practitioners (RCGP) that monitors ILI prevalence via an established sentinel network of GP practices throughout the country. An RCGP ILI rate represents the amount of ILI infections in every 100,000 people in the population of England. For the US Regions, data is obtained from the Centers for Disease Control and Prevention (CDC).[12] An ILI rate from CDC represents the proportion of ILI-related doctor consultations over the total amount of consultations (for any health issue). Hence, the units of the ILI rate in these two monitoring systems (RCGP and CDC) are different. The time span for the obtained ILI rates is the same as the time span for the web search data (see next paragraphs).

---

[10]ETTh1 and ETTh2, `github.com/zhouhaoyi/ETDataset`
[11]Weather data set, `github.com/thuml/Autoformer`
[12]US ILI rates (CDC), `gis.cdc.gov/grasp/fluview/fluportaldashboard.html`

**Influenza season definition.** An annual influenza season for the HHS US Regions is assumed to start on August 1 of year $\chi$ and end on July 31 of year $\chi+1$. For England, this is shifted by a month, i.e. an annual influenza season starts on September 1 of year $\chi$ and ends on August 31 of year $\chi+1$.

**Linear interpolation of weekly ILI rates.** ILI rates are reported on a weekly basis. To generate daily data (that increases the amount of training samples by a factor of 7), we use linear interpolation. It should be noted that for RCGP, Monday determines the start of a week, while for CDC this is determined by Sunday (different ISO specification). We assume that the weekly reported ILI rate is representative of the middle day of a week (Thursday for England and Wednesday for the US Regions) and the ILI rates are then linearly interpolated accordingly.

**Real-time delay in ILI rate availability.** We also note that in practice, the reported ILI rate is delayed, i.e. assuming that it is published by RCGP or CDC at time $t$, it actually refers to an ILI rate representing a previous time $t-\delta$. Specifically, we consider that there is a delay of $\delta=7$ days for RCGP/England, and a delay of $\delta=14$ days for CDC/US Regions (Wagner et al., 2018; Reich et al., 2019). This delay, $\delta$, has an impact on the autoregressive time series (of the target variable, in this case, the ILI rate) in a forecasting task (see also section 2). As the delay increases, the forecaster is expected to rely more on the exogenous predictors (that are not delayed) rather than the autoregressive time series of the target variable.

**Web search frequency time series.** Web search activity trends, if modelled appropriately (Ginsberg et al., 2009; Lazer et al., 2014; Lampos et al., 2015; Kandula & Shaman, 2019), are a good indicator of influenza rates in a population and can become a strong exogenous predictor for influenza forecasting (Dugas et al., 2013; Morris et al., 2023). We obtain web search activity data from the Google Health Trends API; this is not a publicly available API, but access can be provided via an application process.[13] For a day and a certain location, the frequency of a search query is determined as the ratio of searches conducted for a particular term or set of terms divided by the overall search volume. We use a pool of 22,071 unique health-related search queries and obtain their daily frequency from August 1 (US regions) or September 1 (England), 2006 to July 31 (US regions) or August 31 (England), 2019. However, we note that not all search queries are used as exogenous variables in our forecasting models. A feature selection process is described in Appendix A.3.

Given that for the US, we can only obtain data at the state level (as opposed to regional), we use a weighted average of the state-level search query frequencies. The weights are based on the population of each state. In particular, for Region 2, the states of New Jersey and New York have weights of 0.32 and 0.68, whereas for Region 9, the states of Arizona and California have weights of 0.16 and 0.84, respectively. Note that smaller locations (or distant ones) that may be part of an HHS Region are excluded from the web search data sets; given their relatively small population, these locations do not have a significant impact on the reported ILI rates and the data obtained from the Google Health Trends API are quite sparse.

**Training, validation, and test sets.** We assess the accuracy of forecasting models across the last 4 influenza seasons (4 separate test sets) in England and the US regions. Each test period is a complete influenza season (as previously defined). For each test period, we train models based on the 9 influenza seasons that precede it. A part of each training set is used for validating model decisions, including hyperparameters. We construct validation sets using the following strategy. Each validation set has 180 days in total, using 60-day periods to capture the onset, peak, and the period after the peak (or the outset) of an influenza season. Each 60-day period comes from a different influenza season in the training set; we use the last 3 influenza seasons in the training data to make sure our validation process has a recency effect. The last, penultimate, and third from last influenza seasons are used to define the outset, peak, and onset validation periods, respectively. We use CDC's definition to determine the onset of an influenza season, i.e. a time point is deemed to be the onset when the subsequent ILI rates exceed a threshold for 2 consecutive weeks.[14] The peak point is simply the highest ILI rate within an influenza season. The outset has the inverse definition from the onset, i.e. the last time point where the ILI rate exceeds the onset threshold for two consecutive weeks. Once these

---

[13]Google Health Trends API, `support.google.com/trends/contact/trends_api`
[14]CDC's influenza season onset definition, `cdc.gov/fluview/overview`

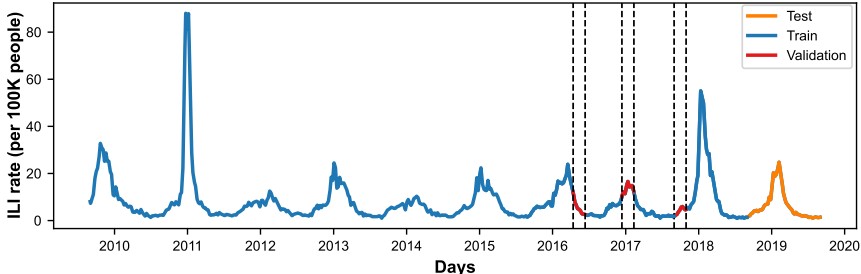

Figure S1: An example of how the training, validation, and test sets are constructed when the test influenza season is 2018/19 (England). The lines in blue, red, and orange colour denote the training, validation, and test periods, respectively. To form the validation set from our training data, we select the period after the peak (outset) from the third to last influenza season, the period around the peak from the penultimate season, and a period around influenza onset from the last season.

time points are determined, they become the 30th point in a 60-day validation period. Figure S1 shows an example of the validation periods that were determined when training an ILI rate model for England using (training) data from September 2009 to August 2018 (the test period being September 2018 to August 2019).

### A.3 Feature selection for web search activity time series

Originally, we consider 22,071 search queries. We then perform two feature selection steps to maintain more relevant queries to the ILI forecasting task. First, we apply a semantic filter to remove queries that are not related to the topic of influenza, similarly to the approach presented by Lampos et al. (2017). We obtain an embedding representation for each search query using a pre-trained sentence BERT model.[15] We also obtain the embeddings of influenza-related (base) terms and expressions, such as "flu", "flu symptoms" and so on.[16] We compute the cosine similarity between each base term and search query and maintain the top-1000 (England) or top-500 (US regions) search queries (per base term). This leaves 4,398 and 2,479 queries respectively for England and the US regions. The second selection step is dynamic, i.e. it might have a different outcome as the training data change. In this step, we compute the linear correlation between the remaining search queries and the target variable (ILI rate) in the 5 most recent influenza seasons in the training data (to impose recency). We maintain search queries with a correlation that exceeds a correlation threshold $\tau$, which is a learnable hyperparameter (see also Appendix D.2).

### A.4 Criticism of data sets used to benchmark forecasting methods

Although we present results using the ETTh1, ETTh2, and weather data sets to compare with other forecasting methods in the literature, we consider these data sets to have several shortcomings. Clearly, a weather data set that covers only a 1-year period and is based on the sensors from just one geographical location cannot be expected to provide good enough insights into meteorological forecasting. It captures a negligible portion of meteorological data and, as a result, derived forecasting models cannot have any practical impact (compared to actual models that are used in meteorology). Hence, using this data set for training and evaluating a forecasting model may, unfortunately, result in misleading or at least inconclusive outcomes. Similar issues, but perhaps to a smaller extent, are present in the ETTh1 and ETTh2 data sets. On the one hand, these data sets seem to have a more solid practical application. However, experiments on their ETTm1 and ETTm2 variants, where the temporal resolution changes from 1 hour to 15 minutes (same data, shorter temporal resolution), showcase part of the problem. When we compared the forecasting performance of SOTA baselines between the two different temporal resolutions, we surprisingly found that the prediction based on the hourly sampled data set has better accuracy compared to the quarter-hourly sampled data set even at longer forecasting horizons. For example, models that conduct 192 or even 336 time steps (equivalent

---

[15]Sentence BERT, `huggingface.co/bert-base-uncased`

[16]The complete list of base terms is provided at `github.com/ClaudiaShu/DeformTime`.

to 8 or 14 days) ahead forecasting with ETTh1 and ETTh2 achieve a lower MAE compared to a 720 time steps ahead forecast with ETTm1 and ETTm2 (equivalent to 7.5 days ahead forecast), respectively (Nie et al., 2023; Wu et al., 2023; Luo & Wang, 2024b).

We also noticed that some papers run experiments on US national weekly ILI rates obtained from the CDC (Zhou et al., 2022b; Wang et al., 2023; Nie et al., 2023). The first concerning observation was that in these experiments, the input variables were based on variates (columns) from the CDC extract (spreadsheet) that only provide redundant insights compared to the ILI rate (e.g. the raw numerator and denominator of an ILI rate or the population unweighted ILI rate). In addition, some forecasting horizons that were explored in these benchmarks were practically unreasonable (even from a biological perspective, there are limits in predicting the future, especially when it comes to viruses). SOTA forecasting models for influenza generally provide good accuracy at approximately a 2 weeks ahead forecasting horizon (Osthus & Moran, 2021; Morris et al., 2023). Anything beyond that with satisfactory performance should be considered as a very important development within the disease modelling community. We do think DEFORMTIME might fall in that category as it delivers good results for 3 or 4 weeks ahead forecasting horizons. Contrary to that, in the aforementioned ILI rate benchmark, there were forecasting horizons of 24 or 60 weeks ahead (i.e. even more than a year!). Of course, the accuracy of the forecasting models under these forecasting horizons was reportedly detrimentally poor (influenza does not have a strong periodicity), and consequently could not be used for any meaningful comparative conclusions. If all forecasting models are performing poorly, even the least poorly performing one still is an inadequate forecaster (for the underlying task).

Beyond this, we have also excluded data sets, such as traffic (Wu et al., 2021) and electricity consumption (Zhou et al., 2021), that do not have one explicitly defined target variable, but instead focus on multivariate predictions (multi-task learning). Consequently, these data sets are not entirely compatible with our forecasting task or method (DEFORMTIME makes predictions about one target variable).

To compensate for that we have developed a set of 3 influenza forecasting tasks. Models from these tasks can find direct real-world applications, i.e. become part of influenza monitoring systems in England or the US. The evaluation process is rigorous, i.e. across 4 consecutive influenza seasons as opposed to using 1 fixed test period that may again lead to biased insights.

## B   Detailed explanation in deformation using V-DAB

In this section, we provide a detailed description of how positional deformation is conducted in DEFORMTIME, taking V-DAB as an example. Suppose we have an element $\mathbf{Z}_\mathtt{p}(m, n)$ that was originally on position $\mathbf{p} = (m, n)$ of the patch matrix $\mathbf{Z}_\mathtt{p}$. In this example, let's assume that $m = 5$ and $n = 10$ and that the learned positional offset for that position is $(\Delta m, \Delta n) = (-0.1, 1.5)$. Then, the value for $\mathbf{Z}_\mathtt{d}(m, n)$ is the value sampled from $\mathbf{Z}_\mathtt{p}(m + \Delta m, n + \Delta n)$ or, in this example, $\mathbf{Z}_\mathtt{p}(5 - 0.1, 10 + 1.5)$ with bilinear interpolation. Specifically, the deformed point $\mathbf{p} = (4.9, 11.5)$ lies within a $2 \times 2$ index matrix of actual discrete positions, i.e. $(4, 11)$, $(4, 12)$, $(5, 11)$, and $(5, 12)$. We first linearly interpolate the values across the rows of the index matrix. We note that the deformed position in this example is equal to 11.5 and that means that there is an equal distance between the indices 11 and 12. We therefore perform:

$$\mathbf{Z}_\mathtt{p}(4, 11.5) = \mathbf{Z}_\mathtt{p}(4, 11) \times 0.5 + \mathbf{Z}_\mathtt{p}(4, 12) \times (1 - 0.5) \text{ and}$$
$$\mathbf{Z}_\mathtt{p}(5, 11.5) = \mathbf{Z}_\mathtt{p}(5, 11) \times 0.5 + \mathbf{Z}_\mathtt{p}(5, 12) \times (1 - 0.5) \ .$$

We then conduct linear interpolation along the column to obtain the final value of the bilinear interpolation. Here we note that the deformed position is equal to 4.9 and hence it does not have the same distance between indices 4 and 5. We therefore perform:

$$\mathbf{Z}_\mathtt{p}(4.9, 11.5) = \mathbf{Z}_\mathtt{p}(4, 11.5) \times 0.1 + \mathbf{Z}_\mathtt{p}(5, 11.5) \times (1 - 0.1) \ .$$

This operation is conducted on every $\mathbf{p} \in \mathbf{Z}_\mathtt{p}$ to obtain $\mathbf{Z}_\mathtt{d}$.

$\mathbf{Z}_\mathtt{d}$ then forms the key ($\mathbf{K}$) and value ($\mathbf{V}$) embeddings with learnable weight matrices. Given a scalar element $z_{(i,j)} \in \mathbf{Z}_\mathtt{p}$, $\mathbf{K}$ and $\mathbf{V}$ contain elements sampled from the neighbouring latent variables ($\mathbf{z}_{i-\alpha}$, $\mathbf{z}_{i+\alpha}$) in V-DAB or neighbouring time steps ($z_{(i,j-\alpha)}, z_{(i,j+\alpha)}$) in T-DAB, where $\alpha$ is the offset amplitude.

# C   Summary of other forecasting models

Here we provide a list of the models that we are comparing DEFORMTIME to:

– **LightTS** (Zhang et al., 2022) uses MLP layers as the building blocks to extract both inter-variable and intra-variable dependencies upon time series with different temporal resolutions.

– **DLinear** (Zeng et al., 2023) uses a fully connected layer along the temporal dimension with seasonal-trend decomposition to regress the historical values for future predictions.

– **Crossformer** (Zhang & Yan, 2023) is a transformer-based model that divides the input into patches and proposes a two-stage attention layer to capture inter- and intra-variable dependencies.

– **PatchTST** (Nie et al., 2023) is a transformer-based model that takes segmented patches as input tokens for the model. The prediction of each variable is designed to be independent of one another, i.e. with no inter-variable dependencies established.

– **iTransformer** (Liu et al., 2024) is a transformer-based model that embeds the sequence dimension rather than the input variable dimension. We note that although iTransformer captures inter-variable dependencies, it achieves that by embedding the input along the sequence dimension. This disrupts the temporal structure within each variable, leading to a sub-optimal performance in our empirical evaluation.

– **TimeMixer** (Wang et al., 2024a) is an MLP-based model that predicts the seasonal and trend components at different sampling scales and mixes forecasts to form the final prediction. Note that the authors consider variable mixing optional for their model. We opted to deploy the version that conducts variable mixing for the ILI rate forecasting tasks.

– **TimeXer** (Wang et al., 2024b) is a transformer-based model that captures exogenous information with cross-attention. It currently claims to offer SOTA accuracy on longer-term forecasting tasks.[17] Similarly to iTransformer, TimeXer embeds the exogenous variables along the sequence dimension which disrupts the temporal structure.

– **ModernTCN** (Luo & Wang, 2024b) is a CNN-based model that convolutes over both temporal and variable dimensions respectively with large receptive fields to capture inter- and intra-variable dependencies.

– **CycleNet** (Lin et al., 2024) is an MLP-based model that disentangles periodic patterns to capture temporal dependencies. Similarly to PatchTST, CycleNet predicts each variable independently. Note that the authors propose two versions of CycleNet. We report results for the version that has 1 linear layer, which has higher accuracy for overlapping tasks (ETT and weather).

– The **persistence model** is a naïve baseline that uses the last seen value of the target variable in the input as the forecast.

# D   Supplementary experiment settings

This section is supplementary to section 4.1 in the main paper. All experiments were conducted using a Linux server with 2 NVIDIA A40 GPUs (and 2 AMD EPYC 7443 CPUs), except for the ablation experiments which were conducted using a Linux server with 3 NVIDIA L40S GPUs (and 2 AMD EPYC 9354 CPUs).

## D.1   Input variable re-arrangement based on correlation with the target variable

In the ETT and weather forecasting tasks, we rearrange variables (see NAE in section 3.2) based on their linear correlation with the target variable across all time steps preceding the validation set. In the ILI rate forecasting tasks where training sets cover a period of 9 years prior to the test period, we only use the last 5 influenza seasons in the training set to obtain correlations. This is because both user search behaviour and search engine characteristics (e.g. automatic search suggestions) evolve, and this approach to feature selection reinforces a weak recency effect that is ultimately beneficial to prediction accuracy (Yang et al., 2015; Morris et al., 2023).

---

[17]Long-term forecast tasks ranking, `github.com/thuml/Time-Series-Library`

Table S1: ILI rate forecasting results in England (ILI-ENG) using web search frequency time series as exogenous variables. $\rho$ and $\epsilon$ % denote linear correlation and sMAPE, respectively. The best results are in **bold** font and the second best are underlined.

| | | 2015/16 | | | 2016/17 | | | 2017/18 | | | 2018/19 | | | Average | | |
|---|---|---|---|---|---|---|---|---|---|---|---|---|---|---|---|---|
| H | Model | $\rho$ | MAE | $\epsilon$ % | $\rho$ | MAE | $\epsilon$ % | $\rho$ | MAE | $\epsilon$ % | $\rho$ | MAE | $\epsilon$ % | $\rho$ | MAE | $\epsilon$ % |
| | Persistence | 0.9072 | 1.7077 | 22.43 | 0.9129 | 1.2287 | 22.67 | 0.8539 | 4.0441 | 31.08 | 0.8944 | 1.7037 | 23.67 | 0.8921 | 2.1710 | **24.96** |
| | DLinear | 0.8710 | 2.6985 | 38.86 | 0.8769 | 1.6879 | 33.77 | 0.8231 | 4.2734 | 50.14 | 0.8906 | 2.6258 | 49.30 | 0.8654 | 2.8214 | 43.02 |
| | PatchTST | 0.8832 | 1.9823 | 26.13 | 0.8744 | 1.4307 | 25.39 | 0.8475 | 3.7925 | 31.23 | 0.8526 | 2.0403 | 27.70 | 0.8644 | 2.3115 | 27.61 |
| | CycleNet | 0.8750 | 2.0579 | 27.08 | 0.8598 | 1.5396 | 27.54 | 0.7891 | 4.5348 | 35.72 | 0.8473 | 2.0892 | 28.02 | 0.8428 | 2.5554 | 29.59 |
| | iTransformer | 0.8773 | 2.0645 | 27.26 | 0.8911 | 1.3734 | 24.06 | 0.8180 | 4.1077 | 31.26 | 0.8911 | 1.6881 | 22.94 | 0.8694 | 2.3084 | 26.38 |
| 7 | TimeXer | 0.8927 | 1.9807 | 26.29 | 0.8773 | 1.5327 | 29.94 | 0.6974 | 5.1695 | 43.18 | 0.7564 | 2.5507 | 35.22 | 0.8060 | 2.8084 | 33.66 |
| | TimeMixer | 0.9056 | 1.7179 | 22.27 | 0.8616 | 1.4676 | 24.97 | 0.8694 | 3.4828 | 28.09 | 0.8388 | 2.0310 | 27.39 | 0.8688 | 2.1748 | 25.68 |
| | LightTS | 0.8385 | 2.2625 | 38.87 | 0.9432 | 2.4880 | 89.24 | 0.9441 | 2.6512 | 52.01 | 0.9426 | 1.5570 | 28.87 | 0.9171 | 2.2397 | 52.25 |
| | Crossformer | 0.8879 | 2.3314 | 36.08 | 0.9591 | **0.8606** | **17.82** | 0.9234 | 2.9133 | **27.04** | 0.9436 | **1.3737** | **21.89** | 0.9285 | 1.8698 | 25.71 |
| | ModernTCN | 0.9490 | 1.6043 | **17.82** | 0.9606 | 1.1988 | 22.75 | **0.9679** | 3.2340 | 40.55 | 0.9316 | 1.7584 | 31.98 | 0.9523 | 1.9489 | 28.28 |
| | DEFORMTIME | **0.9728** | **1.3786** | 19.57 | 0.9665 | 1.3083 | 29.49 | 0.9585 | 2.6313 | 41.40 | 0.9436 | **1.2485** | 23.96 | **0.9603** | **1.6417** | 28.61 |
| | Persistence | 0.8292 | 2.3161 | 30.16 | 0.8414 | 1.6867 | 29.54 | 0.7289 | 5.8180 | 42.30 | 0.8031 | 2.4291 | 33.08 | 0.8006 | 3.0625 | 33.77 |
| | DLinear | 0.7810 | 3.6354 | 52.29 | 0.7857 | 2.1872 | 42.47 | 0.6834 | 5.5368 | 60.91 | 0.8351 | 3.8092 | 65.47 | 0.7713 | 3.7922 | 55.28 |
| | PatchTST | 0.8078 | 2.6267 | 35.59 | 0.7730 | 1.9562 | 33.55 | 0.6722 | 5.6075 | 43.64 | 0.7218 | 2.8284 | 38.25 | 0.7437 | 3.2547 | 37.76 |
| | CycleNet | 0.7970 | 2.6843 | 35.97 | 0.7402 | 2.1182 | 36.59 | 0.6127 | 6.0212 | 47.53 | 0.7326 | 2.7406 | 37.61 | 0.7206 | 3.3911 | 39.42 |
| | iTransformer | 0.7731 | 2.7620 | 36.46 | 0.7776 | 1.9929 | 34.17 | 0.6576 | 5.7796 | 44.34 | 0.7850 | 2.3862 | 31.71 | 0.7483 | 3.2301 | 36.67 |
| 14 | TimeXer | 0.7891 | 2.6580 | 35.27 | 0.7910 | 1.9864 | 36.99 | 0.5503 | 6.2678 | 53.64 | 0.6646 | 3.0626 | 41.61 | 0.6988 | 3.4937 | 41.88 |
| | TimeMixer | 0.7706 | 2.6783 | 34.25 | 0.8684 | 1.5621 | 29.06 | 0.7164 | 5.0940 | 39.59 | 0.7503 | 2.7491 | 38.66 | 0.7764 | 3.0209 | 35.39 |
| | LightTS | 0.8109 | 2.5494 | 32.28 | 0.9233 | 1.6292 | 36.46 | 0.7821 | 4.4367 | 39.46 | 0.8915 | 2.1365 | 44.97 | 0.8519 | 2.6879 | 38.29 |
| | Crossformer | 0.7940 | 2.9124 | 41.75 | 0.8843 | 1.4711 | 23.39 | 0.8470 | 3.8295 | 30.88 | 0.9007 | 2.4044 | **27.86** | 0.8565 | 2.6543 | **30.97** |
| | ModernTCN | 0.8814 | 2.1727 | 31.93 | 0.8864 | 1.4953 | 28.61 | 0.8905 | 4.6759 | 48.74 | 0.8441 | 2.4760 | 34.76 | 0.8756 | 2.7050 | 36.01 |
| | DEFORMTIME | **0.9259** | **2.0556** | **25.36** | **0.9400** | **1.3180** | 33.86 | **0.8964** | 3.7631 | 47.33 | **0.9154** | **1.7863** | 29.36 | **0.9194** | **2.2308** | 33.98 |
| | Persistence | 0.7357 | 2.8604 | 36.98 | 0.7465 | 2.1552 | 36.45 | 0.6002 | 7.3119 | 52.46 | 0.6952 | 3.1193 | 42.23 | 0.6944 | 3.8617 | 42.03 |
| | DLinear | 0.4466 | 4.1958 | 56.02 | 0.6441 | 2.8294 | 52.56 | 0.4920 | 6.9514 | 71.66 | 0.6608 | 3.9192 | 64.75 | 0.5609 | 4.4739 | 61.25 |
| | PatchTST | 0.6126 | 3.6607 | 48.57 | 0.6214 | 2.6377 | 44.77 | 0.4427 | 7.3828 | 60.54 | 0.5861 | 3.5956 | 50.55 | 0.5657 | 4.3192 | 51.11 |
| | CycleNet | 0.5596 | 3.9238 | 52.84 | 0.5857 | 2.7603 | 46.01 | 0.4114 | 7.7331 | 65.57 | 0.5988 | 3.3905 | 46.92 | 0.5389 | 4.4519 | 52.83 |
| | iTransformer | 0.5310 | 4.1390 | 55.62 | 0.6981 | 2.6718 | 43.65 | 0.4403 | 7.0816 | 55.17 | 0.6556 | 3.0464 | 41.29 | 0.5564 | 4.2347 | 48.93 |
| 21 | TimeXer | 0.5958 | 3.7654 | 49.46 | 0.6981 | 2.4045 | 41.17 | 0.4377 | 7.3686 | 62.71 | 0.5301 | 3.7962 | 52.91 | 0.5654 | 4.3337 | 51.56 |
| | TimeMixer | 0.5398 | 3.8805 | 56.71 | 0.7799 | 1.7940 | 30.45 | 0.7497 | 5.5338 | 58.68 | 0.7539 | 2.9919 | 51.58 | 0.7058 | 3.5501 | 49.36 |
| | LightTS | 0.7082 | 3.3630 | 45.16 | 0.8696 | 1.9335 | 58.65 | 0.8132 | 5.6422 | 51.74 | 0.8594 | 2.5078 | 51.59 | 0.8126 | 3.3616 | 51.78 |
| | Crossformer | 0.8113 | 2.6710 | 36.42 | 0.8014 | 1.8035 | 31.29 | 0.8504 | **4.5615** | 49.08 | 0.7331 | 2.9697 | 45.48 | 0.7991 | 3.0014 | 40.57 |
| | ModernTCN | 0.7869 | 2.3283 | 27.42 | 0.8590 | **1.6790** | 30.86 | 0.9085 | 5.7489 | 61.11 | 0.8357 | 2.4040 | 40.67 | 0.8475 | 3.0400 | 40.02 |
| | DEFORMTIME | **0.8819** | **2.1700** | **27.01** | **0.8859** | 1.8980 | 32.65 | **0.9110** | 4.8984 | 44.78 | **0.9092** | **1.6335** | 26.36 | **0.8970** | **2.6500** | **32.70** |
| | Persistence | 0.6408 | 3.3786 | 43.17 | 0.6344 | 2.6018 | 42.68 | 0.4813 | 8.5576 | 60.33 | 0.5733 | 3.8047 | 51.79 | 0.5825 | 4.5857 | 49.49 |
| | DLinear | 0.3917 | 4.5039 | 60.30 | 0.5078 | 3.3928 | 60.94 | 0.3911 | 7.7738 | 77.58 | 0.5774 | 4.4683 | 72.19 | 0.4670 | 5.0347 | 67.75 |
| | PatchTST | 0.4271 | 4.5240 | 60.22 | 0.4477 | 3.2658 | 53.01 | 0.3203 | 8.0693 | 65.91 | 0.4486 | 4.1265 | 59.27 | 0.4109 | 4.9964 | 59.60 |
| | CycleNet | 0.4681 | 4.3576 | 58.52 | 0.4710 | 3.1149 | 51.40 | 0.3109 | 8.7494 | 73.07 | 0.5037 | 3.8816 | 56.73 | 0.4384 | 5.0259 | 59.93 |
| | iTransformer | 0.3492 | 4.9542 | 65.45 | 0.6761 | 2.2745 | 38.38 | 0.2947 | 8.3463 | 69.39 | 0.5286 | 3.6750 | 48.18 | 0.4622 | 4.8125 | 55.35 |
| 28 | TimeXer | 0.4978 | 4.2217 | 56.56 | 0.5523 | 2.8584 | 48.91 | 0.3413 | 8.1493 | 71.44 | 0.3337 | 4.3758 | 69.50 | 0.4313 | 4.9013 | 61.60 |
| | TimeMixer | 0.5395 | 3.8587 | 57.17 | 0.6890 | 2.4063 | 45.76 | 0.4882 | 6.7783 | 62.93 | 0.7592 | 2.9567 | 51.21 | 0.6190 | 4.0000 | 54.27 |
| | LightTS | 0.5812 | 3.4254 | 43.91 | 0.8311 | 1.9020 | 36.26 | 0.7151 | 6.0675 | 83.54 | 0.8900 | 2.2578 | 58.66 | 0.7543 | 3.4132 | 55.59 |
| | Crossformer | 0.7299 | 3.1253 | 41.56 | 0.8082 | 1.8197 | **33.57** | 0.7841 | 5.4882 | 55.97 | 0.9349 | 2.3599 | 53.48 | 0.8143 | 3.1983 | 46.14 |
| | ModernTCN | **0.8017** | **2.6486** | **36.17** | 0.7628 | 2.6508 | 51.35 | 0.7544 | **4.8228** | 49.19 | 0.8152 | 3.3222 | 54.78 | 0.7835 | 3.3611 | 47.87 |
| | DEFORMTIME | 0.7919 | 2.6540 | 49.29 | **0.9229** | **1.8100** | 41.87 | **0.8493** | 4.8324 | **37.53** | **0.9499** | **1.5947** | **33.06** | **0.8785** | **2.7228** | **40.44** |

## D.2 Hyperparameters specific to the ILI forecasting task

In the ILI rate forecasting task, the number of exogenous variables we use is a learnable parameter that depends on the linear correlation threshold $\tau$ (see Appendix A.3). For US regions, $\tau$ is selected from $\{.3, .4, .5\}$. For England, $\tau$ is selected from $\{.05, .1, .2, .3, .4, .5\}$. Given these thresholds, a model can select from 13 to 51 search queries for US regions and from 80 to 752 for England (the higher the correlation threshold, the fewer queries are being selected). Note that for ModernTCN, PatchTST and Crossformer, we restricted $\tau \in \{.3, .4, .5\}$ for all locations as the models required an excessive amount of GPU memory for larger sets of exogenous variables (see also section 4.5 for PatchTST and Crossformer).

## D.3 Random seed initialisation

For the ILI rate forecasting tasks, we use a fixed seed equal to '42' during training. For the ETTh1, ETTh2, and weather tasks, we set this to '2021' based on previous work (Nie et al., 2023; Zeng et al., 2023; Wang et al., 2024a), with the exception of iTransformer ('2023'), and CycleNet, where results are averaged across 5 seeds ('2024', '2025', '2026', '2027', '2028') in accordance with their official configuration.

Table S2: ILI rate forecasting results in US Region 2 (ILI-US2) using web search frequency time series as exogenous variables. $\rho$ and $\epsilon$ % denote linear correlation and sMAPE, respectively. The best results are in **bold** font and the second best are underlined.

| H | Model | 2015/16 | | | 2016/17 | | | 2017/18 | | | 2018/19 | | | Average | | |
|---|---|---|---|---|---|---|---|---|---|---|---|---|---|---|---|---|
| | | $\rho$ | MAE | $\epsilon$ % | $\rho$ | MAE | $\epsilon$ % | $\rho$ | MAE | $\epsilon$ % | $\rho$ | MAE | $\epsilon$ % | $\rho$ | MAE | $\epsilon$ % |
| 7 | Persistence | 0.7758 | 0.4114 | 22.26 | 0.8271 | 0.6803 | 24.57 | 0.7589 | 1.0284 | 25.48 | 0.8954 | 0.4696 | 17.59 | 0.8143 | 0.6474 | 22.48 |
| | DLinear | 0.7456 | 0.4982 | 29.23 | 0.7980 | 0.8210 | 31.85 | 0.7133 | 1.0910 | 30.82 | 0.8955 | 0.5318 | 19.84 | 0.7881 | 0.7355 | 27.94 |
| | PatchTST | 0.7666 | 0.4290 | 23.78 | 0.7973 | 0.7546 | 26.97 | 0.6937 | 1.1018 | 26.52 | 0.8624 | 0.5533 | 20.79 | 0.7800 | 0.7097 | 24.52 |
| | CycleNet | 0.7577 | 0.4461 | 24.63 | 0.8134 | 0.7300 | 26.99 | 0.7345 | 1.0930 | 27.47 | 0.8782 | 0.5113 | 19.58 | 0.7959 | 0.6951 | 24.67 |
| | iTransformer | 0.7986 | 0.3865 | **21.30** | 0.8171 | 0.7367 | 27.07 | 0.7784 | 0.9080 | 23.22 | 0.8545 | 0.5717 | 21.38 | 0.8122 | 0.6507 | 23.24 |
| | TimeXer | 0.8024 | 0.4214 | 24.76 | 0.8480 | 0.6882 | 27.16 | 0.8494 | 0.8233 | 22.90 | 0.8859 | 0.5001 | 18.70 | 0.8464 | 0.6083 | 23.38 |
| | TimeMixer | 0.8237 | 0.3842 | 21.76 | 0.8710 | 0.6449 | 24.16 | 0.9124 | 0.6255 | 17.87 | 0.8877 | 0.4588 | 16.49 | 0.8737 | 0.5284 | 20.07 |
| | LightTS | 0.7930 | 0.4045 | 23.23 | 0.9056 | 0.5806 | 18.01 | 0.9359 | 0.5688 | 15.39 | 0.9451 | 0.2989 | **10.33** | 0.8949 | 0.4632 | 16.74 |
| | Crossformer | 0.8796 | **0.3382** | 21.39 | 0.9107 | 0.4975 | 18.32 | 0.9009 | 0.6309 | 15.61 | 0.9523 | 0.2936 | 10.52 | 0.9109 | 0.4400 | 16.46 |
| | ModernTCN | 0.8745 | 0.3874 | 21.99 | 0.9359 | 0.5216 | **17.45** | **0.9569** | **0.4782** | 14.12 | 0.9535 | 0.3722 | 12.64 | **0.9302** | 0.4398 | 16.55 |
| | DeformTime | **0.8887** | 0.3428 | 21.86 | **0.9463** | **0.4796** | 18.66 | 0.9008 | 0.5369 | **12.88** | **0.9622** | **0.2894** | 10.64 | 0.9245 | **0.4122** | **16.01** |
| 14 | Persistence | 0.6872 | 0.5000 | 26.67 | 0.7617 | 0.8545 | 31.23 | 0.6331 | 1.3027 | 32.42 | 0.8436 | 0.5966 | 22.65 | 0.7314 | 0.8135 | 28.24 |
| | DLinear | 0.6546 | 0.5761 | 33.62 | 0.7335 | 0.9406 | 36.90 | 0.6007 | 1.2213 | 34.62 | 0.8261 | 0.6360 | 23.74 | 0.7037 | 0.8435 | 32.22 |
| | PatchTST | 0.6628 | 0.5103 | 27.27 | 0.7350 | 0.9267 | 34.44 | 0.5993 | 1.3085 | 31.82 | 0.7922 | 0.7085 | 26.90 | 0.6973 | 0.8635 | 30.11 |
| | CycleNet | 0.7019 | 0.4754 | 25.40 | 0.7208 | 0.9457 | 35.28 | 0.6201 | 1.2199 | 30.54 | 0.8265 | 0.6468 | 25.39 | 0.7173 | 0.8219 | 29.15 |
| | iTransformer | 0.6864 | 0.4894 | 26.35 | 0.7548 | 0.8934 | 33.26 | 0.6025 | 1.1639 | 30.28 | 0.8294 | 0.6115 | 22.78 | 0.7183 | 0.7896 | 28.17 |
| | TimeXer | 0.7178 | 0.4823 | 27.24 | 0.7446 | 0.9211 | 36.47 | 0.7118 | 1.0505 | 28.09 | 0.8164 | 0.6362 | 24.47 | 0.7476 | 0.7725 | 29.07 |
| | TimeMixer | 0.7741 | 0.4438 | 25.01 | 0.8403 | 0.7314 | 27.32 | 0.8043 | 0.8911 | 25.13 | 0.8622 | 0.5560 | 20.99 | 0.8202 | 0.6556 | 24.61 |
| | LightTS | 0.7203 | 0.5086 | 28.62 | 0.8601 | 0.7079 | 22.95 | 0.9191 | 0.6698 | 23.47 | 0.9409 | 0.4445 | 17.40 | 0.8601 | 0.5827 | 23.11 |
| | Crossformer | 0.7872 | **0.4357** | 24.61 | 0.8640 | 0.6502 | 22.95 | 0.8057 | 0.8159 | 20.83 | 0.9241 | 0.4389 | 15.54 | 0.8453 | 0.5852 | 20.98 |
| | ModernTCN | **0.8093** | 0.4647 | 25.78 | 0.9000 | 0.6482 | 23.18 | **0.9360** | **0.6020** | **14.87** | 0.9572 | 0.3968 | 14.87 | 0.9006 | 0.5279 | 20.22 |
| | DeformTime | 0.8050 | 0.4404 | **23.82** | **0.9271** | **0.5029** | **18.53** | 0.9126 | 0.6351 | 16.34 | **0.9652** | **0.3226** | **12.21** | **0.9025** | **0.4752** | **17.73** |
| 21 | Persistence | 0.5918 | 0.5680 | 30.04 | 0.7017 | 1.0055 | 37.20 | 0.5137 | 1.5443 | 38.93 | 0.7792 | 0.7361 | 27.88 | 0.6466 | 0.9635 | 33.51 |
| | DLinear | 0.6016 | 0.6023 | 35.11 | 0.6897 | 0.9671 | 38.06 | 0.4966 | 1.3272 | 38.76 | 0.7936 | 0.7530 | 27.80 | 0.6454 | 0.9124 | 34.93 |
| | PatchTST | 0.5775 | 0.5592 | 29.41 | 0.6286 | 1.1661 | 43.41 | 0.5189 | 1.4395 | 38.13 | 0.6478 | 0.9495 | 35.86 | 0.5932 | 1.0286 | 36.70 |
| | CycleNet | 0.6097 | 0.5216 | 27.54 | 0.5883 | 1.1875 | 44.72 | 0.4663 | 1.4768 | 41.27 | 0.5979 | 1.0018 | 38.41 | 0.5656 | 1.0469 | 37.98 |
| | iTransformer | 0.6418 | 0.4790 | **24.54** | 0.7515 | 0.8784 | 34.22 | 0.8230 | 1.0845 | 31.85 | 0.7321 | 0.7748 | 29.52 | 0.7371 | 0.8042 | 30.03 |
| | TimeXer | 0.4856 | 0.5815 | 32.17 | 0.7739 | 0.8696 | 33.97 | 0.6529 | 1.1238 | 32.12 | 0.7482 | 0.7225 | 27.56 | 0.6651 | 0.8243 | 31.46 |
| | TimeMixer | 0.7368 | 0.4655 | 27.79 | 0.8281 | 0.7853 | 31.00 | 0.8509 | 0.9275 | 30.07 | 0.8803 | 0.5394 | 21.86 | 0.8240 | 0.6794 | 27.68 |
| | LightTS | **0.8484** | 0.5606 | 32.47 | 0.8936 | 0.6749 | 34.80 | 0.9177 | 0.8065 | 28.47 | 0.8951 | 0.6310 | 21.34 | **0.8887** | 0.6683 | 29.27 |
| | Crossformer | 0.7729 | 0.4634 | 26.54 | 0.8996 | 0.6688 | **25.62** | 0.7621 | 0.8882 | 23.17 | 0.9354 | 0.4777 | **13.82** | 0.8425 | 0.6245 | 22.29 |
| | ModernTCN | 0.8104 | 0.4656 | 26.17 | 0.8746 | 0.6790 | 30.30 | 0.9105 | 0.7258 | **20.75** | **0.9487** | 0.4421 | 18.19 | 0.8860 | 0.5781 | 23.85 |
| | DeformTime | 0.7414 | **0.4568** | 25.94 | **0.9028** | **0.6189** | 26.44 | **0.9313** | **0.6981** | 21.04 | 0.9408 | **0.3963** | 15.09 | 0.8791 | **0.5425** | **22.13** |
| 28 | Persistence | 0.4860 | 0.6393 | 33.84 | 0.6374 | 1.1447 | 42.55 | 0.4046 | 1.7516 | 44.79 | 0.7071 | 0.8674 | 32.98 | 0.5588 | 1.1007 | 38.54 |
| | DLinear | 0.4836 | 0.6512 | 37.96 | 0.6112 | 1.0328 | 41.19 | 0.4125 | 1.4045 | 40.27 | 0.6840 | 0.8337 | 31.08 | 0.5479 | 0.9805 | 37.62 |
| | PatchTST | 0.4013 | 0.6678 | 36.55 | 0.5274 | 1.2615 | 48.85 | 0.3903 | 1.6234 | 44.67 | 0.5579 | 1.0576 | 40.39 | 0.4692 | 1.1525 | 42.61 |
| | CycleNet | 0.4497 | 0.6150 | 33.74 | 0.5323 | 1.2092 | 46.79 | 0.3997 | 1.5371 | 43.37 | 0.4523 | 1.1940 | 45.34 | 0.4585 | 1.1388 | 42.31 |
| | iTransformer | 0.5258 | 0.5949 | 32.54 | 0.6567 | 1.0632 | 41.38 | 0.5726 | 1.3053 | 38.82 | 0.6738 | 0.8842 | 34.26 | 0.6072 | 0.9619 | 36.75 |
| | TimeXer | 0.3979 | 0.5856 | 32.82 | 0.7017 | 0.9748 | 39.33 | 0.6137 | 1.2462 | 35.29 | 0.7037 | 0.8229 | 31.45 | 0.6042 | 0.9074 | 34.72 |
| | TimeMixer | 0.5756 | 0.5835 | 36.06 | 0.7251 | 1.0106 | 39.21 | 0.7694 | 1.2470 | 43.49 | 0.7467 | 0.7000 | 27.37 | 0.7042 | 0.8853 | 36.53 |
| | LightTS | 0.8262 | 0.6018 | 33.56 | 0.8332 | 0.8180 | 31.54 | 0.8556 | 0.8913 | 27.03 | 0.9011 | 0.5589 | 18.78 | 0.8540 | 0.7175 | 27.73 |
| | Crossformer | 0.7102 | **0.4697** | 27.38 | 0.9064 | 0.7624 | 28.93 | 0.8432 | 0.8253 | 20.69 | 0.8782 | 0.5474 | 18.64 | 0.8345 | 0.6512 | 23.91 |
| | ModernTCN | 0.7635 | 0.4702 | **27.09** | 0.8667 | 0.6742 | 27.31 | 0.9223 | 0.6618 | 19.83 | **0.9604** | **0.4776** | 20.41 | 0.8782 | 0.5710 | 23.66 |
| | DeformTime | **0.8369** | 0.4716 | 27.21 | **0.9562** | **0.6017** | **25.66** | **0.9253** | **0.5954** | **17.62** | 0.9294 | 0.5463 | **18.52** | **0.9119** | **0.5538** | **22.25** |

## D.4 Data normalisation

In all forecasting tasks, we standardise all variables (zero mean, unit standard deviation), each time based on the training data. The prediction output is de-normalised back to its original scale prior to being compared with the (de-normalised) ground truth. In addition to that, for the ETT and weather data sets, we also standardise each variable within the input's look-back window, in accordance with previous work (Liu et al., 2022b; Zhou et al., 2022a; Nie et al., 2023).

# E Supplementary results

## E.1 Symmetric Mean Absolute Percentage Error

To compare results across different tasks that have different units, we use the sMAPE metric. We note the general (and correct) perception that occasionally sMAPE may mislead (by over- or under-estimating error), and hence it can only be used for partial insights in conjunction with a more robust error metric. Hence, our

Table S3: ILI rate forecasting results in US Region 9 (ILI-US9) using web search frequency time series as exogenous variables. $\rho$ and $\epsilon$ % denote linear correlation and sMAPE, respectively. The best results are in **bold** font and the second best are underlined.

| | | 2015/16 | | | 2016/17 | | | 2017/18 | | | 2018/19 | | | Average | | |
|---|---|---|---|---|---|---|---|---|---|---|---|---|---|---|---|---|
| H | Model | $\rho$ | MAE | $\epsilon$ % | $\rho$ | MAE | $\epsilon$ % | $\rho$ | MAE | $\epsilon$ % | $\rho$ | MAE | $\epsilon$ % | $\rho$ | MAE | $\epsilon$ % |
| | Persistence | 0.8384 | 0.3945 | 18.58 | 0.8494 | 0.2791 | 16.25 | 0.7304 | 0.5949 | 21.55 | 0.8898 | 0.3543 | 17.57 | 0.8270 | 0.4057 | 18.49 |
| | DLinear | 0.8071 | 0.4769 | 25.07 | 0.8092 | 0.3727 | 22.02 | 0.6759 | 0.5704 | 22.96 | 0.8925 | 0.4499 | 23.82 | 0.7962 | 0.4675 | 23.47 |
| | PatchTST | 0.8248 | 0.3993 | 19.30 | 0.8318 | 0.2802 | 16.91 | 0.7068 | 0.6078 | 23.25 | 0.8796 | 0.3592 | 17.91 | 0.8107 | 0.4116 | 19.34 |
| | CycleNet | 0.8101 | 0.4156 | 20.13 | 0.7754 | 0.3265 | 19.50 | 0.6685 | 0.6897 | 26.34 | 0.8843 | 0.3603 | 18.22 | 0.7846 | 0.4480 | 21.05 |
| | iTransformer | 0.8380 | 0.3736 | 17.55 | 0.8446 | 0.2721 | 16.22 | 0.6664 | 0.6856 | 26.00 | 0.9230 | 0.2915 | 14.52 | 0.8180 | 0.4057 | 18.57 |
| 7 | TimeXer | 0.8237 | 0.4071 | 20.04 | 0.8471 | 0.2669 | 15.50 | 0.7638 | 0.5235 | 20.92 | 0.9021 | 0.3277 | 16.36 | 0.8342 | 0.3813 | 18.20 |
| | TimeMixer | 0.9079 | 0.2892 | 13.61 | 0.8521 | 0.2733 | 16.29 | 0.7923 | 0.5049 | 18.47 | 0.9525 | 0.2435 | 12.48 | 0.8762 | 0.3239 | 15.21 |
| | LightTS | 0.8896 | 0.2790 | 13.63 | 0.8858 | 0.2931 | 16.95 | 0.8277 | 0.4474 | 16.82 | 0.9433 | 0.2543 | 15.20 | 0.8866 | 0.3185 | 15.65 |
| | Crossformer | **0.9305** | 0.2556 | **12.21** | **0.9524** | 0.2874 | 14.89 | 0.8213 | 0.4524 | 17.08 | 0.9435 | 0.2640 | 13.58 | 0.9119 | 0.3149 | 14.44 |
| | ModernTCN | 0.9263 | 0.2615 | 13.08 | 0.8886 | 0.2482 | 14.86 | 0.8706 | 0.4745 | 20.37 | **0.9770** | **0.1754** | **8.37** | 0.9156 | 0.2899 | 14.17 |
| | DEFORMTIME | 0.9161 | **0.2437** | 12.46 | 0.9356 | **0.2364** | **11.69** | **0.8744** | **0.3664** | **13.61** | 0.9675 | 0.2023 | 11.29 | **0.9234** | **0.2622** | **12.26** |
| | Persistence | 0.7572 | 0.4875 | 23.00 | 0.7869 | 0.3385 | 19.84 | 0.6439 | 0.7283 | 26.96 | 0.8239 | 0.4488 | 22.49 | 0.7530 | 0.5008 | 23.07 |
| | DLinear | 0.7323 | 0.5767 | 29.99 | 0.7371 | 0.4384 | 25.44 | 0.6105 | 0.6635 | 27.69 | 0.8216 | 0.5085 | 26.27 | 0.7254 | 0.5467 | 27.35 |
| | PatchTST | 0.7626 | 0.4810 | 23.33 | 0.7411 | 0.3524 | 21.48 | 0.6246 | 0.7164 | 28.11 | 0.8102 | 0.4583 | 23.44 | 0.7346 | 0.5020 | 24.09 |
| | CycleNet | 0.7369 | 0.5006 | 24.59 | 0.7279 | 0.3686 | 22.02 | 0.6372 | 0.7082 | 26.12 | 0.8256 | 0.4513 | 23.34 | 0.7319 | 0.5072 | 24.02 |
| | iTransformer | 0.7837 | 0.4310 | 20.63 | 0.7629 | 0.3448 | 20.49 | 0.6138 | 0.6853 | 27.21 | 0.8495 | 0.4196 | 21.41 | 0.7525 | 0.4702 | 22.44 |
| 14 | TimeXer | 0.7423 | 0.4795 | 23.26 | 0.7872 | 0.3078 | 18.07 | 0.6443 | 0.6440 | 25.49 | 0.8261 | 0.4345 | 21.76 | 0.7500 | 0.4665 | 22.14 |
| | TimeMixer | 0.8319 | 0.3600 | 17.41 | 0.8221 | 0.3153 | 18.22 | 0.6668 | 0.6111 | 23.48 | 0.9139 | 0.3377 | 17.21 | 0.8087 | 0.4060 | 19.08 |
| | LightTS | 0.8697 | 0.4018 | 20.65 | 0.8237 | 0.3418 | 20.50 | 0.7802 | 0.4999 | 20.31 | 0.9366 | 0.2730 | 14.70 | 0.8525 | 0.3791 | 19.04 |
| | Crossformer | 0.8787 | 0.3333 | 16.05 | 0.8965 | 0.2980 | 16.74 | 0.8339 | 0.4325 | 16.91 | 0.9169 | 0.3647 | 19.21 | 0.8815 | 0.3571 | 17.23 |
| | ModernTCN | **0.9250** | 0.3582 | 15.44 | 0.8543 | 0.3175 | 17.00 | **0.8690** | 0.4244 | 15.32 | 0.9320 | 0.2666 | 13.40 | **0.8952** | 0.3417 | 15.29 |
| | DEFORMTIME | 0.9033 | **0.2962** | **14.39** | **0.9374** | **0.2893** | **13.15** | 0.8490 | **0.3897** | **14.62** | **0.9378** | **0.2584** | **13.05** | 0.9069 | **0.3084** | **13.80** |
| | Persistence | 0.6682 | 0.5831 | 27.54 | 0.7150 | 0.4013 | 23.26 | 0.5734 | 0.8466 | 31.93 | 0.7520 | 0.5313 | 26.92 | 0.6772 | 0.5906 | 27.41 |
| | DLinear | 0.6687 | 0.6032 | 30.66 | 0.6920 | 0.4960 | 28.49 | 0.5830 | 0.7252 | 30.15 | 0.7778 | 0.5760 | 29.33 | 0.6804 | 0.6001 | 29.66 |
| | PatchTST | 0.6899 | 0.5705 | 28.41 | 0.6399 | 0.4255 | 26.14 | 0.5214 | 0.7963 | 32.80 | 0.7137 | 0.5815 | 30.27 | 0.6412 | 0.5935 | 29.40 |
| | CycleNet | 0.7092 | 0.5656 | 28.42 | 0.6695 | 0.4043 | 24.79 | 0.5731 | 0.8140 | 30.35 | 0.6997 | 0.5864 | 30.31 | 0.6629 | 0.5926 | 28.47 |
| | iTransformer | 0.8659 | 0.3645 | 17.75 | 0.7078 | 0.3702 | 21.73 | 0.4561 | 0.8642 | 33.94 | 0.8298 | 0.4433 | 23.01 | 0.7149 | 0.5106 | 24.11 |
| 21 | TimeXer | 0.6203 | 0.5899 | 29.37 | 0.6373 | 0.4132 | 24.19 | 0.5158 | 0.7825 | 31.10 | 0.7460 | 0.5006 | 25.04 | 0.6298 | 0.5715 | 27.42 |
| | TimeMixer | 0.8280 | 0.4112 | 19.07 | 0.8180 | 0.3058 | 17.67 | 0.6555 | 0.6812 | 25.45 | 0.8634 | 0.4322 | 23.41 | 0.7912 | 0.4576 | 21.40 |
| | LightTS | 0.8354 | 0.4252 | 20.19 | 0.7748 | 0.4021 | 25.39 | 0.6920 | 0.6201 | 25.57 | 0.8699 | 0.4542 | 23.80 | 0.7930 | 0.4754 | 23.74 |
| | Crossformer | **0.9565** | 0.3267 | 15.49 | 0.8669 | 0.3168 | 15.95 | 0.8777 | 0.3964 | 15.11 | 0.9135 | 0.3273 | 17.07 | **0.9036** | 0.3418 | 15.90 |
| | ModernTCN | 0.8695 | **0.3118** | **12.39** | 0.8775 | 0.3695 | 16.88 | 0.7572 | 0.5057 | 16.88 | 0.8997 | **0.2968** | 14.08 | 0.8510 | 0.3710 | 15.43 |
| | DEFORMTIME | 0.8331 | 0.3543 | 15.62 | **0.8966** | **0.3039** | **14.99** | **0.9141** | **0.3536** | **13.06** | **0.9249** | 0.2598 | 13.27 | 0.8922 | **0.3179** | **14.24** |
| | Persistence | 0.5763 | 0.6781 | 32.15 | 0.6261 | 0.4733 | 27.09 | 0.5059 | 0.9541 | 36.29 | 0.6705 | 0.6140 | 31.14 | 0.5947 | 0.6799 | 31.67 |
| | DLinear | 0.6083 | 0.6496 | 32.81 | 0.6016 | 0.5329 | 30.18 | 0.5506 | 0.7893 | 32.75 | 0.7438 | 0.6539 | 32.89 | 0.6261 | 0.6564 | 32.16 |
| | PatchTST | 0.6067 | 0.6457 | 32.75 | 0.5192 | 0.4903 | 30.18 | 0.4424 | 0.8809 | 36.70 | 0.6555 | 0.6492 | 33.78 | 0.5560 | 0.6665 | 33.35 |
| | CycleNet | 0.6075 | 0.6436 | 31.94 | 0.6075 | 0.4860 | 30.15 | 0.4990 | 0.8940 | 35.95 | 0.4816 | 0.7887 | 39.96 | 0.5321 | 0.7031 | 34.50 |
| | iTransformer | 0.6984 | 0.5531 | 27.80 | 0.5779 | 0.4444 | 25.70 | 0.3211 | 0.9942 | 40.08 | 0.6797 | 0.6077 | 30.60 | 0.5693 | 0.6498 | 31.04 |
| 28 | TimeXer | 0.5345 | 0.6372 | 32.39 | 0.6320 | 0.4302 | 23.80 | 0.3998 | 0.9296 | 37.46 | 0.6127 | 0.6248 | 31.64 | 0.5448 | 0.6555 | 31.32 |
| | TimeMixer | 0.8130 | 0.4056 | 21.18 | 0.8184 | 0.3244 | 19.25 | 0.6444 | 0.7787 | 31.60 | 0.7816 | 0.5409 | 24.41 | 0.7644 | 0.5124 | 24.11 |
| | LightTS | 0.8462 | 0.4765 | 24.49 | 0.6952 | 0.4834 | 25.81 | 0.7176 | 0.5616 | 22.25 | 0.9037 | 0.3862 | 20.33 | 0.7907 | 0.4769 | 23.22 |
| | Crossformer | **0.8969** | **0.3398** | 16.47 | 0.8346 | 0.3647 | 16.99 | 0.8514 | 0.4365 | **15.42** | 0.8923 | 0.3578 | 16.87 | 0.8688 | 0.3747 | 16.44 |
| | ModernTCN | 0.8552 | 0.3611 | 17.38 | **0.9076** | 0.3748 | 17.59 | 0.8068 | 0.5156 | 18.99 | 0.8645 | 0.3245 | **14.79** | 0.8585 | 0.3940 | 17.19 |
| | DEFORMTIME | 0.8888 | 0.3718 | **15.55** | 0.9046 | **0.3153** | **14.57** | **0.9037** | 0.4185 | 16.18 | **0.9260** | **0.3069** | 16.68 | **0.9058** | 0.3532 | **15.74** |

main error metric is MAE. For the ILI rate forecasting tasks, we additionally show linear correlation as an established metric in related literature (Ginsberg et al., 2009; Lampos et al., 2015; Yang et al., 2015).

We use the following definition of sMAPE. For a series of estimates $\hat{\mathbf{y}} \in \mathbb{R}^n$ and a corresponding series of true values $\mathbf{y} \in \mathbb{R}^n$, sMAPE is given by

$$\texttt{sMAPE}(\hat{\mathbf{y}}, \mathbf{y}) = \frac{100}{n} \sum_{j=1}^{n} \frac{|\hat{y}_j - y_j|}{0.5\,(|\hat{y}_j| + |y_j|)} \ . \tag{S1}$$

In Tables throughout the manuscript, we denote sMAPE using $\epsilon$ %.

### E.2 Ablation study for DeformTime – Additional information

Here we provide additional details to supplement the ablation study presented in section 4.3.

$\neg$ V-DAB: Denotes that we train a DEFORMTIME model without the V-DAB encoder branch (see section 3.3). Within each encoder layer, we only use the T-DAB branch for sequence encoding. Instead of projecting the

Table S4: Ablation study on different groups $G$ for DEFORMTIME using the ILI-ENG and ILI-US9 data sets. Results are averaged across all 4 seasons.

| | | $G=2$ | | $G=4$ | | $G=8$ | | $G=16$ | | | | $G=2$ | | $G=4$ | | $G=8$ | | $G=16$ | |
|---|---|---|---|---|---|---|---|---|---|---|---|---|---|---|---|---|---|---|---|
| | $H$ | MAE | $\epsilon$ % | MAE | $\epsilon$ % | MAE | $\epsilon$ % | MAE | $\epsilon$ % | | $H$ | MAE | $\epsilon$ % | MAE | $\epsilon$ % | MAE | $\epsilon$ % | MAE | $\epsilon$ % |
| ILI-ENG | 7 | 1.7733 | 34.79 | **1.6417** | 28.61 | 1.7325 | **23.52** | 1.7085 | 24.73 | ILI-US9 | 7 | 0.2665 | 12.80 | **0.2622** | 12.26 | 0.2650 | **11.85** | 0.2675 | 12.84 |
| | 14 | 2.3754 | 32.92 | 2.2308 | 33.98 | **2.1954** | 31.52 | 2.3009 | **29.50** | | 14 | 0.3253 | 14.82 | 0.3084 | 13.80 | **0.2926** | **13.39** | 0.2995 | 13.71 |
| | 21 | 2.7963 | 35.23 | **2.6500** | **32.70** | 2.7759 | 43.23 | 2.8562 | 34.70 | | 21 | 0.3475 | 16.27 | **0.3179** | **14.24** | 0.3273 | 15.35 | 0.3534 | 16.12 |
| | 28 | 3.1379 | 44.84 | **2.7228** | **40.44** | 2.9543 | 42.16 | 3.0996 | 41.44 | | 28 | 0.3769 | 16.89 | **0.3532** | **15.74** | 0.3550 | 15.96 | 0.3679 | 16.73 |

concatenated output of two encoder branches, we take $\mathbf{Z_c}$ directly from T-DAB as the output $\mathbf{Z}_j$ of the $j$-th encoder layer.

$\neg$ T-DAB: Denotes that we train a DEFORMTIME model without the T-DAB encoder branch (see section 3.4). We take $\mathbf{Z_c}$ directly from V-DAB as the output $\mathbf{Z}_j$ of the $j$-th encoder layer.

$\neg$ $\mathbf{P_{v,t}}$: Denotes that we do not use the relative positional biases (both $\mathbf{P_v}$ and $\mathbf{P_t}$) when obtaining the value embedding for both the V-DAB and T-DAB modules.

$\neg$ NAE: Denotes that we train a DEFORMTIME model without the NAE module. For an input $\mathbf{Z} \in \mathbb{R}^{L \times (C+1)}$, we simply use a fully connected layer to embed the $C+1$ variables into a hidden dimension of size $d$ and obtain $\mathbf{E} \in \mathbb{R}^{L \times d}$.

$\neg$ $\mathbf{P_n}$: Denotes that we do not use the fixed position embedding $\mathbf{P_n}$ from the embedding procedure, i.e. instead of Equation 2, we use $\mathbf{Z_e} = \mathtt{LN}(\mathbf{E})$.

### E.3   Detailed ILI forecasting results

Complete results (all test periods, models, and forecasting horizons) in the ILI rate forecasting tasks for England (ILI-ENG), US Region 2 (ILI-US2), and US Region 9 (ILI-US9) are presented in Tables S1, S2, and S3, respectively. In addition to MAE and sMAPE, we also show the linear correlation (denoted by $\rho$) between estimates and the target variable throughout each test period.

### E.4   The effect of variable grouping in the NAE module

We further explored the effect of the grouping parameter $G$ in DEFORMTIME, which determines both the number of groups in NAE and the number of heads in the multi-head attention of T-DAB. Experiments are conducted using the ILI forecasting task for England (ILI-ENG) and US Region 9 (ILI-US9). Hyperparameters are re-tuned for each value of $G \in \{2, 4, 8, 16\}$.

Results are enumerated in Table S4. It can be argued that more groups tend to benefit performance for shorter forecasting horizons, whereas a more moderate level of grouping is required for the optimal performance as the forecasting horizon increases. Notably, extensive ($G = 16$) or overly small ($G = 2$) groupings result in performance degradation. This observation could be further interpreted in conjunction with the results presented in Table 3, where we see that DEFORMTIME performs better in shorter-term forecasting tasks when it uses more variables ($C$). Presumably, the model benefits from an adaptive $G$, where $G$ is proportional to the number of input variables $C$. Nevertheless, to reduce the complexity of hyperparameter optimisation in our experiments we have chosen to use a constant value, setting $G = 4$.

We note that the observed influence of grouping may pose a potential limitation of DEFORMTIME, as it affects how dependencies are captured within the latent variables. Specifically, smaller grouping numbers allow the dependencies to be captured among variables holding less linear correlations and vice versa. However, there likely exists a trade-off between allowing more variables to be able to be captured by V-DAB and the model being overfitted as we have empirically shown in Tables 3 and S4, where both removing the NAE and using an overly small number of groups lead to an inferior performance.

Table S5: Seed control (5 seeds) for DEFORMTIME using the ILI-ENG data set across all forecasting horizons ($H$). $\mu$ and $\sigma$ denote the mean and standard deviation of the 5 obtained MAEs per test period. The results in the main paper were obtained for seed '42'.

| $H$ | seed | 2015/16 | 2016/17 | 2017/18 | 2018/19 | Average |
|---|---|---|---|---|---|---|
| 7 | 42 | 1.379 | 1.308 | 2.631 | 1.248 | 1.642 |
| | 10 | 1.416 | 1.063 | 2.266 | 1.360 | 1.526 |
| | 111 | 1.386 | 0.886 | 2.315 | 1.690 | 1.569 |
| | 1111 | 1.476 | 1.404 | 1.986 | 1.533 | 1.600 |
| | 1234 | 1.443 | 1.109 | 2.197 | 1.226 | 1.494 |
| | $\mu$ ($\sigma$) | 1.42 (0.04) | 1.15 (0.18) | 2.28 (0.21) | 1.41 (0.18) | 1.57 (0.06) |
| 14 | 42 | 2.056 | 1.318 | 3.763 | 1.786 | 2.231 |
| | 10 | 1.696 | 1.514 | 3.439 | 1.712 | 2.090 |
| | 111 | 1.861 | 1.297 | 3.556 | 1.842 | 2.139 |
| | 1111 | 1.621 | 1.495 | 3.327 | 1.396 | 1.960 |
| | 1234 | 1.837 | 1.137 | 3.879 | 1.391 | 2.061 |
| | $\mu$ ($\sigma$) | 1.81 (0.15) | 1.35 (0.14) | 3.59 (0.20) | 1.63 (0.19) | 2.10 (0.10) |

| $H$ | seed | 2015/16 | 2016/17 | 2017/18 | 2018/19 | Average |
|---|---|---|---|---|---|---|
| 21 | 42 | 2.170 | 1.898 | 4.898 | 1.633 | 2.650 |
| | 10 | 2.079 | 1.507 | 4.871 | 1.874 | 2.583 |
| | 111 | 2.354 | 1.825 | 4.668 | 1.944 | 2.697 |
| | 1111 | 2.236 | 1.943 | 4.158 | 1.625 | 2.491 |
| | 1234 | 2.147 | 1.595 | 4.406 | 1.842 | 2.498 |
| | $\mu$ ($\sigma$) | 2.20 (0.09) | 1.75 (0.17) | 4.60 (0.28) | 1.78 (0.13) | 2.58 (0.09) |
| 28 | 42 | 2.654 | 1.810 | 4.832 | 1.595 | 2.723 |
| | 10 | 2.486 | 1.692 | 5.194 | 1.807 | 2.795 |
| | 111 | 2.655 | 1.825 | 4.887 | 1.907 | 2.819 |
| | 1111 | 2.251 | 1.799 | 4.529 | 2.082 | 2.665 |
| | 1234 | 2.692 | 1.808 | 4.939 | 1.613 | 2.763 |
| | $\mu$ ($\sigma$) | 2.55 (0.17) | 1.79 (0.05) | 4.88 (0.21) | 1.80 (0.18) | 2.75 (0.06) |

### E.5 Computational complexity and efficiency of DeformTime – Additional information

The total number of operations (based on multiplications) for DEFORMTIME is given by

$$
\underbrace{(C+1)\,dL+d}_{\text{NAE}} + \underbrace{2d^2L}_{\text{Decoder}} + \\
\underbrace{2\left\{\underbrace{n\left[(k^2+1)\ell d+3\ell d^2+3\ell^2 d+n\ell dL\right]}_{\text{V-DAB}}+\underbrace{r\left[(k+1)\kappa d+3\kappa d^2+3\kappa^2 d\right]}_{\text{T-DAB}}+2d+6d^2L\right\}}_{\text{Encoder}}. \tag{S2}
$$

The main components that affect the order of operations are the following:

$$
\text{V-DAB:}\quad \mathcal{V}=n\left[\left(k^2+1\right)\ell d+3\ell d^2+3\ell^2 d+n\ell dL\right]\ \text{operations} \tag{S3}
$$
$$
L\approx n\ell \implies \mathcal{O}\left(d^2L+dL^2\right)
$$

$$
\text{T-DAB:}\quad \mathcal{T}=r\left[(k+1)\kappa d+3\kappa d^2+3\kappa^2 d\right]\ \text{operations} \tag{S4}
$$
$$
L=\kappa r \implies \mathcal{O}\left(d^2L\right)
$$

$$
\text{Encoder:}\quad \mathcal{E}=2\left(\mathcal{V}+\mathcal{T}+2d+6d^2L\right)\ \text{operations} \tag{S5}
$$
$$
\implies \mathcal{O}\left(d^2L+dL^2\right)
$$

$$
\text{Decoder:}\quad \mathcal{G}=2d^2L\ \text{operations} \tag{S6}
$$
$$
\implies \mathcal{O}\left(d^2L\right).
$$

Hence, we conclude that the order of operations for DEFORMTIME is $\mathcal{O}\left(d^2L+dL^2\right)$, where $L$ denotes the length (time steps) of the look-back window, and $d$ is the size of the hidden layers used throughout our method.

### E.6 Seed robustness for DeformTime

We have examined the robustness of DEFORMTIME across different seeds with hyperparameter tuning by re-running the experiments on the ILI-ENG data set across all forecasting horizons and test sets. The results are enumerated in Table S5. The seed used for the results presented in the main paper ('42') provided a performance that did not deviate significantly compared to other seeds. Hence, we conclude that DEFORMTIME is robust to random seed initialisation. We note that compared to the average performance across the explored seeds, seed '42' provides a rather conservative estimate.

Table S6: Forecasting accuracy results (MAE) using the entire output sequence (as opposed to using the last prediction only at the target forecasting horizon) across all tasks and methods. $H$ denotes the forecasting horizon. For the ILI tasks, errors are averaged across all 4 test seasons for each region.

| Task | $H$ | DEFORMTIME | ModernTCN | CycleNet | TimeXer | PatchTST | iTransformer | TimeMixer | Crossformer | LightTS | DLinear |
|------|-----|-----------|-----------|----------|---------|----------|--------------|-----------|-------------|---------|---------|
| ETTh1 | 96 | 0.1840 | 0.1774 | **0.1746** | 0.1807 | 0.1757 | 0.1825 | 0.1834 | 0.1989 | 0.2089 | 0.2045 |
|  | 192 | **0.1988** | 0.2013 | 0.1990 | 0.2043 | 0.1999 | 0.2051 | 0.2049 | 0.2149 | 0.2267 | 0.2834 |
|  | 336 | **0.2024** | 0.2123 | 0.2189 | 0.2226 | 0.2248 | 0.2322 | 0.2310 | 0.2577 | 0.2487 | 0.3877 |
|  | 720 | 0.2390 | **0.2288** | 0.2367 | 0.2292 | 0.2500 | 0.2449 | 0.2463 | 0.3000 | 0.4450 | 0.5007 |
| ETTh2 | 96 | 0.2924 | 0.2790 | 0.2849 | 0.2797 | **0.2765** | 0.2991 | 0.2963 | 0.3254 | 0.3125 | 0.2840 |
|  | 192 | **0.3010** | 0.3183 | 0.3223 | 0.3331 | 0.3174 | 0.3463 | 0.3428 | 0.3534 | 0.3496 | 0.3313 |
|  | 336 | **0.3172** | 0.3350 | 0.3536 | 0.3769 | 0.3417 | 0.3857 | 0.3758 | 0.3891 | 0.3939 | 0.4056 |
|  | 720 | **0.3577** | 0.4259 | 0.3976 | 0.4069 | 0.3887 | 0.3945 | 0.3940 | 0.4248 | 0.4339 | 0.5391 |
| Weather | 96 | 0.0226 | 0.0235 | 0.0228 | 0.0270 | **0.0204** | 0.0218 | 0.0272 | 0.0247 | 0.0263 | 0.0206 |
|  | 192 | **0.0237** | 0.0265 | 0.0257 | 0.0294 | 0.0239 | 0.0253 | 0.0309 | 0.0271 | 0.0292 | 0.0239 |
|  | 336 | **0.0258** | 0.0296 | 0.0279 | 0.0312 | 0.0259 | 0.0269 | 0.0325 | 0.0311 | 0.0294 | 0.0261 |
|  | 720 | 0.0308 | 0.0365 | 0.0333 | 0.0355 | 0.0316 | 0.0312 | 0.0369 | 0.0343 | 0.0339 | **0.0307** |
| ILI-ENG | 7 | **1.2802** | 1.7351 | 2.0997 | 1.7086 | 1.9402 | 1.7746 | 1.7745 | 1.3243 | 1.9713 | 2.1892 |
|  | 14 | **1.6521** | 2.2656 | 2.6170 | 1.9703 | 2.4857 | 2.3660 | 2.2577 | 1.9326 | 2.3676 | 2.7452 |
|  | 21 | **2.3246** | 2.3331 | 3.2527 | 2.8852 | 2.8458 | 3.0904 | 2.9285 | 2.8733 | 2.7742 | 3.2983 |
|  | 28 | 2.7829 | **2.5948** | 3.6091 | 2.7894 | 3.2161 | 3.4784 | 3.2425 | 3.0740 | 3.2559 | 3.6805 |
| ILI-US2 | 7 | **0.4142** | 0.4523 | 0.6974 | 0.4931 | 0.6400 | 0.5455 | 0.5156 | 0.4444 | 0.4323 | 0.7214 |
|  | 14 | **0.4544** | 0.5501 | 0.7601 | 0.5402 | 0.7302 | 0.6357 | 0.5496 | 0.4695 | 0.4723 | 0.7899 |
|  | 21 | **0.5374** | 0.5576 | 0.8130 | 0.6190 | 0.7489 | 0.6554 | 0.5966 | 0.5720 | 0.6432 | 0.8642 |
|  | 28 | **0.5554** | 0.5892 | 0.8729 | 0.6808 | 0.8450 | 0.7370 | 0.6688 | 0.5842 | 0.6886 | 0.8920 |
| ILI-US9 | 7 | **0.2614** | 0.2945 | 0.4126 | 0.3435 | 0.3882 | 0.3725 | 0.3322 | 0.3222 | 0.3078 | 0.4348 |
|  | 14 | **0.3040** | 0.3398 | 0.4584 | 0.3715 | 0.4334 | 0.4049 | 0.3590 | 0.3254 | 0.4322 | 0.4780 |
|  | 21 | **0.3187** | 0.3558 | 0.5224 | 0.4430 | 0.4664 | 0.4504 | 0.3714 | 0.3285 | 0.4385 | 0.5315 |
|  | 28 | 0.3913 | **0.3786** | 0.5727 | 0.4774 | 0.5002 | 0.5535 | 0.4236 | 0.4018 | 0.5665 | 0.5622 |

## E.7 A brief note about the performance and evaluation of DLinear

Although DLinear achieved competitive results in prior work (Zeng et al., 2023), the performance drop in our paper originates from the fact that we evaluate models based on their estimate for $y_{t+H}$, i.e. the target variable's value at time step $t+H$ (see a detailed description of the forecasting task in section 2). In fact, for most tasks this is the target forecast (represented by the corresponding forecasting horizon $H$) and that is what forecasting accuracy should be measured on. Contrary to common sense, DLinear (as well as other works, e.g. Wang et al. (2024b) or Liu et al. (2024)) was evaluated on the entire time series (entire sequence of predictions), i.e. from $y_{t+1}$ to $y_{t+H}$. Oddly, this evaluation was conducted in a uniform way, i.e. the error for each time step incurred the same penalty. That may be relevant for some tasks, but it is not relevant to tasks where forecasting $H$ time steps ahead really necessitates obtaining an accurate forecast $H$ time steps ahead (and that should be the case for most, if not all, forecasting tasks). The reason behind this is that forecasting errors tend to increase as the forecasting horizon extends and the forecasting task becomes harder (Hyndman & Koehler, 2006). Models may perform differently at short versus long horizons. Averaging the error across all the outputs may favour forecasters that are very accurate early on, but very inaccurate closer to and at the target forecasting horizon. DLinear was obviously more accurate in early time steps (lower degree of difficulty), but very inaccurate in later ones (greater degree of difficulty). Hence, in our experiments, DLinear displayed the worst forecasting performance. The following section (E.8) offers additional insights.

## E.8 Evaluating forecasts across the entire output time series

For a more comprehensive assessment of model performance, we provide forecasting accuracy results where we consider the entire output sequence (the error is averaged across the entire output sequence). Results are enumerated in Table S6. DEFORMTIME outperforms baselines on 17 out of 24 tasks, with an averaged MAE reduction by 1.7%. When compared to ModernTCN, the best-performing baseline, DEFORMTIME reduces the MAE by 7.2% on average across all tasks.

For the ETT and weather benchmark datasets, the best-performing baseline is PatchTST; DEFORMTIME reduces PatchTST's MAE by 1.5% on average. If we compare that to obtaining MAE based on the forecasts at the target forecasting horizon only (Table 1), we can see a major discrepancy: DEFORMTIME reduces PatchTST MAE by 9.8% (on the ETT and weather forecast tasks; reduction is greater

for the ILI forecasting task) when forecasting accuracy is evaluated based on the actual target forecast. Going back to using the entire output time series, for the ILI tasks, DEFORMTIME reduces the MAE by 3.4% on average. For these tasks, Crossformer achieves the best MAE performance amongst baselines and DEFORMTIME reduces that by 8.2% on average.

Notably, while PatchTST outperforms other baselines when evaluated over the entire sequence in the ETT and weather data sets, CycleNet was the best-performing baseline in our main assessment (Table 1). Hence, by this observation alone, it is evident that while CycleNet is producing more accurate forecasts at the target horizon compared to PatchTST, it yields an inferior average MAE when considering the entire series of forecast outputs from time step $t+1$ to time step $t+H$. A similar ranking difference of baseline performance is also observed in the ILI tasks, where ModernTCN was the most competitive baseline when evaluating the results over the target horizon (DEFORMTIME reduces ModernTCN MAE by 11.2% on the ILI tasks as enumerated in Table 1). The overall changes in the ranking of baselines consolidate our argument in section E.7 that convoluting the error at the target forecasting horizon with errors in time steps prior to that can produce a distorted picture of the actual forecasting capacity of a model.

### E.9  Should forecasting models mix variables in the embedding space?

The paper that presented the PatchTST method (Nie et al., 2023) provided some empirical evidence for the potential benefits of not modelling inter-variable dependencies. Contrary to that, our results demonstrate that this may not always hold for better-performing models as well as different time series forecasting tasks. Based on our ILI rate forecasting experiments, we deduce that models that mix variables in the embedding space while preserving the temporal structure (ModernTCN, Crossformer, LightTS, TimeMixer, and DEFORMTIME) outperform the rest. Hence, and perhaps as expected, modelling inter-variable relationships should still be considered a viable (and arguably essential) approach in time series forecasting.

## F  Societal impact

As with any scientific development, our work may have positive but also negative societal impact. From a positive perspective, forecasting models can be used to better predict outcomes in various domains and improve quality of life. Contrary to that, forecasting models could also be exploited by malicious actors in various forms of malpractice. We do think this should be common knowledge. We do also note that there is a thin line between safeguarding AI malpractice and significantly restricting scientific progress (and hence, quality of life improvements). This is beyond the scope of our work and is obviously a broader topic of discussion.

We would like to note that outcomes of this work (e.g. an improved variant of DEFORMTIME) may be used in infectious disease monitoring systems by interested stakeholders (e.g. public health organisations such as CDC, ECDC, UKHSA etc.). These, of course, do not constitute malicious actors. However, a potential risk may arise when (if) a forecasting model provides inaccurate insights. Nevertheless, this is a shared risk among various other disease models (including established mechanistic models). To mitigate this risk, public health agencies use multiple endpoints to determine their course of action (e.g. sentinel and syndromic surveillance, laboratory tests, rapid tests, and vetted epidemiological models). Hence, any decision is based on a collection of different predictors. Nonetheless, prior to the adoption of any forecasting model within public policy, a more thorough and focused (to the said application domain) evaluation of the forecasting method is required.

## G  Supplementary ILI rate forecasting figures

Here we present a series of figures for the ILI rate forecasting task. Each figure shows the estimates from all models for a certain forecasting horizon and location.

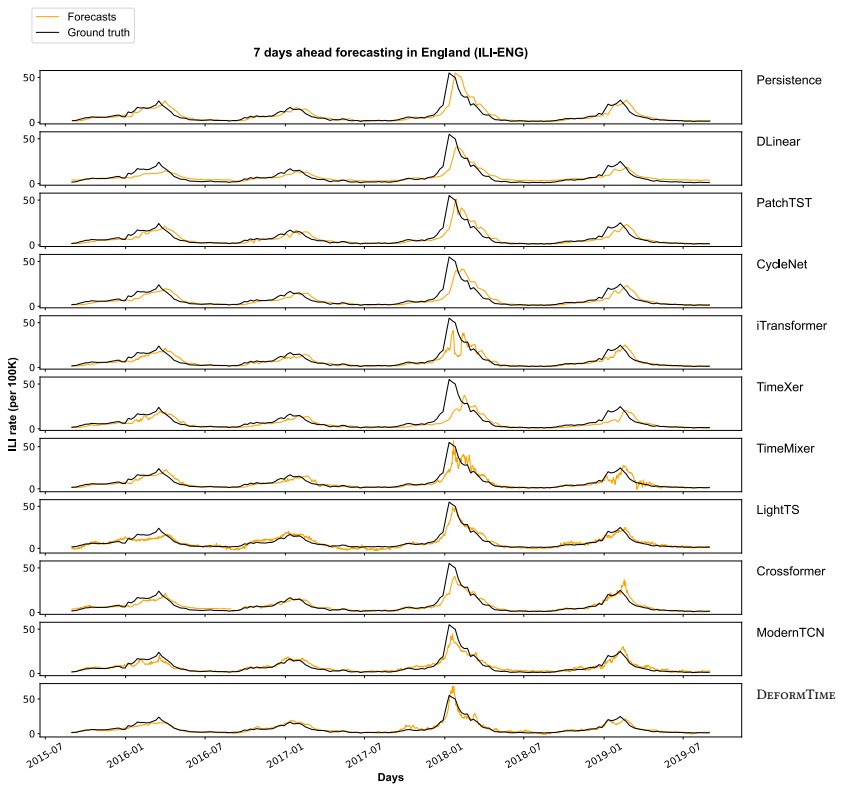

Figure S2: 7 days ahead forecasts for all influenza seasons and models for England (ILI-ENG).

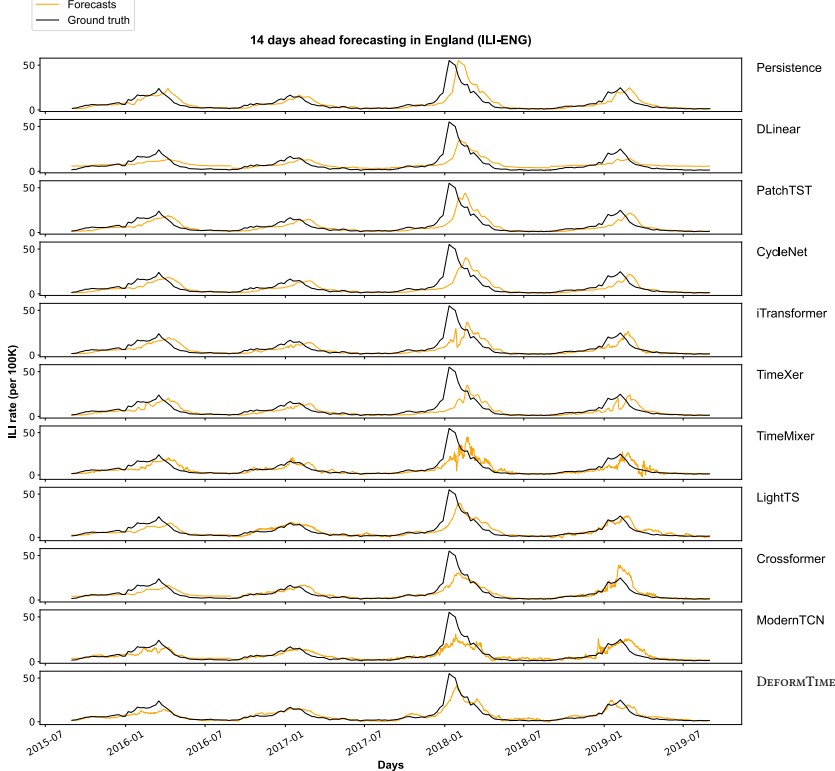

Figure S3: 14 days ahead forecasts for all influenza seasons and models for England (ILI-ENG).

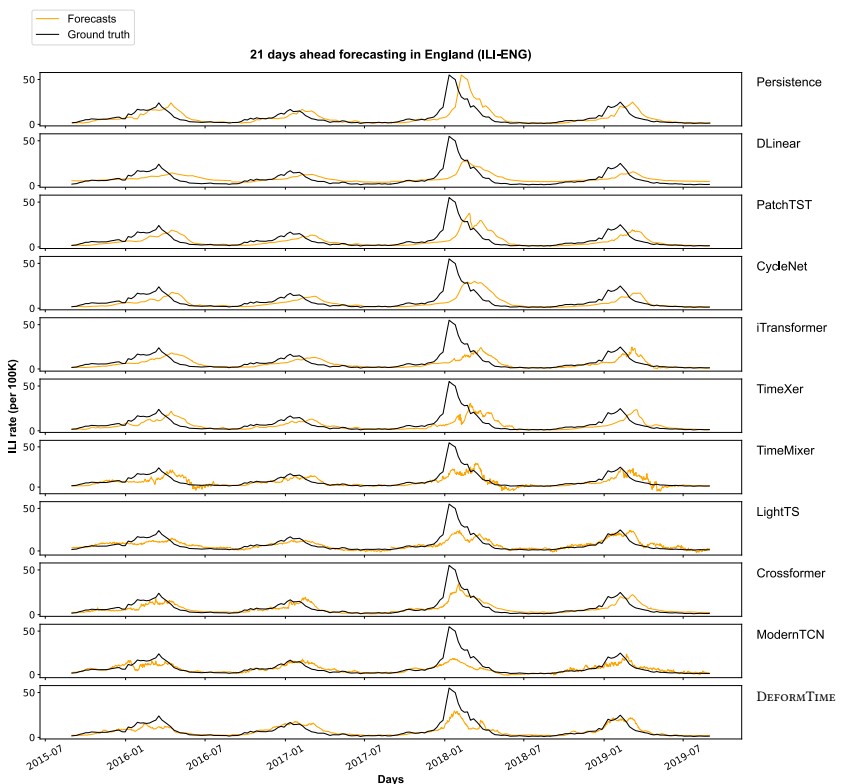

Figure S4: 21 days ahead forecasts for all influenza seasons and models for England (ILI-ENG).

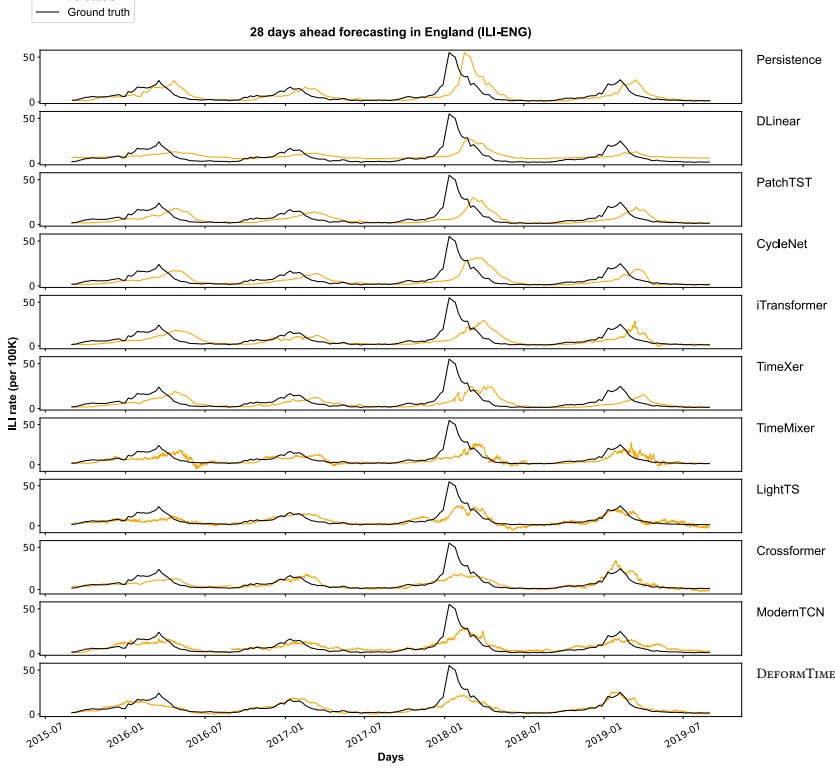

Figure S5: 28 days ahead forecasts for all influenza seasons and models for England (ILI-ENG).

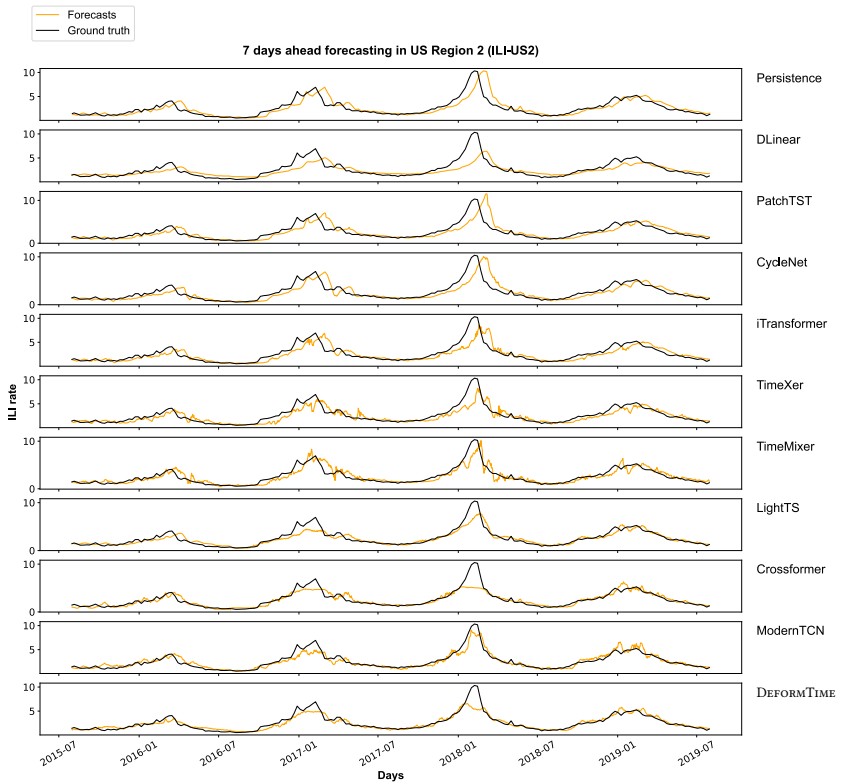

Figure S6: 7 days ahead forecasts for all influenza seasons and models for US Region 2 (ILI-US2).

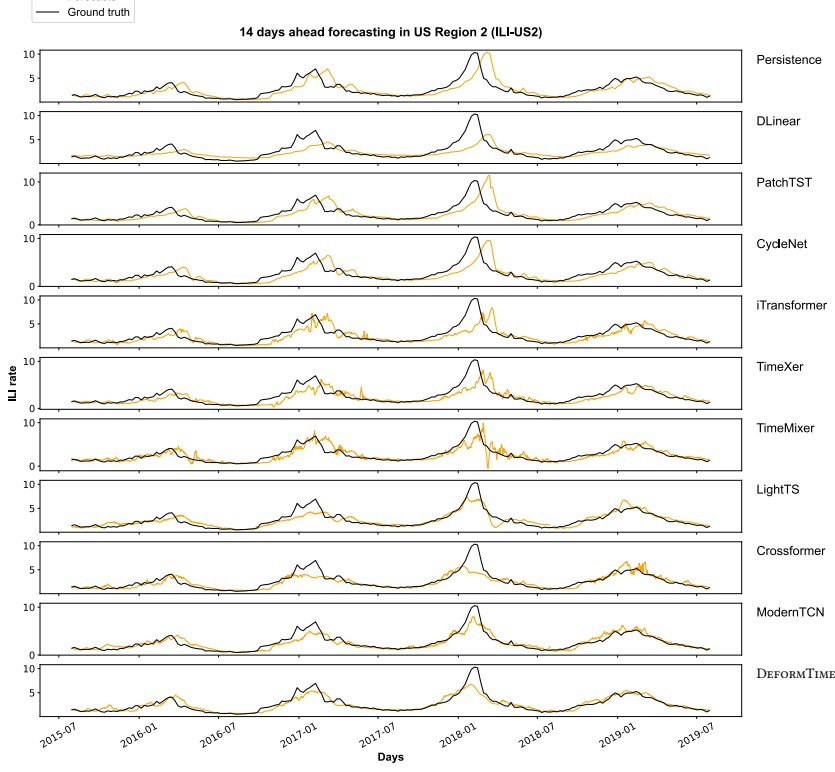

Figure S7: 14 days ahead forecasts for all influenza seasons and models for US Region 2 (ILI-US2).

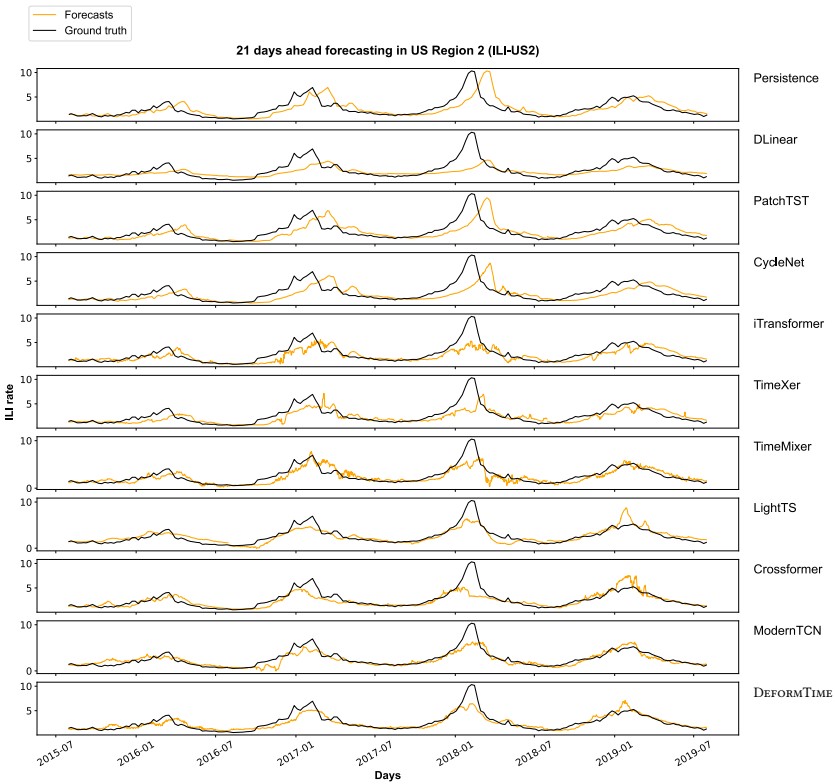

Figure S8: 21 days ahead forecasts for all influenza seasons and models for US Region 2 (ILI-US2).

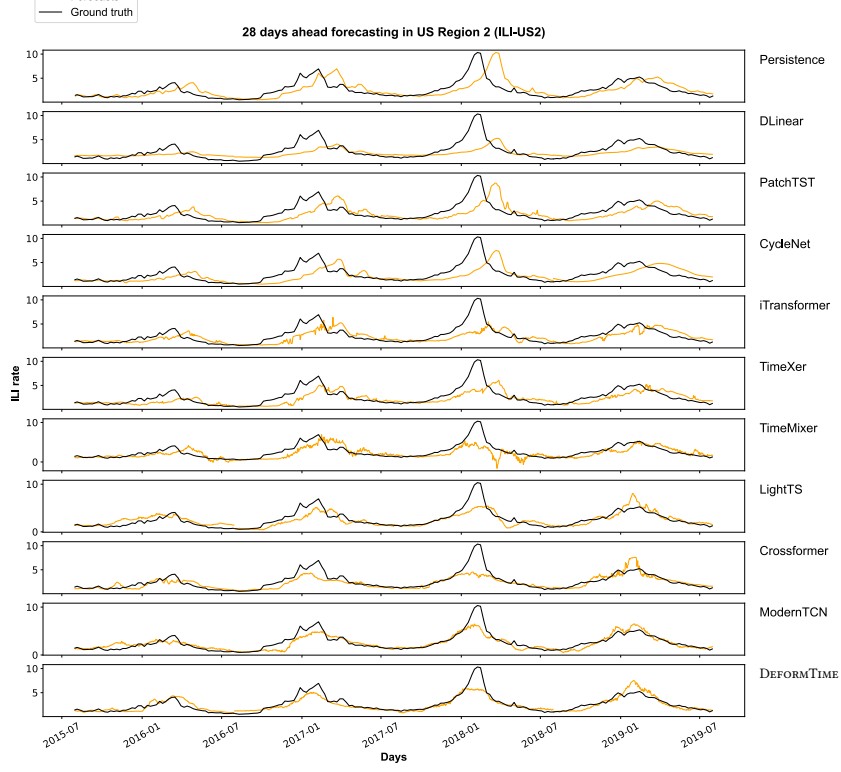

Figure S9: 28 days ahead forecasts for all influenza seasons and models for US Region 2 (ILI-US2).

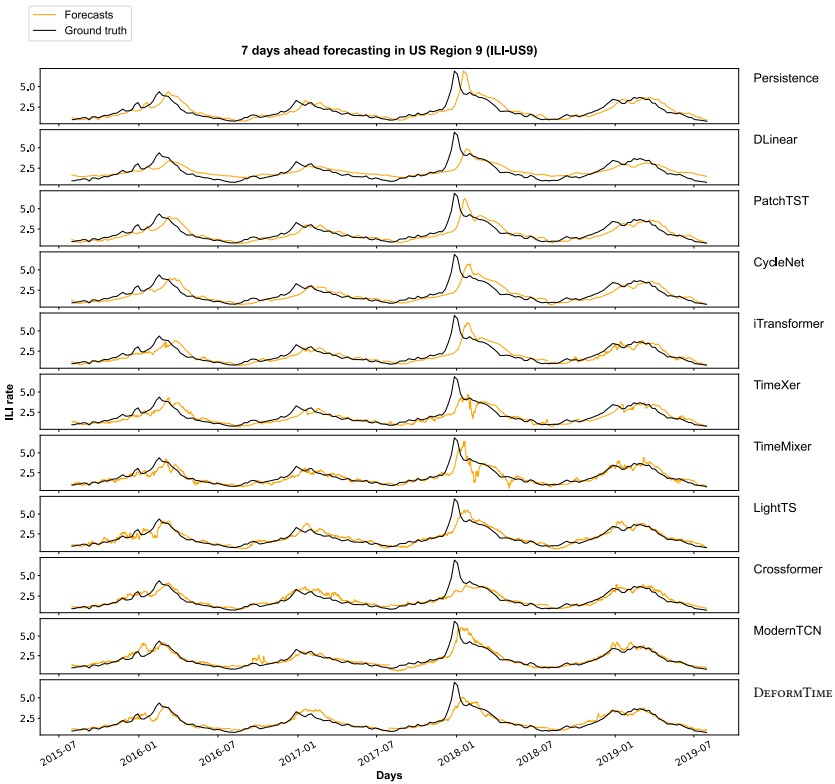

Figure S10: 7 days ahead forecasts for all influenza seasons and models for US Region 9 (ILI-US9).

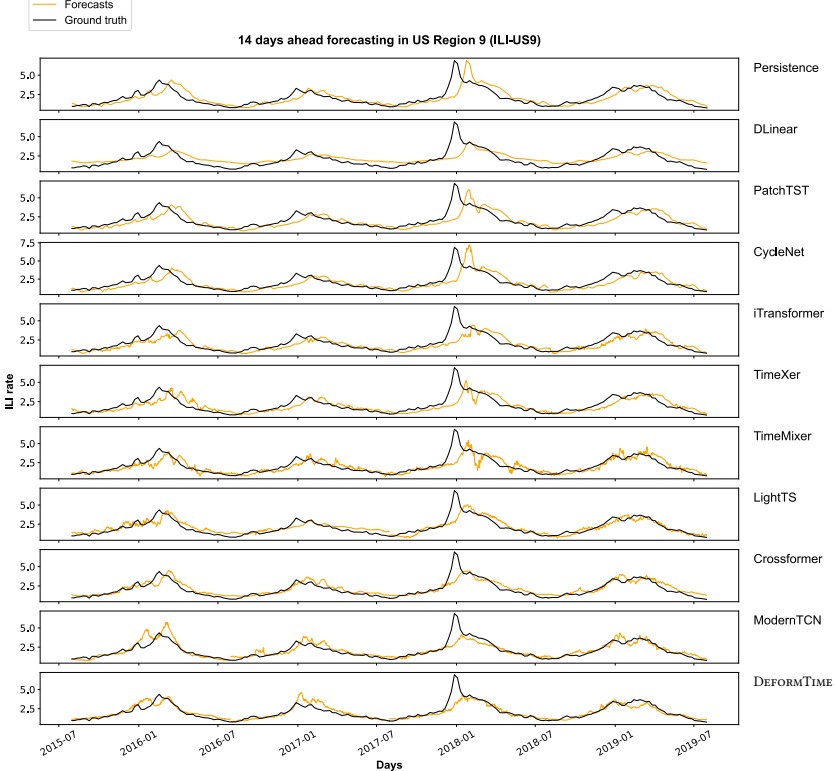

Figure S11: 14 days ahead forecasts for all influenza seasons and models for US Region 9 (ILI-US9).

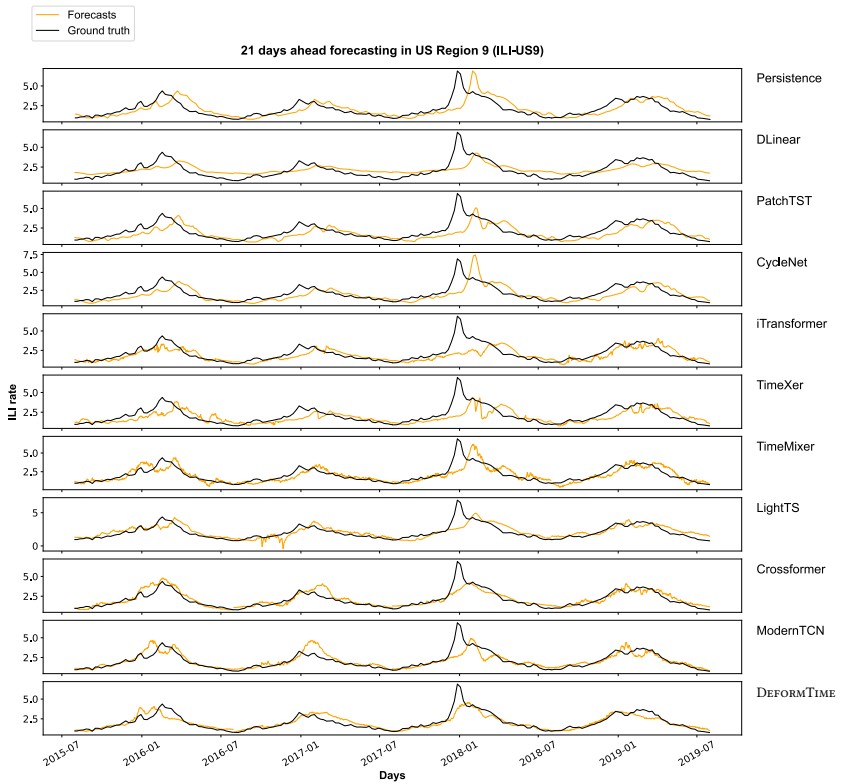

Figure S12: 21 days ahead forecasts for all influenza seasons and models for US Region 9 (ILI-US9).

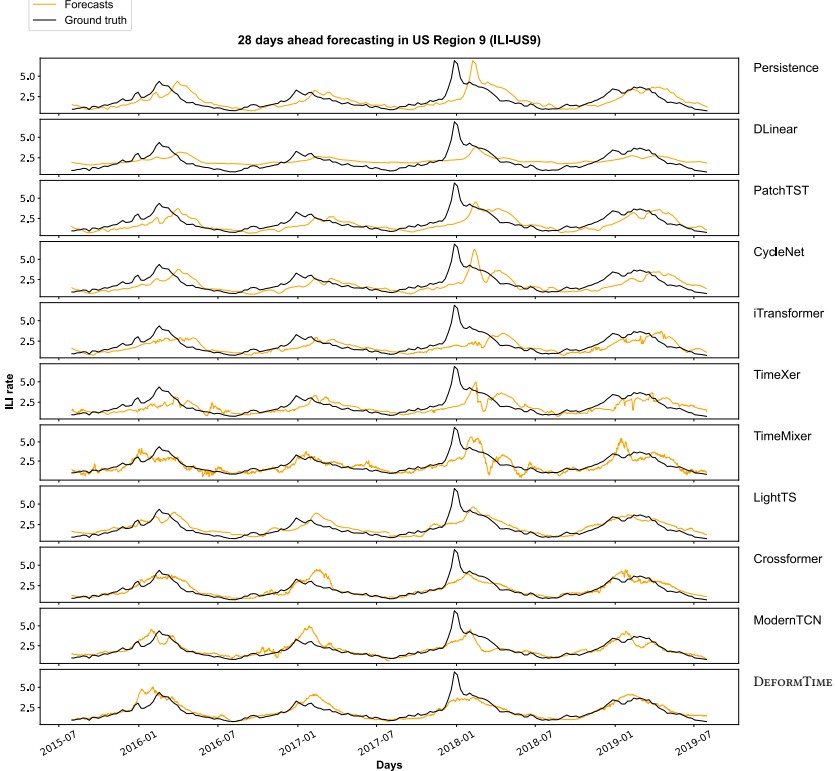

Figure S13: 28 days ahead forecasts for all influenza seasons and models for US Region 9 (ILI-US9).

