# OpenReview forum: "DeformTime: capturing variable dependencies with deformable attention for time series forecasting"
_TMLR — Accepted by TMLR_

### Review · Reviewer_rgws · 2025-01-29

**Summary Of Contributions:**

This paper proposes a new multi-variable time series forecasting method called DeformTime. It uses Deform Attention Blocks on both variable and temporal dimensions. On both existing and proposed new benchmarks, DeformTime demonstrates lower errors than other baselines, as well as better memory usage. Ablation study also highlights the important of the proposed Deform Attention Blocks.

**Audience:**

Yes

**Broader Impact Concerns:**

N/A.

**Claims And Evidence:**

No

**Requested Changes:**

Section 3.2, are the grouping of variables just sequentially taking the d/G variables as one group after the reordering? What motivates such grouping? Why does it make sense to use them?

I didn’t get the Section 3.3. For a patch $Z_p$ of size (l, d), why adding a shift $\Delta P$ to a position p = (i, j) helps to learn the correlation between different variables? What does $\Delta P$ mean? What does $Z_d$ mean? In this section, I think it lacks an overall introduction on how the authors are going to leverage multi-variable information for forecasting and why they believe that’s the right way. The current writing is simply a list of steps without explaining the motivation and the why, which makes it hard to follow. Section 3.4 has the same problem.

Why just compute errors on the last time step in the prediction window? Is there any particular reason? Unless there are strong reasons, I suggest the authors use a more standard way to compute error, that is over all the time steps in the prediction window. Then we can compare numbers to see if they match the baselines in their original papers so that we know the baselines are implemented and used properly.

Minor: Why does the delay need to be explicitly modeled? Can it be part of the preprocessing step?

Minor: How do the authors find so many exogenous variables in Section 4.4? By different search queries?

**Strengths And Weaknesses:**

The strength of the paper is the completeness of a method paper in it's empirical components: from basic baseline comparisons to memory and time analysis, as well as ablation study.

The biggest weakness is lack of motivation and clarity for methods, the "why" question. The current method section is a list of statements without explaining why the authors choose to do the things that way. Hence, it is hard to follow. See more in the Requested Changes Section.

---

> ### Author Response · Authors · 2025-02-02
>
> We thank the reviewer for acknowledging our experiments are comprehensive.
>
> >**Question 1**
> >In section 3.2, are the grouping of variables just sequentially taking the d/G variables as one group after the reordering? What motivates such grouping? Why does it make sense to use them?
>
> The reviewer's understanding of grouping is correct. The motivation for grouping is to preserve neighbouring variables in the latent space as we mentioned in section 3.2, which directly benefits the V-DAB module to capture correlated features within the receptive field (as explained in more detail in section 4.3). As discussed in sections 3.2 and 3.3, the receptive field of V-DAB is determined by a CNN with a sampling offset. By grouping variables in the NAE module, we ensure that variables within the same group exhibit stronger local correlations for the model to capture. Our ablation analysis highlights that removing the NAE grouping results in inferior performance (see Table 2). Further analysis of how the number of variable groups affects performance is provided in Table D4. We will further clarify this in our revised manuscript.
>
> >**Question 2**
> >I didn't get the Section 3.3. For a patch $\mathbf{Z}_p$ of size $(\ell, d)$, why adding a shift $\Delta p$ to a position $p = (i, j)$ helps to learn the correlation between different variables? What does $\Delta p$ mean? What does $\mathbf{Z}_p$ mean? In this section, I think it lacks an overall introduction on how the authors are going to leverage multi-variable information for forecasting and why they believe that's the right way. The current writing is simply a list of steps without explaining the motivation and the why, which makes it hard to follow. Section 3.4 has the same problem.
>
> Thank you for identifying this. We have attempted to clarify in more detail in our response to all reviewers at the top of this page (see section titled **Deformation using V-DAB and T-DAB**). We will use this information to improve the clarity in our revised manuscript.
>
> >**Question 3**
> >Why just compute errors on the last time step in the prediction window? Is there any particular reason? Unless there are strong reasons, I suggest the authors use a more standard way to compute error, that is over all the time steps in the prediction window. Then we can compare numbers to see if they match the baselines in their original papers so that we know the baselines are implemented and used properly.
>
> We thank the reviewer for raising this. We provide a thorough response on this in the preamble of our response at the top of this page (addressed to all reviewers). We could include this additional analysis in our revised manuscript for completeness.
>
> >**Question 4**
> >Minor: Why does the delay need to be explicitly modelled? Can it be part of the preprocessing step?
>
> This is only relevant to the practicalities of the ILI rate forecasting task. Assume that at time $t$ we want to forecast the ILI rate at time $t+H$. The web search query frequencies for time $t$ are available. However, the ILI rate at time $t$ is not yet available. At time $t$, we only know the ILI rate for time $t-\delta$, where $\delta$ can be 1 or 2 weeks depending on different health surveillance systems. On other forecasting tasks we simply set $\delta$ to $0$. In that sense, although this can be dealt with as a data preprocessing task, we do make a more specific note of it. We clarify this further in sections 2, and A.2, noting that this is a common setup in works that model ILI rates [1-3].
>
> **References**
>
> 1. Aiken, E. L., et al. (2020). Real-time estimation of disease activity in emerging outbreaks using internet search information. PLOS computational biology.
> 2. Caldwell, W. K., et al. (2020). Surveilling influenza incidence with Centers for Disease Control and Prevention web traffic data: demonstration using a novel dataset. JMIR.
> 3. Yang, S., et al. (2015). Accurate estimation of influenza epidemics using Google search data via ARGO. PNAS.
>
> >**Question 5**
> >Minor: How do the authors find so many exogenous variables in Section 4.4? By different search queries?
>
> Yes, for the ILI forecasting task, the frequency time series of different search queries are considered as different input variables. We elaborate on how the web search data set is formed in section A.2. We use $22{,}071$ health-related search queries and obtain their frequency time series (daily) using an API offered by Google. In each training / test evaluation for the ILI forecasting task, the number of exogenous variables may differ because we only use the top-$n$ most correlated search queries with the target variable (ILI rates) in the $5$ last flu seasons (in the training data). In the ablation study presented in section 4.4, the number of search queries (in that section denoted by $C$) is determined by simply using the top-$C$ correlated ones. Table 3 in section 4.4 (of the main paper) enlists the amount of exogenous variables used.

---

> ### Comment · Reviewer_rgws · 2025-02-22
> **Reply to authors**
>
> I would like to first thank the authors for their response. The motivations and experiments settings become clearer to me.
>
> I still have one question regarding to the new experiments measured on the entire output sequences. In the Table S6, it seems the MAE on the Weather and ETTh1 are considerably lower than the Table 3 from their [original results](https://github.com/yuqinie98/PatchTST). For example, the MAE on Weather dataset are around 0.02 in this paper while the MAE are at least 0.15 in the original PatchTST paper. Why is that the case? Can the authors reproduce the baseline results?

---

> > ### Author Response · Authors · 2025-02-23
> >
> > >I would like to first thank the authors for their response. The motivations and experiments settings become clearer to me.
> >
> > We thank the reviewer for acknowledging our clarifications.
> >
> > >I still have one question regarding to the new experiments measured on the entire output sequences. In the Table S6, it seems the MAE on the Weather and ETTh1 are considerably lower than the Table 3 from their original results. For example, the MAE on Weather dataset are around 0.02 in this paper while the MAE are at least 0.15 in the original PatchTST paper. Why is that the case? Can the authors reproduce the baseline results?
> >
> > We note that DeformTime is multivariable forecasting method: it uses multiple input variables to predict a single target variable. PatchTST and most baselines (except TimeXer) conduct multivariate forecasting, which has multiple variables as the output [1].
> >
> > The difference in error between Table S6 and the original PatchTST paper is because the PatchTST paper listed forecasting errors evaluated across all output variables while in our paper we only do this for the single target variable. We note that the results in Table S6 are directly comparable with the TimeXer paper as DeformTime and TimeXer are both focusing on multivariable forecasting tasks.
> >
> > To better alleviate any further concerns in our response, please see the Table below. We enumerate the reproduced results (from our experiments) of baseline models on multivariate forecasting. We show MAEs across the entire set of output variables as conducted in the original papers. The reproduced results are referred to using **R** (first number in table cells) while the reported results from the original papers are referred to as **O** (second number in table cells). It can be seen that all reproductions from us are within an acceptable range compared to the original results. Minor discrepancies may be due to updated GPU drivers and/or the use of a different GPU device.
> >
> > | **R / O**    | ModernTCN       | CycleNet       | PatchTST       | iTransformer   | TimeMixer      | DLinear        |
> > |--------------|----------------|----------------|----------------|----------------|----------------|----------------|
> > | ETTh1-96     | 0.3944/0.394    | 0.3935/0.396    | 0.4050/0.399    | 0.4046/0.405    | 0.4028/0.400    | 0.3953/0.399    |
> > | ETTh1-192    | 0.4142/0.413    | 0.4148/0.415    | 0.4211/0.421    | 0.4361/0.436    | 0.4385/0.421    | 0.4273/0.416    |
> > | ETTh1-336    | 0.4122/0.412    | 0.4309/0.430    | 0.4355/0.436    | 0.4622/0.458    | 0.4618/0.458    | 0.4621/0.443    |
> > | ETTh1-720    | 0.4612/0.461    | 0.4639/0.464    | 0.4659/0.466    | 0.4938/0.491    | 0.4771/0.482    | 0.4954/0.490    |
> > | ETTh2-96     | 0.3326/0.332    | 0.3417/0.341    | 0.3361/0.336    | 0.3499/0.349    | 0.3433/0.341    | 0.3441/0.353    |
> > | ETTh2-192    | 0.3733/0.374    | 0.3850/0.385    | 0.3791/0.379    | 0.3988/0.400    | 0.3912/0.392    | 0.4015/0.418    |
> > | ETTh2-336    | 0.3762/0.376    | 0.4138/0.413    | 0.3802/0.380    | 0.4322/0.432    | 0.4428/0.414    | 0.4731/0.465    |
> > | ETTh2-720    | 0.4321/0.433    | 0.4513/0.451    | 0.4215/0.422    | 0.4470/0.445    | 0.4485/0.434    | 0.5581/0.551    |
> > | Weather-96   | 0.2036/0.200    | 0.2213/0.221    | 0.1994/0.199    | 0.2150/0.214    | 0.2102/0.209    | 0.2356/0.237    |
> > | Weather-192  | 0.2462/0.245    | 0.2584/0.258    | 0.2423/0.243    | 0.2567/0.254    | 0.2521/0.250    | 0.2812/0.282    |
> > | Weather-336  | 0.2823/0.277    | 0.2926/0.293    | 0.2832/0.283    | 0.2994/0.296    | 0.2927/0.287    | 0.3177/0.319    |
> > | Weather-720  | 0.3353/0.334    | 0.3398/0.339    | 0.3349/0.335    | 0.3487/0.347    | 0.3444/0.341    | 0.3634/0.362    |
> >
> > *Table: Reproduced (**R**) vs. original (**O**) MAEs for the ETT and weather forecasting tasks. MAEs are reported for multi-task learning (multiple input and output variables) and across the entire output time series.*
> >
> > [1] Hidalgo, Bertha, and Melody Goodman (2013). Multivariate or multivariable regression?

---

> > > ### Comment · Reviewer_rgws · 2025-02-27
> > > **Single target variable**
> > >
> > > Thank you for your response. It seems that the baseline performance is indeed in the same range as the original paper. Good to have that confirmed.
> > >
> > > Do you have the results of evaluating all the outputs in the prediction window on the remaining three ILI datasets, as shown in Table 1? Even though the authors have a different opinion on what constitutes a better evaluation setting, I believe it is important to present the complete picture from both perspectives.
> > >
> > > Additionally, for a multivariate time series dataset (e.g., ETT, weather), a more thorough evaluation would involve treating each variable as a target variable while considering the others as exogenous variables. This is just a suggestion and is not a condition for acceptance.

---

> > > > ### Author Response · Authors · 2025-02-28
> > > >
> > > > >Do you have the results of evaluating all the outputs in the prediction window on the remaining three ILI datasets, as shown in Table 1? Even though the authors have a different opinion on what constitutes a better evaluation setting, I believe it is important to present the complete picture from both perspectives.
> > > >
> > > > We thank the reviewer for this insightful perspective to form a more comprehensive understanding of the model performance on the ILI tasks. We unfortunately did not maintain the full forecasting results for the ILI tasks (we kept only the predicted values at the target forecasting horizon time step). We are now re-running all these experiments with exactly the same hyperparameter tuning setup as in the paper for all regions and test seasons (these experiments need about a working week to complete). We will add the ILI results averaged over the forecasting horizons in our revised paper. We note, however, that given that:
> > > >
> > > > (a) DeformTime yields stronger performance improvements in the ILI tasks (compared to other benchmark tasks), and
> > > >
> > > > (b) we have already shown a comparison between MAEs for the entire sequence of output forecasts (in Table 1) for the benchmark tasks where DeformTime is still the best-performing model,
> > > >
> > > > we expect that these additional results will re-affirm what is already shown in our currently provided experiments. We nevertheless agree that these additional results will make our argument more complete, but they may not be essential to make the argument (as we have already provided evidence for it). We are going to add them to the supplementary material as soon as they become available (and definitely in the camera-ready version of our paper).
> > > >
> > > >
> > > > >Additionally, for a multivariate time series dataset (e.g., ETT, weather), a more thorough evaluation would involve treating each variable as a target variable while considering the others as exogenous variables. This is just a suggestion and is not a condition for acceptance.
> > > >
> > > > We thank the reviewer for their constructive suggestion and we will consider including other climate indicators as the forecasting target in our future work.

---

> ### Author Response · Authors · 2025-03-07
>
> Results of the ILI forecasting tasks when averaging MAE across the entire output sequence (as opposed to considering just the target forecast that matters, i.e. the last one) using the exact same setup as reported in the paper are enumerated in the Table below. We note that the results of TimeMixer are pending due to time constraints, but its performance is not expected to be the most competitive based on the results in our paper. It can be observed that, on average, DeformTime still outperforms other models. Crossformer and ModernTCN remain the best-performing baselines.
>
> | Regions  | H  | DeformTime | ModernTCN | CycleNet | TimeXer | PatchTST | iTransformer | Crossformer | LightTS | DLinear |
> |----------|----|------------|-----------|----------|---------|---------|--------------|-------------|---------|--------|
> | ILI-ENG | 7  | **1.2802** | 1.7351    | 2.0997   | 1.7086  | 1.9402  | 1.7746       | 1.3243    | 1.9713  | 2.1892 |
> | ILI-ENG | 14 | **1.6521** | 2.2656    | 2.6170   | 1.9703  | 2.4857  | 2.3660       | 1.9326    | 2.3676  | 2.7452 |
> | ILI-ENG | 21 | **2.3246** | 2.3331  | 3.2527   | 2.8852  | 2.8458  | 3.0904       | 2.8733      | 2.7742  | 3.2983 |
> | ILI-ENG | 28 | 2.7829   | **2.5948**| 3.6091   | 2.7894  | 3.2161  | 3.4784       | 3.0740     | 3.2559  | 3.6805 |
> | ILI-US2 | 7  | **0.4142** | 0.4523    | 0.6974   | 0.4931  | 0.6400  | 0.5455       | 0.4444      | 0.4323| 0.7214 |
> | ILI-US2 | 14 | **0.4544** | 0.5501    | 0.7601   | 0.5402  | 0.7302  | 0.6357       | 0.4695    | 0.4723  | 0.7899 |
> | ILI-US2 | 21 | **0.5374** | 0.5576  | 0.8130   | 0.6190  | 0.7489  | 0.6554       | 0.5720     | 0.6432  | 0.8642 |
> | ILI-US2 | 28 | **0.5554** | 0.5892    | 0.8729   | 0.6808  | 0.8450  | 0.7370       | 0.5842    | 0.6886  | 0.8920 |
> | ILI-US9 | 7  | **0.2614** | 0.2945  | 0.4126   | 0.3435  | 0.3882  | 0.3725       | 0.3222      | 0.3078  | 0.4348 |
> | ILI-US9 | 14 | **0.3040** | 0.3398    | 0.4584   | 0.3715  | 0.4334  | 0.4049       | 0.3254    | 0.4322  | 0.4780 |
> | ILI-US9 | 21 | **0.3187** | 0.3558    | 0.5224   | 0.4430  | 0.4664  | 0.4504       | 0.3285    | 0.4385  | 0.5315 |
> | ILI-US9 | 28 | 0.3913   | **0.3786**| 0.5727   | 0.4774  | 0.5002  | 0.5535       | 0.4018      | 0.5665  | 0.5622 |
>
> *Table: Forecasting accuracy results (MAE) using the entire output sequence (as opposed to using the last prediction only at the target forecasting horizon) across all tasks and methods. $H$ denotes the forecasting horizon. Errors are averaged across all $4$ test seasons for each region.*
>
> Specifically, DeformTime reduces the averaged MAE by $3.4\\%$ across all forecasting horizons and data sets. When compared to a specific forecasting model, DeformTime reduces the MAE of the best-performing baseline (in this case it is Crossformer) by $8.2\\%$ and the second best baseline (ModernTCN) by $9.2\\%$.
> Notably, the reviewers may have observed a slightly altered ranking of methods for some forecasting horizons in these results. This aligns with our earlier discussion about the difference between measuring error at the target horizon vs. averaging errors across the entire output sequence.
> We note that these new results do not necessarily reflect on which model performs best in the ILI rate forecasting task, as the accuracy of a prediction at the target forecasting horizon (i.e. the last output of a predicted sequence) is the primary metric of interest for this task.
> Nevertheless, for completeness, we will update the results to section E.8 and Table S6 in our revised version. We hope this helps the reviewers gain a clearer understanding of DeformTime's forecasting capacity and reach a consensus about the utility of our proposed method.

---

> > ### Comment · Reviewer_rgws · 2025-03-07
> > **Reply to the authors**
> >
> > I would like to thank the authors for their effort to make the paper more convincing. I have no more questions and submitted my recommendation.

---

> > > ### Author Response · Authors · 2025-03-07
> > >
> > > We thank the reviewer for acknowledging our rebuttal. However, we noticed that the recommendation for claims and evidence is marked as “No.” We kindly seek further clarification on what additional evidence would be required to better support our claims.

---

> > > > ### Comment · Reviewer_rgws · 2025-03-07
> > > >
> > > > Not sure you should have the permission to see that. On my side, I see a different answer than yours.

---

> > > > > ### Author Response · Authors · 2025-03-07
> > > > >
> > > > > We thank the reviewer for their active engagement during the discussion period and acknowledging our rebuttal. The misunderstanding potentially comes from that we were unable to view the final recommendation on our end.

---

### Review · Reviewer_Z84T · 2025-01-29

**Summary Of Contributions:**

This paper addresses the multivariate time series forecasting problem by leveraging information from exogenous variables. The authors introduce two core components: the Variable Deformation Attention Block and the Temporal Deformable Attention Block. The proposed architecture is evaluated against numerous baselines across six datasets and demonstrates strong performance.

**Audience:**

Yes

**Claims And Evidence:**

No

**Requested Changes:**

Please see above.

**Strengths And Weaknesses:**

**Strengths**

-  S1. Tackling time series forecasting with exogenous variables is relevant, especially since most recent literature has focused on purely autoregressive approaches. The application of deformable attention blocks to time series forecasting appears to be both novel and well-motivated.

-  S2. The paper is clearly written and easy to follow, except for the section on the variable deformation attention block. The numerous ablation studies and the detailed appendix effectively highlight the contribution of each architectural component.

-  S3. The experimental results are strong, even though the evaluation setting differs slightly from the traditional forecasting setup. Instead of assessing performance over an entire prediction window, the model is evaluated on a pointwise forecast at horizon H.


**Weaknesses**

-  W1. The T-DAB mechanism is difficult to follow, particularly the interpretation of Δp in Section 3.3. Is the deformable patch a transformation of the currently considered patch, or is it relative to a neighboring patch? In other words, can Δp represent indices from other patches? Clarifying this section with additional explanations and possibly some figures would be helpful.

-  W2. The paper considers only multivariate datasets of the form C×T, where C represents different channels (e.g., temperature, humidity for weather) and T is the number of timestamps. While this setting is interesting, the authors should discuss its specificity compared to other common setups, such as datasets structured as N×T (where N represents different samples, like households in the Electricity dataset or roads in the Traffic dataset). Have you evaluated your method against baselines like PatchTST on these alternative dataset structures (Electricity or Traffic) ?

-  W3. In the discussion on DLinear and the forecasting setup, the paper states:
“That may be relevant for some tasks, but it is not relevant to tasks where forecasting H time steps ahead really necessitates obtaining an accurate forecast H time steps ahead (and that should be the case for most, if not all, forecasting tasks).”
I have two concerns regarding this statement:
    - I do not fully understand the reasoning behind it. Could you clarify?
    - Could you provide forecasting results using the traditional evaluation setting (evaluating over the entire horizon) for the ETT and weather datasets?

---

> ### Author Response · Authors · 2025-02-02
>
> We thank the reviewer for acknowledging that our method is novel and well-motivated, the paper is well-written, and the experiments are strong and comprehensive. Below we address the identified weaknesses.
>
> **W1:**
> All deformations are conducted within each time series patch. $\Delta \mathbf{p}$ is the positional offset that the model learns from the input patch, which can be added to position $\mathbf{p}=(i,j)$ to form the deformed position $\mathbf{p}+\Delta \mathbf{p}$ on the original latent time series patch $\mathbf{Z}\_\texttt{p}$. For V-DAB, $\Delta \mathbf{p}$ contains positional deformation over both time dimension and variable dimension. For T-DAB, $\Delta p$ contains the positional deformation only on the temporal dimension. Both deformations are conducted within each time series patch.
>
> We also provide a detailed explanation of how deformation (using $\Delta \mathbf{p}$) works in practice in our response to all reviewers at the top of this page (see section titled **Deformation using V-DAB and T-DAB**).
>
> We thank the reviewer for their suggestion and we will clarify these elements better in our revised manuscript.
>
> **W2:**
> We acknowledge that in the recent time series forecasting literature, there are some commonly used tasks / data sets like Electricity or Traffic where they consider data obtained from different households / roads as their variables. We justify our choice of data sets for assessing DeformTime in Appendix A.4. We also introduce another task for evaluating DeformTime (ILI across $3$ different regions). For these specific $2$ data sets that we do not use in our experiments (Electricity and Traffic), we note that they do not have one explicitly defined target variable. This would not make them directly compatible with our forecasting task, where we only consider / model one target variable. To conduct multitask learning (multiple target variables), DeformTime would require methodological modifications since it shares weights across all variables in the latent space (i.e. this out-of-scope for the claims of this particular work and could be addressed in follow-up work).
>
> **W3:**
> We thank the reviewer for highlighting this. We provide extensive clarification and results in our common response to all reviewers (at the very top titled **MAE at the target forecasting horizon output ($y_{t+H}$) vs. MAE on the entire output series ($y_{t+1},\dots,y_{t+H}$)**).

---

> ### Comment · Reviewer_Z84T · 2025-02-13
> **Answer**
>
> I thank the authors for their responses.
>
> - I appreciate your clarification regarding the relevance of $\Delta p$ and its definition. However, I find the argument that one must search within the patch to be debatable. Indeed, in the previous step, variables are grouped based on linear correlation, which also assumes that the time series are properly aligned from a correlation perspective. However, one can easily imagine cases where this is not true, leading to relationships being captured between variables that either do not exist or are weaker than those between variables that are linearly close to the target variable.
>
> - Secondly, regarding the prediction of a window rather than a single point, I appreciate the authors' responses. Nevertheless, I would like to suggest a more nuanced perspective:
>    - In several datasets, predicting closer points is not necessarily easier than predicting farther ones. For example, in traffic or electricity data, there are peaks of activity and low-activity periods that are easier to predict than others (as the outcomes are less variable).
>    - Predicting an entire window can be useful, for instance, in the case of a wind farm, where one needs to estimate the total production for the next day based on forecasted wind speeds in order to size the rest of the power generation accordingly. In such a case, a pointwise prediction would be of little use.
>
> - Last, I appreciate your clarification regarding the feasibility and infeasibility of forecasting as stated in the paper. I had understood this upon my initial reading, but I sought confirmation. I believe it would be beneficial to make this aspect more explicit, particularly in the arguments and the abstract, as this paper tackles a specific forecasting framework.

---

> > ### Author Response · Authors · 2025-02-13
> >
> > We thank the reviewer for acknowledging our clarifications.
> >
> > >I appreciate your clarification regarding the relevance of $\Delta{p}$ and its definition. However, I find the argument that one must search within the patch to be debatable. Indeed, in the previous step, variables are grouped based on linear correlation, which also assumes that the time series are properly aligned from a correlation perspective. However, one can easily imagine cases where this is not true, leading to relationships being captured between variables that either do not exist or are weaker than those between variables that are linearly close to the target variable.
> >
> > We thank the reviewer for raising this important point.
> > We would like to note that the inter-variable dependencies are captured over $\mathbf{Z}\_\texttt{p} \in \mathbb{R}^{\ell \times d}$, where $d$ is the latent variable dimension that inclusively contains information from all variables mapped by NAE. The receptive field of DeformTime is determined directly with the kernel size $k$ of $\theta\_\texttt{off}$ as we clarified in section 3.2 and is also influenced by the choice of grouping. While DeformTime can capture variables that are less correlated with an increased $k$ or a reduced number of groups in NAE, we agree with the reviewer that this is potentially a limitation of DeformTime. However, there potentially exists a trade-off between allowing more variables to be able to be captured by V-DAB and the model being overfitted. We empirically show in our ablation study that removing the NAE (Table 2) and using an overly small number of groups (Table S4) both lead to an inferior performance. We will consider expanding the model's capacity to capture less correlated features in our future work.
> >
> > We thank the reviewer for pointing this out and we will make this clearer in our revised version.
> >
> > >Secondly, regarding the prediction of a window rather than a single point, I appreciate the authors' responses. Nevertheless, I would like to suggest a more nuanced perspective:
> > >- In several datasets, predicting closer points is not necessarily easier than predicting farther ones. For example, in traffic or electricity data, there are peaks of activity and low-activity periods that are easier to predict than others (as the outcomes are less variable).
> > >- Predicting an entire window can be useful, for instance, in the case of a wind farm, where one needs to estimate the total production for the next day based on forecasted wind speeds in order to size the rest of the power generation accordingly. In such a case, a pointwise prediction would be of little use.
> >
> > We agree with the reviewer that it is also valuable to evaluate the model performance over the sequence and we provide the experiment results accordingly in our reply (please see the comment to all reviewers in **MAE at the target forecasting horizon output ($y\_{t+H}$) vs. MAE on the entire output series ($y\_{t+1},\dots,y\_{t+H}$)**). Our results show that DeformTime remains the best-performing model among the more competitive baselines when evaluated over the sequence. We have also updated the results in the revised version in Appendix E.8. We sincerely hope that this clarifies the concerns raised by the reviewer.
> >
> > >Last, I appreciate your clarification regarding the feasibility and infeasibility of forecasting as stated in the paper. I had understood this upon my initial reading, but I sought confirmation. I believe it would be beneficial to make this aspect more explicit, particularly in the arguments and the abstract, as this paper tackles a specific forecasting framework.
> >
> > We thank the reviewer for their suggestion and we will further clarify the task definition of DeformTime better in the abstract and introduction section in our revised version.

---

> > > ### Author Response · Authors · 2025-03-07
> > >
> > > We thank the reviewer for their suggestions and acknowledge our rebuttal. However, we noticed that the recommendation for claims and evidence is marked as “No.” We kindly seek further clarification on what additional evidence would be required to better support our claims.

---

### Review · Reviewer_EKtq · 2025-01-31

**Summary Of Contributions:**

- The authors propose a novel time-series prediction framework that can utilize the exogenous predictors efficiently so that it can capture inter- and intra-variable dependencies.
- The authors tackled a specific problem setting where the model predicts a single target variable using disjoint exogenous predictors.
- The experimental results show that the proposed method show better results than the alternatives (with lower computational complexity).

**Audience:**

Yes

**Claims And Evidence:**

Yes

**Requested Changes:**

1. Extension to multiple output prediction
- As the authors mentioned in Section 2, the authors mainly focus on the setting where the inputs are multivariate but the model only predicts "one" output variable.
- Could we expand the proposed method to multivariate output predictions? If yes, please provide some procedure to expand and provide some "empirical results".
- If we can easily do that, it would significantly improve the impact of this paper because in practice, there are many applications where we need multiple variable predictions simultaneously.

2. Inputs and targets variables
- I agreed that there are some settings where target variable and input variables are different (i.e., Q and y are different).
- But in many cases, Q and y are overlapping. Could we easily expand this method to the setting where Q and y are overlapping? If yes, please provide some procedure to expand and provide some "empirical results".

3. Methods
- As far as I know, there are various other methods which can also capture both temporal relationship as well as inter-variable relationship. (A straightforward example is RNN). In that point of view, what is the main contribution of this work?

**Strengths And Weaknesses:**

Strength:
- The experimental results are convincing. The authors utilize various datasets and ablation studies.
- The proposed method is intuitive and the paper is easy to follow.

Weakness:
- The experimental settings are somewhat less practical (using disjoint exogenous predictors to forecast "single" target variables).
- If the authors can provide the extension of the proposed works that can expand the experimental settings (including multiple target variable predictions as well as target variables are also included in the predictors), that makes this paper more impactful.

---

> ### Author Response · Authors · 2025-02-02
>
> We thank the reviewer for acknowledging that our method is novel, our experimental results are comprehensive and convincing, the motivation is clear and the paper is easy to follow. We also thank the reviewer for their thoughtful feedback. We appreciate their suggestions on increasing the paper's impact and we attempt to address their concerns below.
>
> >**Weakness 1**
> >The experimental settings are somewhat less practical (using disjoint exogenous predictors to forecast "single" target variables).
>
> Our work focuses on using exogenous variables (external predictors) along with the past data of the target variable as the primary predictors of a single target variable. This has been a common setup in many forecasting tasks [1, 2]. Examples include:
> - In clinical applications, external risk factors including poverty and smoking are key to predicting the occurrence of coronary heart disease [3].
> - In epidemiology, external factors (e.g., weather, vaccination rates, mobility data, search trends) are key to predicting disease spread [4-6].
> - In day-ahead electricity price forecasting, external variables like wind and solar energy generation serve as vital indicators for price [7, 8].
> - In retail, companies predict sales of a specific product using other factors like promotions, weather or social media trends [9, 10].
>
> We clarify that this paper exclusively focuses on multivariable forecasting (multiple input variables and one target variable) as opposed to multivariate forecasting (multiple input variables and multiple output variables) [11]. We do not make any claims about tasks that may have multiple output variables (multivariate forecasting).
>
> While we don't make claims about DeformTime's capacity to conduct multivariate forecasting, we acknowledge the equal importance of multivariate regression in the field of forecasting and we thank the reviewer for their suggestion. We will consider this in future work.
>
> **References**
>
> 1. Hyndman, R. J. (2018). Forecasting: principles and practice. Chapter 9.
> 2. Box, George EP, et al. (2015). Time series analysis: forecasting and control.
> 3. Katz, M. H. (2003). Multivariable analysis: a primer for readers of medical research.
> 4. Da Silva, Ramon Gomes, et al. (2020). Forecasting Brazilian and American COVID-19 cases based on artificial intelligence coupled with climatic exogenous variables.
> 5. Dugas, Andrea Freyer, et al. (2013). Influenza forecasting with Google flu trends.
> 6. Parino, Francesco, et al. (2025). Integrating dynamical modeling and phylogeographic inference to characterize global influenza circulation.
> 7. Olivares, Kin G., et al. (2023). Neural basis expansion analysis with exogenous variables: Forecasting electricity prices with NBEATSx.
> 8. Lago, Jesus, et al. (2021). Forecasting day-ahead electricity prices: A review of state-of-the-art algorithms, best practices and an open-access benchmark.
> 9. Fildes, Robert, Shaohui Ma, and Stephan Kolassa (2022). Retail forecasting: Research and practice.
> 10. Vosen, Simeon, and Torsten Schmidt (2011). Forecasting private consumption: survey‐based indicators vs. Google trends.
> 11. Hidalgo, Bertha, and Melody Goodman (2013). Multivariate or multivariable regression?

---

> ### Author Response · Authors · 2025-02-02
>
> >**Weakness 2**
> >If the authors can provide the extension of the proposed works that can expand the experimental settings (including multiple target variable predictions as well as target variables are also included in the predictors), that makes this paper more impactful.
>
> We clarify that the experiment settings of this paper, including DeformTime and all compared baseline models, are in accordance with the suggested setup of the reviewer, where the values of the target variables within the past time steps are included as part of the input. Please see the clarification of the definition of $\mathbf{Q}$ and $\mathbf{y}$ in our response to the requested changes. The experimental results show that DeformTime reduces the averaged MAE score by $7.4\\%$ compared to the best baseline models.
>
> Predicting multiple variables is out-of-scope in our work. We make no claims about this. Please also see our response to predicting multiple target variables in the requested changes below.
>
>
> >**Requested change 1**
> >Extension to multiple output prediction
> >- As the authors mentioned in Section 2, the authors mainly focus on the setting where the inputs are multivariate but the model only predicts "one" output variable.
> >- Could we expand the proposed method to multivariate output predictions? If yes, please provide some procedure to expand and provide some "empirical results".
> >- If we can easily do that, it would significantly improve the impact of this paper because in practice, there are many applications where we need multiple variable predictions simultaneously.
>
> It is obvious that the proposed method can be adapted to predict multiple target variables simultaneously by treating each variable as an individual target and retuning the model for each one. However, we would like to politely clarify that multitask learning is out-of-scope for the proposed method. We make no claims for multivariate time series forecasting settings (DeformTime is a multivariable time series forecasting method).
>
> >**Inputs and targets variables**
> >- I agree that there are some settings where target variable and input variables are different (i.e., Q and y are different).
> >- But in many cases, Q and y are overlapping. Could we easily expand this method to the setting where Q and y are overlapping? If yes, please provide some procedure to expand and provide some "empirical results".
>
> We would like to refer the reviewer back to section 2 of our paper where the forecasting task is clearly defined. The input space contains both exogenous predictors (denoted by $\mathbf{Q}\_t$) and past values of the target variable (denoted by $\mathbf{y}\_{t-\delta}$ where $\delta$ is a delay that is relevant only to the ILI forecasting task and is set to $0$ otherwise). Hence, one dimension (or variable) of the input space contains the autoregressive signal (past values of the target variable). Hence, our current empirical results cover the comment of the reviewer. We will try to improve the readability of section 2 to make this clearer.
>
> >**Main contribution**
> >- As far as I know, there are various other methods which can also capture both temporal relationship as well as inter-variable relationship. (A straightforward example is RNN). In that point of view, what is the main contribution of this work?
>
> From a model design perspective, DeformTime facilitates **adaptive** focus across time and variables, which is limited in prior deep-learning models. For example, RNN can only capture the inter-variable dependencies recursively across each time step. Our experiment results show that DeformTime outperforms previous state-of-the-art forecasting models that capture inter-variable dependencies (ModernTCN, TimeXer, iTransformer, TimeMixer, Crossformer, and LightTS) on various time series forecasting tasks, reducing their MAE score by at least $8.6\\%$ on average.

---

> > ### Comment · Reviewer_EKtq · 2025-02-07
> > **Response to the rebuttal**
> >
> > Thanks a lot for providing detailed responses to my comments.
> > Also, thanks for clarifying my misunderstanding on Q and y. Now I got that point.
> >
> > So, in summary, the authors think that it can expand to multiple output predictions; however, the authors did not provide evidence on this extension. Thus, with this paper, the claims are mainly focused on using multivariate inputs and single output predictions. I got that points and scope of this paper. Will incorporate this point in my final recommendation.
> >
> > For RNN, I think RNN can capture both inter and intra variable dependencies because the current hidden states are determined by current inputs and previous hidden states (which include both inter and intra relationships). So, I am not convinced on your last point on the rebuttal.
> >
> > However, in general, the response is clear and I am confident to provide my final recommendation to this paper. Thanks again.

---

### Author Response · Authors · 2025-02-02

### **MAE at the target forecasting horizon output ($y_{t+H}$) vs. MAE on the entire output series ($y_{t+1},\dots,y_{t+H}$)**

We thank reviewers **rgws** and **Z84T** for raising the important point of why we are measuring forecast error using only the output at the target forecast horizon, e.g. when forecasting $H$ time steps ahead, we use output $y_{t+H}$ to evaluate the forecast accuracy as opposed to convoluting the error with outputs $y_{1},y_{2},\dots,y_{t+H-1}$. The reason behind this is that forecasting errors tend to increase as the forecasting horizon extends and the forecasting task becomes harder [1], and models may perform differently at short versus long horizons. Averaging the error across all the outputs may favour forecasters that are very accurate early on, but very inaccurate closer to and at the target forecasting horizon. For example, let's consider a toy example where we aim to forecast $3$ time steps ahead. We have $2$ forecasting models. The first one yields MAEs equal to $\{1, 5, 9\}$ for its forecasts at time steps $\{1, 2, 3\}$, respectively. The second one yields MAEs of $\{4, 5, 6\}$. Both models will yield the same average MAE across all the output forecasting time steps (equal to $5$). However, the second model is more accurate at the target forecasting horizon $3$, i.e. at the actual forecasting task we are trying to propose a solution for.

In fact, we can argue that by considering the entire output series (and doing so in a uniform way for all output time steps) we are not assessing the accuracy of the forecasting function at the specific target forecasting horizon. We note that this is an important issue in some of the recent time series forecasting literature, but the community has already begun to flag this and other caveats [2].

As clarified in sections 2 and D.7, we measure forecasting accuracy at each target horizon. This is a common evaluation strategy [1, 3-5] in established forecasting literature as it assesses a model's capacity to predict at the said horizon. From a practical perspective, this is essential in (most if not all) real-world applications and scenarios, where there is an expectation and necessity to make as accurate predictions as possible at the specified target forecasting horizon [6, 7].

**References**

1. Hyndman, R. J., \& Koehler, A. B. (2006). Another look at measures of forecast accuracy.
2. Hewamalage, H., K. Ackermann, and C. Bergmeir. (2022). Forecast Evaluation for Data Scientists: Common Pitfalls and Best Practices.
3. Hyndman, R. J. (2018). Forecasting: Principles and Practice. Chapter 3.
4. Isiklar, G., \& Lahiri, K. (2007). How far ahead can we forecast? Evidence from cross-country surveys.
5. Kang, I. B. (2003). Multi-period forecasting using different models for different horizons: an application to US economic time series data.
6. Verma, Y., Heinonen, M., \& Garg, V. (2024). ClimODE: Climate and Weather Forecasting with Physics-informed Neural ODEs.
7. Taylor, J. W., \& McSharry, P. E. (2007). Short-Term Load Forecasting Methods: An Evaluation Based on European Data.

---

> ### Author Response · Authors · 2025-02-02
>
> | Dataset     | DeformTime | ModernTCN  | CycleNet   | TimeXer | PatchTST   | iTransformer | DLinear    |
> | ----------- | ---------- | ---------- | ---------- | ------- | ---------- | ------------ | ---------- |
> | ETTh1-96    | 0.1840     | 0.1774     | **0.1746** | 0.1807  | 0.1757     | 0.1825       | 0.2045     |
> | ETTh1-192   | **0.1988** | 0.2013     | 0.1990     | 0.2043  | 0.1999     | 0.2051       | 0.2834     |
> | ETTh1-336   | **0.2024** | 0.2123     | 0.2189     | 0.2226  | 0.2248     | 0.2322       | 0.3877     |
> | ETTh1-720   | 0.2390     | **0.2288** | 0.2367     | 0.2292  | 0.2500     | 0.2449       | 0.5007     |
> | ETTh2-96    | 0.2924     | 0.2790     | 0.2849     | 0.2797  | **0.2765** | 0.2991       | 0.2840     |
> | ETTh2-192   | **0.3010** | 0.3183     | 0.3223     | 0.3331  | 0.3174     | 0.3463       | 0.3313     |
> | ETTh2-336   | **0.3172** | 0.3350     | 0.3536     | 0.3769  | 0.3417     | 0.3857       | 0.4056     |
> | ETTh2-720   | **0.3577** | 0.4259     | 0.3976     | 0.4069  | 0.3887     | 0.3945       | 0.5391     |
> | Weather-96  | 0.0226     | 0.0235     | 0.0228     | 0.0270  | **0.0204** | 0.0218       | 0.0206     |
> | Weather-192 | **0.0237** | 0.0265     | 0.0257     | 0.0294  | 0.0239     | 0.0253       | 0.0239     |
> | Weather-336 | **0.0258** | 0.0296     | 0.0279     | 0.0312  | 0.0259     | 0.0269       | 0.0261     |
> | Weather-720 | 0.0308     | 0.0365     | 0.0333     | 0.0355  | 0.0316     | 0.0312       | **0.0307** |
>
> *Table 1: Forecasting MAE on the entire output sequence for the ETT and weather tasks.*
>
> Nevertheless, we are also considerate of the position of the reviewers. To alleviate any concern about the evaluation, we also provide results where we have used the entire forecasting output time series to compute MAE(see Table 1). We have performed this for the established benchmark tasks **ETT** and **weather**, wherein baselines are more competitive in comparison to DeformTime (DeformTime yields far superior performance on the ILI forecasting tasks). DeformTime outperforms baselines on $7$ out of $12$ tasks.
>
> The best-performing baseline when computing MAE on the entire output time series is PatchTST. DeformTime reduces PatchTST's MAE by $1.5\\%$ on average in the ETT and weather forecasting tasks. Comparing that to obtaining MAE based on the forecasts at the target forecasting horizon only, DeformTime reduces PatchTST MAE by $9.8\\%$ (again on the ETT and weather forecast tasks; reduction is always greater when it comes to the ILI forecasting task).
>
> Importantly, CycleNet was the best-performing baseline (in the ETT and weather tasks) when evaluating the output at the target forecasting horizon. So, by this observation alone, it is evident that while CycleNet is producing more accurate forecasts at the target horizon compared to PatchTST (a.k.a. makes better forecasts), when considering the entire series of forecast outputs from time step $t+1$ to time step $t+H$ (where $H$ determines the target forecasting horizon), it yields an inferior average MAE. The overall changes in the ranking of baselines consolidate our argument that convoluting the error at the target forecasting horizon with errors in time steps prior to that can produce a distorted picture of the actual forecasting capacity of a model.
>
> In any occasion, no matter how we measure error, DeformTime still yields the best performance on average. We are happy to expand our current discussion on this (see section D.7), incorporating elements from our response in the revised manuscript.

---

> ### Author Response · Authors · 2025-02-02
>
> ### **Deformation using V-DAB and T-DAB**
>
> Reflecting on some shared comments from the reviewers (thank you), we would like to clarify the following elements further.
>
> **Term definitions.** $\Delta \mathbf{p}$ is the positional offset the model learns for each input. It can be added to position $\mathbf{p} = (i, j)$ to obtain a deformed sampling position $\mathbf{p} + \Delta \mathbf{p}$. This dynamically adjusts the sampling position to either adjacent time steps (in T-DAB) or other variables (in V-DAB). $\mathbf{Z}\_\texttt{p} \in \mathbf{R} ^ {\ell \times d}$ is the latent time series patch after the segmentation transformation of the embedding $\mathbf{Z}\_\texttt{e}$.
>
> **Clarifying motivations better.** Input variables in an MTS forecasting task may be correlated with past values of the target variable (we capture this using T-DAB) or other input variables within proximal time steps (we capture this using V-DAB). In sections 3.3 and 3.4, the positional offset decides the deformed sampling position $\mathbf{p} + \Delta \mathbf{p}$. The deformed patch, $\mathbf{Z}\_\texttt{d}$, is then sampled from $\mathbf{Z}\_\texttt{p}$ over this deformed position.
>
> For example, in V-DAB, suppose we have an element $\mathbf{Z}\_\texttt{p}(m,n)$ that was originally on position $\mathbf{p} = (m,n)$ of the patch matrix $\mathbf{Z}\_\texttt{p}$. Let's assume that $m = 5$ and $n = 10$ in this example. Let's also assume that the learned positional offset for that position is $(\Delta m, \Delta n) = (-0.1, 1.5)$. Then, the value for $\mathbf{Z}\_\texttt{d}(m,n)$ is the value sampled from $\mathbf{Z}\_\texttt{p}(m+\Delta m,n+\Delta n)$ or, in this example, $\mathbf{Z}\_\texttt{p}(5-0.1,10+1.5)$ with bilinear interpolation. Specifically, the deformed point $\mathbf{p}=(4.9,11.5)$ lies within a $2 \times 2$ index matrix of actual discrete positions, i.e. $(4,11)$, $(4,12)$, $(5,11)$, and $(5,12)$. We first linearly interpolate the values across the rows of the index matrix. We note that the deformed position in this example is equal to $11.5$ and that means that there is an equal distance between the indices $11$ and $12$. We therefore perform:
>
> $\mathbf{Z}\_\texttt{p}(4,11.5) = \mathbf{Z}\_\texttt{p}(4,11) \times 0.5 + \mathbf{Z}\_\texttt{p}(4,12) \times (1-0.5)$
>
> and
>
> $\mathbf{Z}\_\texttt{p}(5,11.5) = \mathbf{Z}\_\texttt{p}(5,11) \times 0.5 + \mathbf{Z}\_\texttt{p}(5,12) \times (1-0.5)$ .
>
> Then, we conduct linear interpolation along the columns to obtain the final value of the bilinear interpolation. Here we note that the deformed position is equal to $4.9$ and hence it does not have the same distance between indices $4$ and $5$ (linear interpolation will use this). We therefore perform:
>
> $\mathbf{Z}\_\texttt{p}(4.9,11.5) = \mathbf{Z}\_\texttt{p}(4,11.5) \times 0.1 + \mathbf{Z}\_\texttt{p}(5,11.5) \times (1-0.1)$ .
>
> This operation is conducted on every $\mathbf{p} \in \mathbf{Z}\_\texttt{p}$ to obtain $\mathbf{Z}\_\texttt{d}$.
>
> This then forms the key ($\mathbf{K}$) and value ($\mathbf{V}$) embeddings with learnable weight matrices. Given a scalar element $z\_{(i,j)} \in \mathbf{Z}\_\texttt{p}$, $\mathbf{K}$ and $\mathbf{V}$ contain elements sampled from the neighbouring latent variables ($\mathbf{z}\_{i-\alpha}$, $\mathbf{z}\_{i+\alpha}$) in V-DAB or neighbouring time steps $(z\_{i,j-\alpha},z\_{i,j+\alpha})$ in T-DAB, where $\alpha$ is the offset amplitude.
>
> With these positional offsets, we conduct cross-attention over the original latent variable (Query embedding) and the deformed latent variable ($\mathbf{K}$, $\mathbf{V}$). Specifically, elements at position $\mathbf{p}$ can attend to elements at position $\mathbf{p} + \Delta \mathbf{p}$. This guarantees that the model does not treat neighbouring variables / time steps in isolation with attention but instead considers their interactions.

---

### Author Response · Authors · 2025-02-04

We thank all reviewers for their constructive feedback. In addition to providing responses to address all the concerns raised by the reviewers, we would like to make the reviewers aware that we have also revised our paper accordingly.

Specifically:

1. We thank reviewers **rgws** and **Z84T** for their valuable suggestions to clarify the motivation behind DeformTime. In response, we have explicitly articulated the motivation in sections 3.2, 3.3, and 3.4. Additionally, we provide a more detailed explanation of how deformation works in practice (in the V-DAB module) with an example (Appendix B).

2. We also thank the reviewers for raising the important point about the way we measure forecast accuracy (using the target horizon forecasts at time step $t+H$ vs. convoluting errors using the entire output series $t+1,\dots,t+H$). We have added supplementary results and clarifications (that we also have provided below in our response) in Appendices E.7 and E.8.

3. We appreciate reviewer **rgws** for pointing out the potential ambiguity in the task definition. We have improved our task description in section 2. We have also improved our description of the NAE, V-DAB, and T-DAB modules (sections 3.2, 3.3, and 3.4).

4. We thank reviewer **EKtq** for their question on the experiment setup in section 4.4 and we have added a detailed description regarding the selection process of the exogenous variables in section 4.4 for improved clarity.

5. We have also made minor changes throughout the manuscript (to address other minor points made by the reviewers). All revisions can be viewed by using the OpenReview functionality to compare our revised submission with our original manuscript.

---

### Decision · Action_Editor_SPbV · 2025-03-14

**Recommendation:** Accept with minor revision

**Comment:**

The paper proposes DeformTime, a transformer-based architecture for multivariable time series forecasting. Specifically, given a number of exogenous variables and a single endogenous variable, DeformTime predicts future values of the endogenous variable. The paper’s main contribution is the design of two types of deformable attention blocks (DABs): temporal DABs for intra-variable dependencies and variable DABs for inter-variable dependencies. Ablation studies and baseline comparisons validate this contribution on established and new benchmark tasks.

The paper initially received mixed reviews. While the reviewers acknowledged the paper’s novel perspective on deformable attention blocks, comprehensive experiments, and promising results, they also raised important concerns. The presentation of the deformable attention blocks (T-DAB, V-DAB) and DeformTime's evaluation at a single forecasting horizon, in particular, spurred discussions. The authors responded well to this criticism and provided an insightful DAB example, an evaluation of DeformTime over entire forecasting windows, and motivation/intuition for key architectural components. The scope of DeformTime, with its single target variable but multiple exogenous predictors, also caused confusion, both presentation-wise and regarding the applicability of the proposed framework overall. While multivariate predictions remain out of scope, the authors reworked their presentation and emphasized DeformTime’s focus on a single target. Some concerns could not be resolved during the discussion phase, most importantly the consequences of grouping variables based on linear correlation. The authors agreed that this might indeed be a limitation, although at least from an empirical standpoint the negative effects can likely be contained.

Ultimately, the majority of reviewers recommended acceptance and I agree with this sentiment. However, it is paramount that the important clarifications made during the discussion phase find their way into the paper; they are a prerequisite for acceptance. While some of these updates have already been incorporated in the latest revision, others have not.

The paper can be accepted under the following conditions:

- Improvements to the positioning of the paper, especially concerning the paper’s focus on a single target variable.
- Improvements to the separation of deformable attention blocks as a prior concept and the specific innovations of this paper.
- Improvements to the technical descriptions of V-DAB (3.3) and T-DAB (3.4), including motivation and intuition. The latest revision is a good step, but I highly encourage the authors to take a third look.
- Inclusion of all experiments related to DeformTime’s evaluation over entire prediction windows (ETT, weather, and ILI).
- Inclusion of the discussion points (positive and negative) related to grouping variables based on linear correlation.
- Inclusion of ILI-US2 in Table 2 (ablation study), with a structure identical to ILI-ENG and ILI-US9.

**Audience:**

Time series forecasting has a long history in the machine learning community and the relationship between exogenous and endogenous variables is of importance in many practical scenarios. The proposed framework models these dependencies with an interesting variant of a transformer-based architecture, a concept which has been highly influential. The time series community should therefore appreciate the perspective presented in this paper.

**Claims And Evidence:**

The claims made in this paper are supported by comprehensive baseline comparisons and ablation studies, both on established and new benchmarks. The experiment protocol is detailed and the supplementary material contains additional information about datasets, baselines, and settings.

---

> ### Author Response · Authors · 2025-03-31
>
> >Improvements to the positioning of the paper, especially concerning the paper’s focus on a single target variable.
>
> We thank the AE for their suggestion. We have improved the clarity of our task definition in both the abstract, the introduction, and section 2.
>
>
> >Improvements to the separation of deformable attention blocks as a prior concept and the specific innovations of this paper.
>
> We thank the AE for their insightful comment on further clarifying the novelty of DeformTime. To address this, we have added additional commentary that highlights DeformTime’s novelty in the introduction section.
>
>
> >Improvements to the technical descriptions of V-DAB (3.3) and T-DAB (3.4), including motivation and intuition. The latest revision is a good step, but I highly encourage the authors to take a third look.
>
> We thank the AE for this suggestion. We have further clarified the motivation behind the design of V-DAB and T-DAB in Sections 3.3 and 3.4, respectively, with an improved clarity on why these modules can capture the inter- and intra-variable dependencies.
>
>
> >Inclusion of all experiments related to DeformTime’s evaluation over entire prediction windows (ETT, weather, and ILI).
>
> The experiments evaluating DeformTime over entire prediction windows on all data sets have been added in the appendix (Section E.8). We further added detailed discussion according to the updated results.
>
>
> >Inclusion of the discussion points (positive and negative) related to grouping variables based on linear correlation.
>
> We thank the AE for their suggestion in discussing this limitation of DeformTime (reliance on grouping), and we have added further discussion about this in Section E.4.
>
>
> >Inclusion of ILI-US2 in Table 2 (ablation study), with a structure identical to ILI-ENG and ILI-US9.
>
> We thank the AE for this constructive suggestion. We have added the ablation study for ILI-US2 in Table 2, using the same experimental setup as for ILI-ENG and ILI-US9. Additionally, we have reanalysed the effect of different modules in Section 4.3, and our conclusions remain consistent with the one prior to adding these experiments.